*Resource*

EMBO
Molecular Medicine

# Molecular determinants of cardiac lymphatic dysfunction in a chronic pressure-overload model

Coraline Heron [1,8], Theo Lemarcis [1,8], Océane Laguerre [1], Bénjamin Bourgeois [2], Corentin Thuilliez [1,3], Chloé Valentin[1], Anais Dumesnil[1], Manon Valet[1], David Godefroy [4], Damien Schapman[2], Gaetan Riou[5], Sophie Candon [5], Céline Derambure[6], Alma Zernecke [7], Caroline Berard[3], Hélène Dauchel [2,3✉], Virginie Tardif[1] & Ebba Brakenhielm [1✉]

## Abstract

**Cardiac lymphatic alterations and insufficient lymphatic drainage have been found in cardiovascular diseases (CVDs). To unravel the mechanisms underlying lymphatic dysfunction, we applied single-cell (sc) analyses in murine heart failure (HF) models. Transaortic constriction (TAC) in C57BL/6J and BALB/c mice modeled chronic pressure -overload-induced cardiac hypertrophy and HF, respectively. Cardiac lymphatic (LEC) and blood vascular endothelial cells (BEC) were analyzed by scRNAseq, and targets validated by immunohistochemistry and human LEC cultures. While LEC profiles were comparable between strains in healthy mice, we found expansion of lymphatic capillaries and loss of valves post-TAC only in BALB/c. Differentially expressed gene (DEG) analysis revealed a reduction post-TAC only in BALB/c of lymphatic junctional components. Conversely, LEC expression of immune cell cross-talk mediators was mostly preserved post-TAC. Interestingly, around 35% of DEGs identified in cardiac LECs post-TAC were similarly altered in interleukin (IL)1β-stimulated human LECs. In conclusion, loss of lymphatic valves and dysregulated lymphatic barrier may underlie poor drainage capacity during pressure-overload-induced HF, despite potent lymphangiogenesis and preserved LEC immune attraction. Our work provides tractable targets to restore lymphatic health in CVDs.**

**Keywords** Hypertrophy; Inflammation; Valve; Precollectors; Claudin-5
**Subject Categories** Cardiovascular System; Chromatin, Transcription & Genomics; Vascular Biology & Angiogenesis

## Introduction

Recent research has demonstrated that cardiac lymphatics remodel in CVDs in mice and men (Brakenhielm and Alitalo, 2019; Oliver et al, 2020; Harris et al, 2023). This includes not only structural changes, impacting lymphatic density and morphology, but also functional changes (Henri et al, 2016; Bizou et al, 2021; Harris et al, 2021), together determining cardiac lymphatic drainage capacity. Given that lymphatics are essential for cardiac homeostasis (Miller, 2011), such development in CVDs of cardiac lymphatic dysfunction, and/or insufficient lymphangiogenesis, may aggravate cardiac inflammation, fibrosis, and adverse ventricular remodeling and contribute to the development of heart failure (HF). While suppression of lymphangiogenesis in CVD models has not always has been associated with aggravated cardiac dysfunction (Brakenhielm et al, 2024), promisingly, therapeutic lymphangiogenesis suffices to reduce cardiac dysfunction in models of myocardial infarction (MI) and chronic pressure-overload (Henri et al, 2016; Yang et al, 2014; Vieira et al, 2018; Trincot et al, 2019; Liu et al, 2020a; Houssari et al, 2020; Song et al, 2020; Glinton et al, 2022). We recently demonstrated that pressure-overload-induced left ventricular (LV) dilation, following transaortic constriction (TAC), triggers massive endogenous cardiac lymphangiogenesis in BALB/c mice (Heron et al, 2023a). Intriguingly, the expanded lymphatic network failed to resolve chronic myocardial inflammation and edema, and the mice progressed to HF. We argued that inflammatory mediators, including IL1β and IL6, produced in the heart notably by macrophages during pressure overload (Heron et al, 2023a; Amrute et al, 2024), may modulate lymphangiogenesis and/or lymphatic function. Previous work has identified cardiac macrophages as a key source of VEGF-C (Heron et al, 2023a; Glinton et al, 2022), and we recently demonstrated that IL1β promotes cardiac lymphangiogenesis post-TAC by stimulating macrophage production of VEGF-C (Heron et al, 2023b). However, it remains unknown what cellular and molecular changes in lymphatic endothelial cells (LECs) may contribute to poor lymphatic uptake and/or drainage during myocardial

[1]UnivRouen Normandy, INSERM UMR1096 (EnVI Laboratory), Rouen F-76000, France. [2]UnivRouen Normandy, INSERM US51 CNRS UAR 2026 (HeRacLeS Laboratory), Mont Saint-Aignan, France. [3]UnivRouen Normandy, LITIS UR4108, Mont Saint-Aignan, France. [4]UnivRouen Normandy, INSERM UMR1239 (NorDic Laboratory), Mont Saint-Aignan, France. [5]UnivRouen Normandy, INSERM UMR1234 (PANTHER Laboratory), Rouen F-76000, France. [6]UnivRouen Normandy, INSERM UMR1245 (CBG Laboratory), Rouen F-76000, France. [7]Institute of Experimental Biomedicine, University Hospital Würzburg, Würzburg 97080, Germany. [8]These authors contributed equally: Coraline Heron, Theo Lemarcis.✉E-mail: helene.dauchel@univ-rouen.fr; ebba.brakenhielm@inserm.fr

inflammation. To date, our knowledge of the molecular underpinnings of cardiac lymphatic (dys)function in CVDs is essentially limited to a few key players, including a cell junctional molecule (VE-Cadherin (Harris et al, 2021)) and select mediators of lymphatic-immune cell cross-talk (Lyve1 (Vieira et al, 2018), Ccl21 (Houssari et al, 2020; Song et al, 2020; Heron et al, 2023a, 2023b)). Indeed, the hyaladherin Lyve1 has been shown to be essential for cardiac immune cell exit through lymphatics post-MI, by interacting with hyaluronan-coated CD44-expressing immune cells to mediate leukocyte rolling and transmigration (Vieira et al, 2018). In contrast, pre-clinical investigations of lymphatics in other tissues, such as inflamed gut, skin, lymph nodes, and tumors, have revealed widespread transcriptional changes in LECs during acute or chronic inflammation. Examples include alterations of capillary LEC cell junctions (Baluk et al, 2007); loss of lymphatic valves (Czepielewski et al, 2021); increased LEC expression of cell adhesion molecules (*Icam, Vcam, Selp*) and chemokines (*Ccl21, Cxcl12, Cx3cl1*) (Aebischer et al, 2014); altered lymphatic production of bioactive lipids, e.g., sphingosine-1 phosphate (S1P) (Harlé et al, 2021; Kim et al, 2023); and finally increased LEC expression of immune-modulators, including major histocompatibility complex (MHC) for antigen presentation, coupled to immune-suppressive co-signaling molecules, such as *Pdl1* (Programmed Death Ligand-1) (Li et al, 2022). We hypothesized that cardiac lymphatic dysfunction in CVDs may be caused not only by insufficient lymphangiogenesis, but also by structural and molecular changes in cardiac LECs induced by cardiac inflammation. Here, we set out to investigate, using scRNAseq and 3D imaging, the cellular and molecular shifts that may accompany cardiac lymphangiogenesis in the setting of inflammation induced by chronic pressure overload in BALB/c mice, with the ultimate aim to uncover potential new targets, beyond lymphangiogenic growth factor therapy, to restore lymphatic drainage in CVDs. We further compared the profiles of cardiac LECs between BALB/c and C57BL6/J (C57) strains in health and disease. Of note, the TAC model in the C57 strain is associated with less myocardial inflammation and cardiac dysfunction, as compared to BALB/c (Heron et al, 2023a), although lymphatic transport dysfunction also has been noted in C57 mice following chronic pressure overload (Bizou et al, 2021; Song et al, 2020). Finally, we compared our datasets to published reports on lymphatic molecular profiles in other organs and disease settings to identify potential targets specific to cardiac lymphatics. In parallel, we investigated molecular changes occurring in cardiac blood vascular endothelial cells (BECs) in pressure overload, as a comparison to highlight target genes selectively altered in cardiac LECs.

## Results

### Distinct blood vascular and lymphatic profiles in healthy mouse hearts

To define the transcriptome of healthy murine cardiac vascular endothelial cells (ECs), we sorted by FACS cardiac CD45⁻/CD31⁺ cells from 20 sham-operated adult female BALB/c mice, further enriching the samples for Lyve1 and Podoplanin (Pdpn)-expressing LECs and for lymphatic-marker-negative BECs prior to scRNAseq

(Fig. 1A). After quality controls (removing genes expressed by <10 cells; and removing cells expressing <500 genes, and >10% mitochondrial-encoded genes), we recovered 1814 cardiac EC transcriptomes. These distributed (Fig. 1B,C) into three major clusters: *LEC* (402 transcriptomes), capillary/arterial BECs (*BEC*, 1196 transcriptomes), and venous BECs (*vBEC*, 216 transcriptomes), as determined using Seurat graph-based unsupervised clustering and visualization by Uniform Manifold Approximation and Projection (UMAP). All EC clusters expressed classical vascular markers (*Pecam1, Kdr*) but not immune cell or smooth muscle cell markers (*Ptprc, Cd68, Acta2*) (Fig. EV1A). On average, in each cell, we detected 2699 mRNA transcripts (reads), representing 1341 unique genes (Appendix Table S1). The cell cycle phases of cardiac ECs were majoritarian G1 or G2/M, with slightly more BECs than LECs or vBECs in S phase in healthy adult mouse hearts (Fig. EV1B,C).

Comparing mean gene expression levels between cardiac EC clusters to identify population markers, we found 22, 45, and 35 genes distinguishing, respectively, LECs, BECs, and vBECs (Fig. 1D–F; Dataset EV1). BECs expressed higher levels, compared to the other clusters, of key blood vascular EC-selective genes, including *Cxcl12, Kdr* (VEGFR2), *Nrp1* (Neuropilin-1), *Aqp1* (Aquaporin-1), and the transcription factors *Id1* and *Ets1*. Cardiac BECs were further distinguished by elevated expression of lipolysis-related genes (Fig. 1D,F). Indeed, regulators of fatty acid (FA) uptake and cytosolic lipid trafficking, including *Lpl* (Lipoprotein Lipase), *Cd36*, and *Fabp4* (FA binding protein-4), were preferentially expressed in cardiac BECs (Figs. 1F and EV2A), while other key lipolysis-related genes, e.g., the mitochondrial lipid transporter *Cpt1a* (Carnitine palmitoyltransferase), were expressed at comparable levels in all cardiac EC clusters, as were several key glycolysis regulators (e.g. *Notch1, Foxo1, Hif1α, Slc2a1* (Glut1), *Slc2a8* (Glut8), *Hk1* (Hexokinase-1)) (Fig. EV2A). Our findings are in line with previous reports, comparing cardiac and brain ECs, revealing low levels of glucose transporters, but high levels of FA uptake genes in the cardiac vasculature (Kalucka et al, 2020).

Cardiac vBECs were characterized by elevated expression of canonical venous markers, including *Dcn* (Decorin), *Vwf* (Von Willebrand Factor), and *Mgp* (Matrix Gla Protein), but also many other genes, such as *Npr3 (Natriuretic peptide receptor 3)* and *Cfh* (Complement factor H) (Fig. 1D–F; Dataset EV1). vBECs were further distinguished by elevated expression of *H2az1* (MHC class I), compared to BECs and LECs, although *H2-D1* and *H2-K1* were the majoritarian MHC genes in all cardiac EC populations (Fig. EV2B).

The cardiac LEC cluster was characterized by elevated expression of lymphatic markers, including *Ccl21a, Mmrn1* (Multimerin-1/Emilin4), *Flt4* (VEGFR3), and the transcription factors *Prox1, Nfat5/TonEBP*, and *Nfatc1* (Fig. 1D–F, Dataset EV1). It was also distinguished by elevated expression of many other genes, including *Sema3d* (Semaphorin 3d), *Cd9* (Tetraspanin family member), *Lbp* (Lipopolysaccharide-binding protein), *Fcgrt* (Fcγ immunoglobulin receptor and transporter), and *Fgl2* (Fibrinogen-Like 2). The latter, expressed by the majority of cardiac LECs while barely detectable in BEC and vBEC clusters (Fig. 1E), encodes a secreted protein with immune-modulatory functions (Yan et al, 2019). Of note, several of the identified non-canonical lymphatic genes expressed by cardiac LECs, including *Sema3d, Fgl2, Cd9*, and *Thy1* (CD90), have been

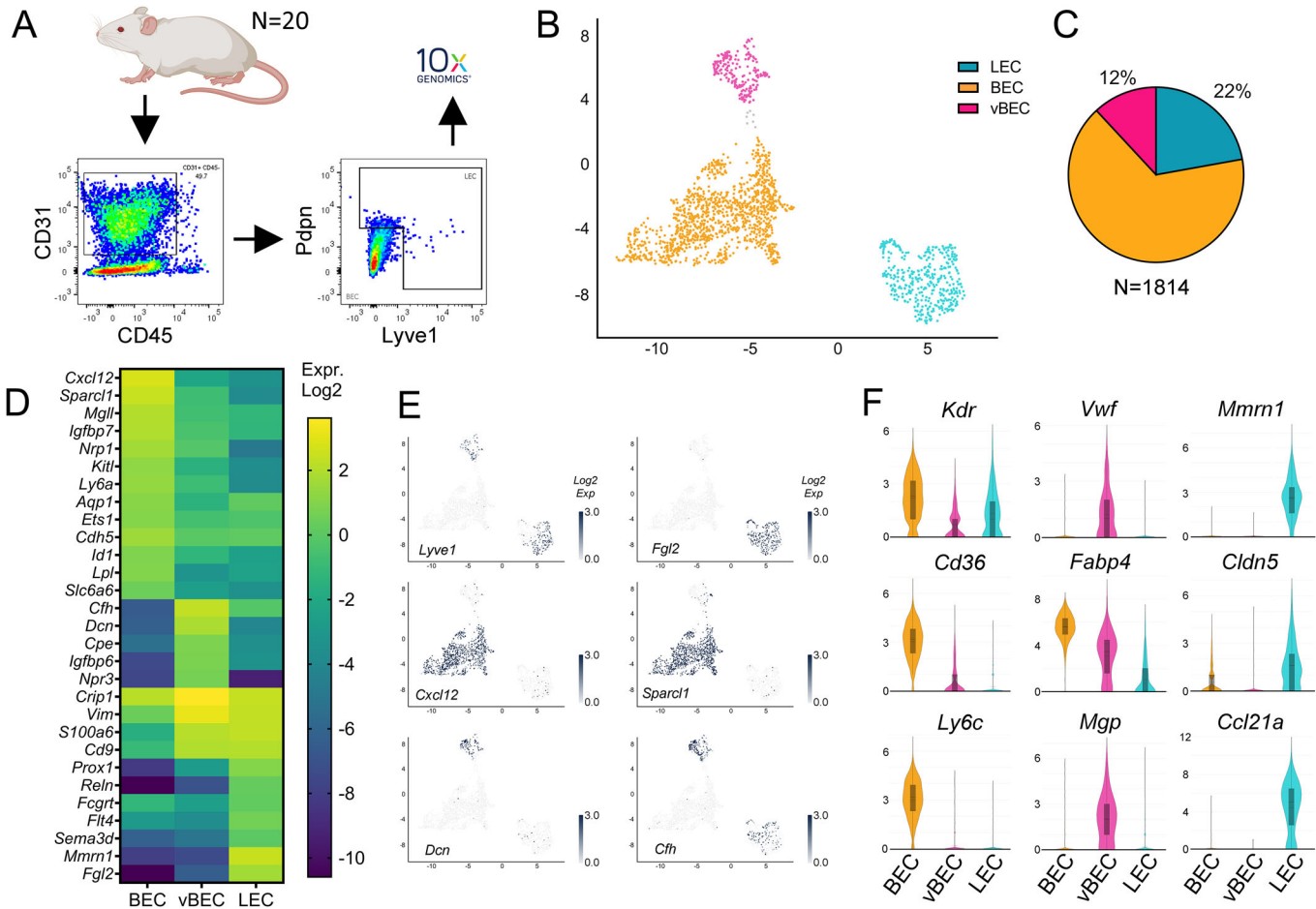

**Figure 1. Distinguishing markers of healthy cardiac endothelial cell populations.**

(A) Experimental overview: single cell preparations of cardiac cells from 20 healthy (sham-operated) mice were sorted by FACS to select CD31$^+$/CD45$^{neg}$ cardiac endothelial cells (ECs), among which we sorted (i) a rare subpopulation of ECs positive for lymphatic (LEC) markers, Lyve1 and Pdpn; and (ii) blood vascular ECs negative for these markers. The two sorted EC populations were then pooled before inclusion in the 10x Genomics pipeline. (B) UMAP clustering ($N = 1814$ transcriptomes) revealed three distinct cardiac vascular EC populations: BEC (orange), LEC (blue), and vBEC (pink). (C) These represented 66%, 22%, and 12% of all ECs, respectively. (D) Heatmap examples of expression levels of significantly enriched marker genes (Log2 normalized median values). (E) Examples of selectively enriched genes for each cluster (Log2 normalized read counts). (F) Violin plot examples of gene expression levels (Log2 normalized read counts), including canonical lymphatic (*Ccl21, Mmrn1, Lyve1, Prox1, Flt4*); blood capillary and arteriolar (*Cxcl12, Nrp1, Kdr, Cd36*); and venous (*Dcn, vWF, Mgp, Chf*) markers. Violin plots were generated in Loupe Browser, the median range is indicated by a box. For details, see Appendix table S2. Source data are available online for this figure.

previously detected in lymphatics in different organs in mice and humans (Kalucka et al, 2020; Jurisic et al, 2010, 2012).

In the heart, maintenance of a tight vascular barrier represents a key function of blood vessels necessary for cardiac health, while lymphatic capillaries, in physiology, are permeable to allow fluid, macromolecule, and immune cell entry. Indeed, lymphatics are endowed with discontinuous button-junctions in capillaries, while precollector lymphatics display tight "zipper-type" junctions to promote efficient transport to lymph nodes (Baluk et al, 2007). In both blood vessels and lymphatics, these vascular barriers are constituted by a combination of adherence (e.g., Cadherins, Catenins, Nectins) and tight junction (e.g., Claudins, Occludin, Esam, ZO) proteins (Fig. EV3A,B). We found in healthy hearts that these components displayed differential expression, with more *Cdh5* (VE-Cadherin) and *Jam2* (Junctional adhesion molecule B) in BECs, more *Cldn5* (Claudin-5) and *Nectin2* in LECs, and more

*Cdh13* (T-Cadherin) in vBECs (Figs. 1D,F and EV3A). In contrast, gene expression levels of other junctional components, and of a major regulator of vascular barrier integrity, the S1P receptor *S1pr1*, were comparable between cardiac EC clusters.

Reclustering analysis of the BEC population was performed to further investigate the expression of vascular barrier components in BEC subpopulations (see Dataset EV2 for subcluster marker genes). We found that among the identified eight BEC subpopulations, elevated *Cldn5* was present only in two subclusters: BEC9 (Fig. EV3C), enriched for arteriolar markers (e.g., *Id1*, Notch-target gene *Hey1*), and BEC6, enriched for angiogenic markers (e.g., *Aplnr2*, Apelin receptor 2). Similarly, preferential arteriolar localization of Cldn-5 has been demonstrated in mouse kidneys (Morita et al, 1999), whereas most blood vessels in the brain depend on Cldn-5 for maintenance of the blood–brain barrier (Nitta et al, 2003).

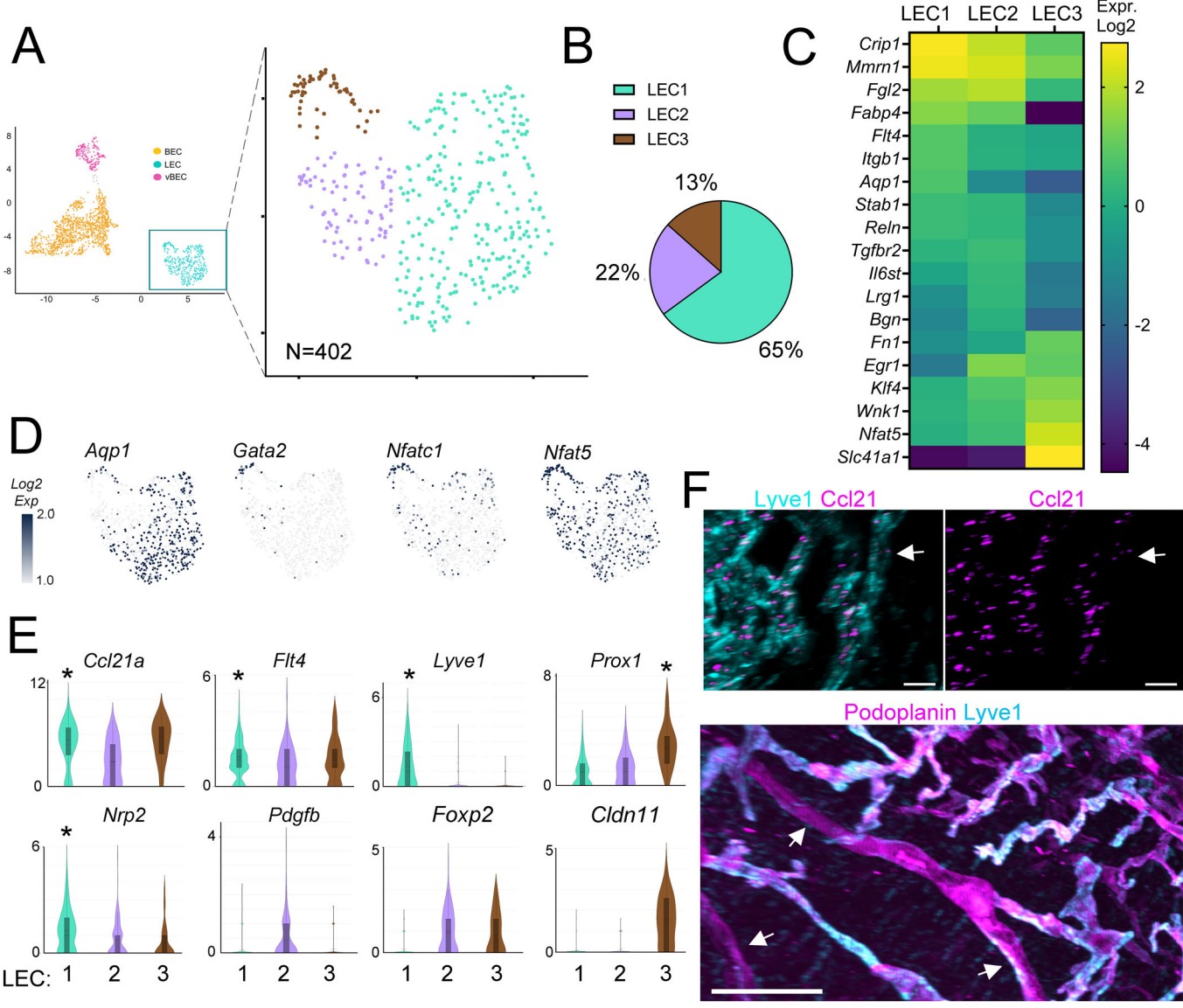

**Figure 2. Identifying cardiac LEC subpopulations in healthy mouse hearts.**

(A) Cardiac LECs (N = 402) obtained from healthy sham-operated BALB/c mice (n = 20) clustered into three subpopulations: LEC1 (cyan), LEC2 (purple), and LEC3 (brown). (B) These represented 65%, 22%, and 13% of all analyzed LECs, respectively. (C) Heatmap examples of significantly enriched marker genes for each cluster (Log2 normalized median values). (D) Examples of select marker genes (Log2 normalized read counts). (E) Violin plot examples of the expression levels of LEC subpopulation-enriched genes (Log2 normalized read counts), including capillary markers (*Ccl21, Mmrn1, Lyve1, Aqp1, Flt4, Nrp2*), precollector markers (*Pdgfb, Foxp2*), and lymphatic valve markers (*Gata2, Prox1, Nfat5, Cldn11*). (* denotes genes significantly enriched in a given cluster in (E). For exact P values, see Dataset EV3. Violin plots were generated in Loupe Browser, the median range is indicated by a box). (F) Cardiac lightsheet imaging: Top panel: examples of Ccl21-expressing lymphatic vessels (Ccl21, *magenta*; Lyve1, *cyan*, scale bar 40 μm), including a Lyve1/Ccl21 weakly-expressing precollector segment (indicated by white arrow). Bottom panel: examples of lymphatic Lyve1low precollector segments (Lyve1, *cyan*; Pdpn, *magenta*; *White arrow*: precollector segments, scale bar 300 μm). Source data are available online for this figure.

## Identification of cardiac lymphatic subpopulations in healthy mice

To investigate cardiac lymphatic subpopulations, we next performed reclustering analysis of LECs, which revealed three distinct clusters: LEC1 (65% of cells), LEC2 (22% of cells), and LEC3 (13% of cells) (Fig. 2A,B). The majoritarian LEC1 cluster was distinguished by elevated expression of canonical lymphatic capillary markers, including *Mmrn1, Lyve1, Nrp2, Aqp1*, and *Flt4* (Fig. 2C–E; Dataset EV3).

The lymphatic-selective chemokine *Ccl21a* was highly expressed by all cardiac LECs, with the highest levels found in the capillary LEC1 cluster (Fig. 2E), similar as reported for lymphatics in other organs. Whole-mount confocal analyses further demonstrated steep lymphatic gradients of Ccl21 confined to the subepicardial regions of the heart, with a large proportion of the signal detected inside capillary LECs (Fig. 2F; Movie EV1).

The LEC2 subpopulation more frequently expressed *Pdgfb* (32% of cells) and *Foxp2* (55% of cells), as compared to LEC1 (8% and

13% of cells, respectively). The LEC2 cluster, which we identified as precollector-type, was also distinguished by higher expression of *Bgn* (Biglycan), *Cfh*, *Egr1* (Early Growth Response 1), and *Selenop* (encoding a secreted antioxidant selenoprotein). We confirmed, using confocal and lightsheet imaging, that precollector segments displayed similar Pdpn expression, but lower levels of *Lyve1* and *Ccl21a*, as compared to capillary lymphatics (Fig. 2C–F; Dataset EV3).

The small LEC3 cluster was characterized by elevated expression of several valve-regulating transcription factors, including *Prox1, Nfat5,* and *Klf4* (Dataset EV3), and it also more frequently expressed other key transcription factors (*Nfatc1, Gata2, Foxo1, Foxc2, Nr2f2* (COUP-TF2)) (Figs. 2C–E and EV4).

Overall, the molecular signatures of these cardiac LEC clusters were very similar to profiles previously reported for lymphatic subpopulations in other tissues, e.g., mouse skin (Petkova et al, 2023). Indeed, studies comparing LECs from different organs have found few organ-specific differences in the transcriptomic signatures of healthy lymphatics (Kalucka et al, 2020), despite accumulating evidence of organ-selective differences in developmental LEC origins (Petrova and Koh, 2018).

## Altered molecular profiles in cardiac ECs during pressure-overload-induced HF

Next, to define the transcriptome of cardiac ECs during pressure overload, we repeated the sorting and sequencing procedures in 10 TAC-operated BALB/c mice at a stage of chronic HF, as described (Heron et al, 2023a). After quality control, as above, we obtained 1455 EC transcriptomes. Following expression matrix normalization and batch-correction to integrate this dataset with healthy BALB/c EC transcriptomes, we found a similar distribution post-TAC of ECs clustering into three main populations: LECs (449 transcriptomes), BECs (838 transcriptomes), and vBECs (168 transcriptomes), as determined by Seurat graph-based clustering and visualization using UMAP (Fig. 3A,B). On average, in each cell, we detected 4096 transcripts and 1761 unique genes per cell (Appendix Table S1). The cell cycle phases of cardiac ECs post-TAC were similar to those in healthy sham-operated hearts (Fig. EV1B,C). This finding of low EC proliferation is supported by our previous immunohistochemical data indicating rare Ki67+ cardiac ECs at 8 weeks post-TAC, as both angiogenic and lymphangiogenic responses triggered by the pressure overload have abated in the chronic HF stage (Heron et al, 2023a).

Next, comparing changes in gene expression induced by TAC in the three main cardiac EC clusters, we identified a total of 416 differentially expressed genes (DEGs), including 139 in LECs (Dataset EV4), 150 in BECs (Dataset EV5), and 127 genes altered in vBECs (Dataset EV6). Whereas the BEC cluster was well-balanced in the number of up- and downregulated genes, vBECs displayed slightly more upregulated genes (57% of DEGs), while more downregulated genes were found post-TAC in LECs (66% of DEGs) (Fig. 3C).

Previous studies have identified an important non-canonical role of cardiac lymphatics related to the production of Reelin (Liu et al, 2020a), a large secreted glycoprotein that binds to several cell receptors, such as ApoE-R2, VLDLR, integrins, and ephrins. It is released by lymphatics under conditions of ischemia and/or inflammation, and has been shown in mice to act as a

lymphangiocrine factor stimulating cardiomyocyte survival post-MI (Oliver et al, 2020; Liu et al, 2020a). We found that while *Reln* (Reelin) expression was not significantly increased, it was more frequently expressed (90% of LECs post-TAC, vs. 62% of LECs in healthy hearts).

In contrast, several immune-related genes were significantly altered in cardiac LECs post-TAC. For example, *Lyve1, Ccl21,* and *Thy1* were upregulated, while *H2-T23* (MHC-I) was downregulated (Fig. 3D–F; Dataset EV4). However, more LECs expressed *H2-D1* (94% of cells) and *H2-K1* (78% of cells) post-TAC, as compared to healthy cardiac LECs (75% and 46% of cells, respectively) (Fig. EV2B; Dataset EV4). In addition, *Aplp2* (Amyloid-like protein 2) was downregulated (Fig. 3E; Dataset EV4). The encoded protein stimulates endocytosis, notably of MHC-I molecules (Tuli et al, 2009), thus its reduction may influence MHC-I plasma membrane levels in lymphatics. These shifts indicate that LEC-mediated immune cell recruitment and antigen presentation may be altered post-TAC.

We found that more LECs expressed *Serinc1* (Serine incorporator-1) post-TAC (78% vs. 48% of LECs). This gene encodes an enzyme stimulating synthesis of serine-derived lipids, such as S1P. The sphingosine biosynthetic pathway genes (kinases *Sphk1, Sphk2,* and lysase *Sgpl1*) were however not altered, while cardiac LECs tended to increase *S1pr1* expression post-TAC (Fig. EV3A). In contrast, reduced *S1pr1* levels have been reported in inflamed dermal lymphatics in the setting of lymphedema (Kim et al, 2023).

Concerning vascular barrier components, we found that *Cldn5* was strikingly downregulated in the cardiac LEC cluster post-TAC (Fig. 3D,F), as was another tight junction component, *Tjp1* (Fig. EV3A). This indicates that the vascular barrier in capillary and/or precollector lymphatics may be altered following pressure overload. Of note, previous work in other organs has demonstrated that inflammation reduces Cldn-5 expression in lymphatics, contributing to increased lymphatic permeability (Kajiya et al, 2012).

We further found that the expression levels of several lymphatic transcription factors, including *Prox1, Nfat5, Nfatc1,* and *Klf4* (Kruppel-Like factor-4), were reduced (Fig. 3D–F; Dataset EV4). However, their frequency of expression, as well as that of other transcription factors, notably *Nr2f2,* was maintained or increased in cardiac LECs post-TAC (Fig. EV4B). These changes, which may relate to alterations in LEC subpopulations, indicate a shift in the transcription factor landscape in LECs expected to influence lymphatic activity and function following chronic pressure overload. Of note, key Prox1 target genes, including *Flt4* and *Cdkn1c,* were not reduced, while *Lyve1,* as mentioned above, was increased (Fig. 3F), indicating that Prox1 activity is still sufficient to drive expression of these genes.

In addition, cardiac LECs post-TAC displayed globally increased expression of genes encoding extracellular matrix components, including *Eln* (Elastin), *Fbln2* (Fibulin-2), and *Loxl2* (Lysyl oxidase Like-2, a collagen cross-linking enzyme), as well as a proteinase inhibitor (*Serping1,* C1 inhibitor). It remains to be determined whether the activated lymphatics post-TAC may contribute to local remodeling of the extracellular matrix during pressure overload.

Functional enrichment analyses, by Over Representation Analysis (ORA), were carried out to identify biological processes altered in cardiac LECs post-TAC. Top hits (Dataset EV7, all with *q* value < 3E$^{-19}$) included: "*RNA splicing*" (GO:0008380, *padj* 3E$^{-35}$),

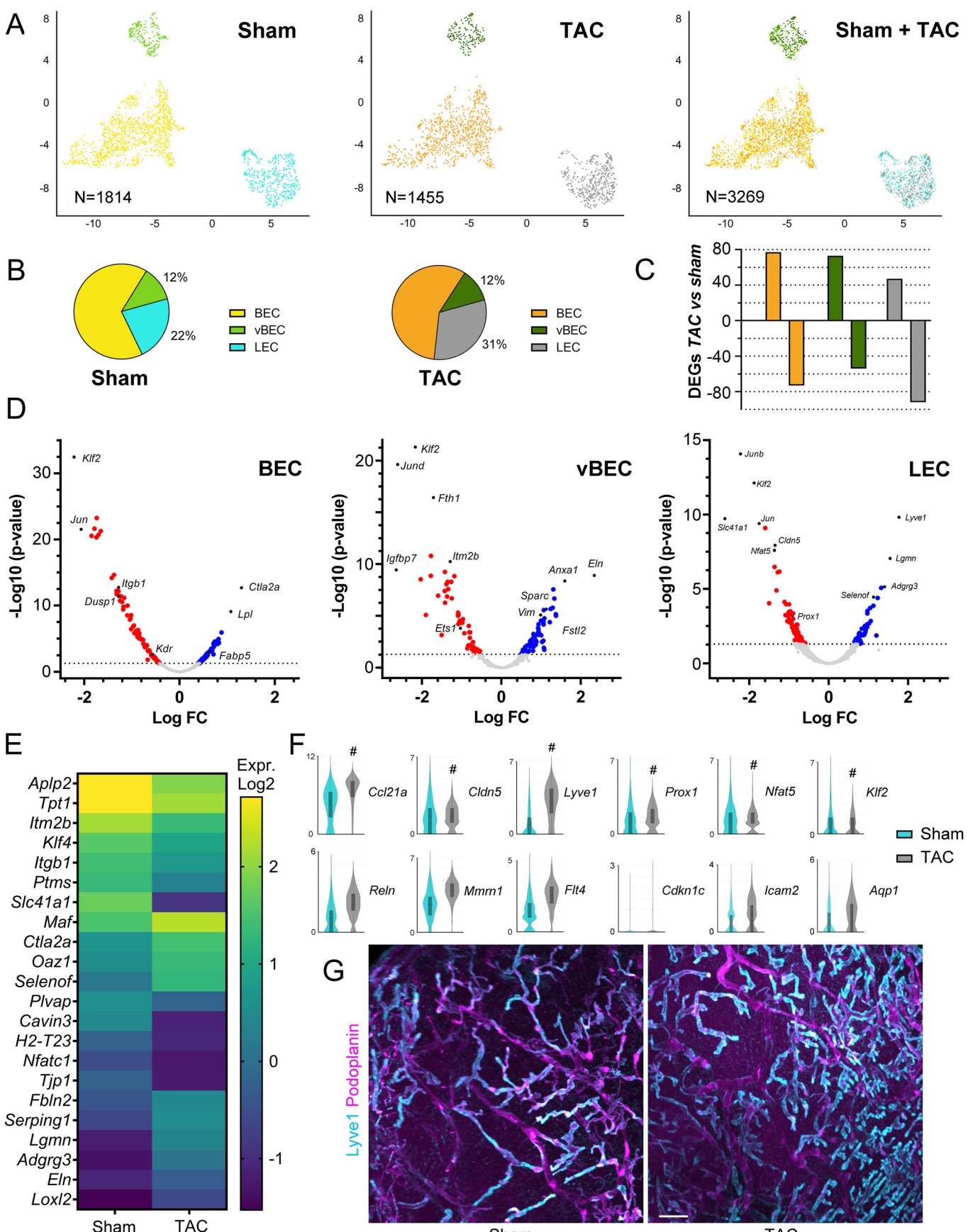

◀ **Figure 3. Molecular profile changes in the cardiac vasculature post-TAC in BALB/c.**

(A, B) Cardiac ECs ($N = 1455$) from post-TAC BALB/c mice ($n = 10$) clustered together with healthy cardiac ECs ($N = 1814$) into three main EC populations: BEC, LEC, and vBEC. (C) Numbers of DEGs (upregulated vs downregulated post-TAC as compared to corresponding healthy clusters) in BEC, LEC, and vBECs. (D) Volcano plot visualization of downregulated (red) and upregulated (blue) genes in BEC, vBEC, and LEC clusters post-TAC, as compared to healthy hearts. For the full list of DEGs, see Datasets EV4, EV5, and EV6. (E) Heatmap examples of significantly altered genes in cardiac LECs post-TAC (Log2 normalized median values). (F) Violin plot examples of select genes of interest in healthy (cyan) vs. post-TAC (gray) LECs (Log2 normalized read counts). Genes significantly altered post-TAC, identified using exact negative binomial test (sSeq method) in 10x Loupe Browser, indicated with # in (F). For exact $P$ values, see Dataset EV4. Violin plots were generated in Loupe Browser, the median range is indicated by a box. (G) Cardiac imaging of healthy cardiac lymphatics versus expanded lymphatics at 8 weeks post-TAC. Lyve1, cyan; Pdpn, magenta. Scale bar 100 μm. Source data are available online for this figure.

"*mRNA processing*" (GO:0050684, *padj* 3E$^{-19}$), "*Actin reorganization*" (GO:0032956, *padj* 1E$^{-21}$), and several processes related to cellular metabolism (e.g., GO:0015980, *padj* 7E$^{-19}$). Further, "*Ribosomal Biogenesis*" (GO:0042273, *padj* 1E$^{-3}$) was upregulated post-TAC. Of note, previous work has demonstrated that ribosomal biogenesis is increased in ageing BECs (Ximerakis et al, 2019), and our functional enrichment analyses indicated that stressed cardiac LECs during pressure overload also may activate this process.

Further, in line with the observed reduced expression of tight junction components in cardiac LECs post-TAC, KEGG pathway enrichment analysis identified several related downregulated pathways, including "*Focal adhesion*" (mmu04510, *padj* 2E$^{-2}$), and "*Fluid shear-stress*" (mmu05418, *padj* 2E$^{-2}$).

Finally, Reactome pathway enrichment analysis identified "*Cellular response to stress*" (R-MMU-2262752, *padj* 3E$^{-17}$) and "*Cross-presentation of antigens*" (R-MMU-1236978, *padj* 7E$^{-19}$) among key upregulated processes post-TAC.

Next, we similarly investigated the effects of chronic pressure overload on blood vascular ECs transcriptomes. In the BEC cluster, we found upregulation of genes involved in lipid uptake and FA cytosolic trafficking (*Lpl, Fabp4, Fabp5*), and in regulation of extracellular matrix (*Timp4, Sparc*) post-TAC (Dataset EV5; Fig. EV2A).

Further, while *Id3, S1pr1, Esam*, and *Icam2* expression levels were increased, *Ets1, Kdr, Tjp1*, and *H2-D1* were reduced post-TAC (Figs. 3D, EV2B, EV3A, and EV5A, For full list of DEGs, see Dataset EV5). These profile changes are indicative of generalized vascular EC activation, as previously described during pressure-overload-induced cardiac inflammation (Liu et al, 2020b). Further investigating BEC subpopulations, we found similar subclustering of cardiac BECs post-TAC as in healthy hearts (Fig. EV6A–D). Of note, all BEC clusters displayed alteration of their gene expression profiles post-TAC (Figs. EV3C and EV6E, for the full list of DEGs, see Dataset EV8). For example, the BEC1 cluster (enriched in capillary markers, including *Aqp1* (Aquaporin-1)) showed reduced expression levels post-TAC of *Ets1* and *Kdr*, and increased levels of *Timp4* and *Id3* (Fig. EV6F). Of note, *Lpl* was increased post-TAC in several distinct BEC subpopulations, while *Fabp4* was increased selectively in the *Notch1*-expressing BEC4 cluster (Dataset EV8). These changes suggest that BEC-mediated lipid uptake may be enhanced during the development of HF. Indeed, maintenance of cardiac FA uptake has been shown to limit myocardial metabolic remodeling in pressure overload (Umbarawan et al, 2018).

In the vBEC cluster, which represented around 15% of all blood vascular ECs in both healthy and failing hearts, we found many upregulated genes post-TAC (Fig. 3C,D, for the full list of DEGs, see Dataset EV6). These included *Vim* (Vimentin; type III

intermediate filament protein), *Sparc, Aqp1, Selenof* (an endoplasmic reticulum-resident selenoprotein), and *Fstl1* (Follistatin-related protein-1). In contrast, expression levels of *Nrp2, ApoE, H2-D1*, and *H2-K1* were reduced post-TAC (Figs. EV2B and EV5B). However, the frequency of expression of the two major MHC-I class molecules was increased (96% and 89% of vBECs expressed, respectively, *H2-D1* and *H2-K1* post-TAC), as compared to healthy vBECs (85% and 73% of cells expressing, respectively, *H2-D1* and *H2-K1*). Of note, *Aplp2* was also downregulated in BECs and vBECs post-TAC (Datasets EV5 and EV6), which may stimulate MHC-I-mediated vascular antigen presentation in pressure overload.

## Lymphatic subpopulation-selective molecular alterations and loss of lymphatic valves in failing hearts

Focusing next on LEC subpopulation changes, we first established that the three cardiac LEC clusters identified post-TAC displayed similar molecular markers as in healthy mice, with no apparent unique cell subpopulation emerging post-TAC (Fig. 4A; Dataset EV3).

We previously demonstrated that at 8 weeks post-TAC in BALB/c mice, the cardiac lymphatic network is massively expanded (Heron et al, 2023a), with selective overgrowth of Lyve1$^{high}$ lymphatic capillaries (Fig. 3G). In agreement, our scRNAseq analyses of cardiac LEC subpopulations revealed an increase post-TAC of the capillary LEC1 cluster, coupled to a reduction of precollector LEC2 cells, and especially of the lymphatic valve LEC3 cluster (Fig. 4A,B). Indeed, whereas LEC1 and LEC2 represented 82% and 14%, respectively, the LEC3 cluster only included 4% of all cardiac LECs post-TAC. This represents a striking fourfold decrease as compared to healthy sham-operated hearts (compare with Fig. 2B).

Our recent studies indicated that cardiac lymphatics may be dysfunctional post-TAC, as both cardiac immune cell infiltration and edema persist despite massive lymphatic expansion (Heron et al, 2023a). Loss of lymphatic valves has been shown to lead to lymphatic transport dysfunction in different organs in mice (Czepielewski et al, 2021). However, no previous study has investigated whether cardiac lymphatic valves are altered in CVDs. As our scRNAseq data indicated striking rarefaction of the LEC3 population, we hypothesized that the number of lymphatic valves may be reduced in the heart post-TAC. To image these elusive structures, we performed whole-mount imaging of Lyve1 together with Podocalyxin, which labels both lymphatic valves and blood capillaries (Fig. 4D; Movie EV2). In healthy hearts, we found that lymphatic precollectors (low Lyve1-intensity, straighter vessel segments), but surprisingly also capillaries (defined as Lyve1$^{high}$,

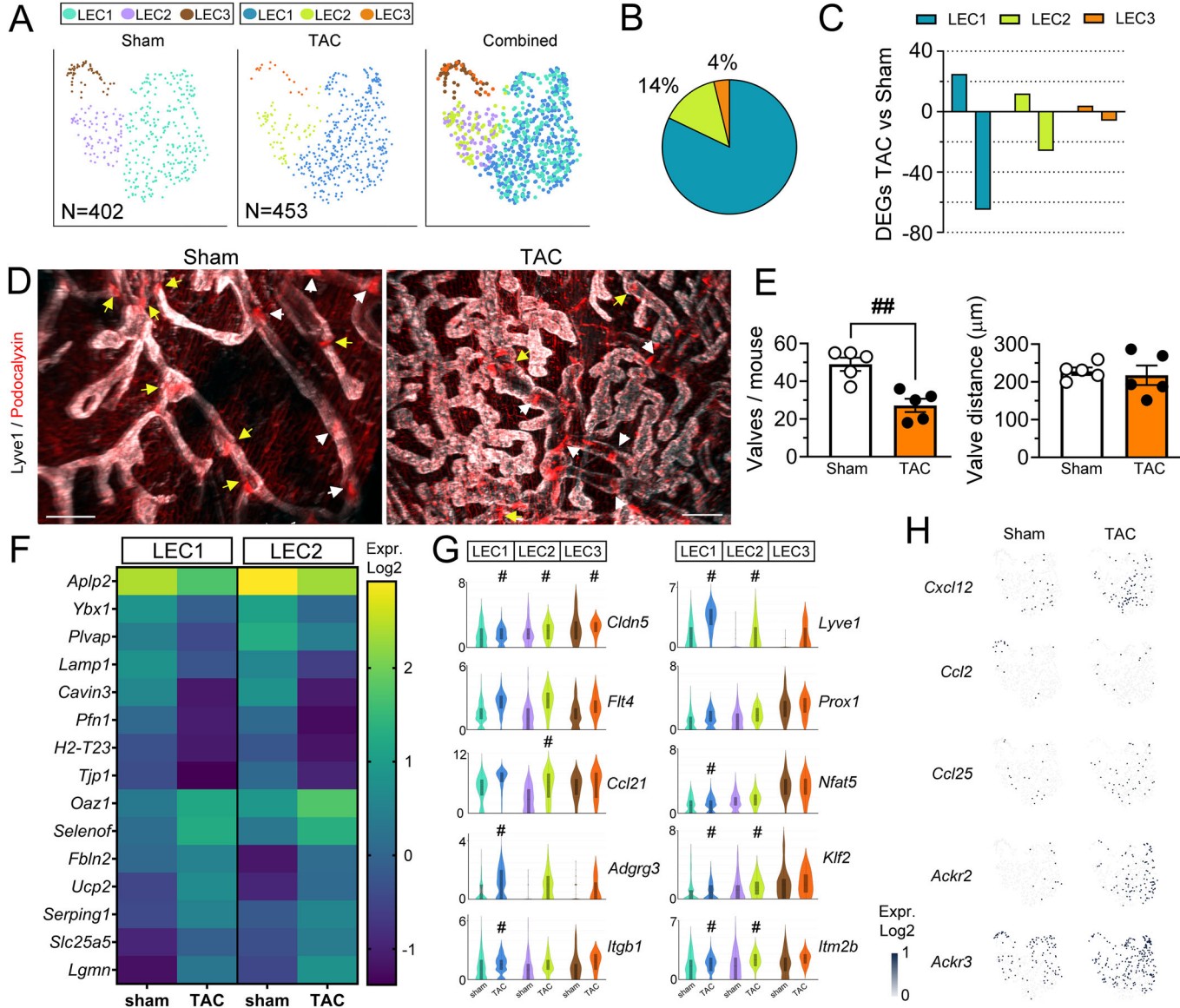

**Figure 4. Modification of cardiac LEC subpopulations post-TAC in BALB/c.**

(A) Cardiac LECs (N = 453), obtained post-TAC in *BALB/c* mice (n = 10), included three subpopulations: LEC1, LEC2, and LEC3. These cells clustered together with healthy cardiac LECs. (B) The three LEC clusters post-TAC represented, respectively, 82%, 14%, and 4% of all LECs. (C) Quantification of DEGs induced by TAC in the respective LEC clusters as compared to the corresponding healthy (sham) cardiac LEC clusters. (D) Examples of cardiac lymphatic valves in healthy versus post-TAC mice (Lyve1, gray; Podocalyxin, red; yellow arrows: lymphatic valves in capillaries, white arrowheads: valved precollectors. Scale bar, 200 μm. (E) Quantification of lymphatic capillary valves (n = 5 mice/group) in sham (white circles) and TAC (black circles), and assessment of average lymphatic intervalve distances. ##P < 0.0079 Mann–Whitney U test. Data shown as mean ± s.e.m. (F) Heatmap examples of genes significantly altered post-TAC in cardiac LEC subpopulations (Log2 normalized median values). (G) Violin plot visualization of genes of interest (Log2 normalized read counts). Violin plots were generated in Loupe Browser, the median range is indicated by a box. Significantly altered genes, identified using exact negative binomial test (sSeq method) in 10x Loupe Browser, indicated (#) for each cluster in (G). For exact *P* values, see Dataset EV9. (H) Examples of cardiac LEC expression of chemokines (*Cxcl12*, *Ccl2*, *Ccl25*), and *Ackr2* and *Ackr3* in healthy and TAC mice (Log2 normalized read counts). Source data are available online for this figure.

tortuous and looped segments), contained valves, with only the outermost tips of lymphatic capillaries completely devoid of valves. This organization is reminiscent of reports on dermal lymphatics (Pujol et al, 2017). In contrast, we found that the expanded lymphatic capillary network established by 8 weeks post-TAC was almost completely devoid of valves (Fig. 4D,E). To further investigate lymphatic valve properties in the heart, we analyzed

the distance between consecutive valves, finding an average intervalve distance of 200 μm in healthy mice (Fig. 4E). In the remaining valves in TAC-operated mice, this valve distance was unaltered, suggesting that insufficient valvulogenesis, not valve drop-out, may be involved. Taken together, our data indicate that the newly expanded capillary lymphatic network developing post-TAC may lack signals to drive lymphatic valvulogenesis, and this

structural deficit is very likely to restrict lymphatic drainage capacity despite elevated lymphatic density in the heart.

Next, we investigated whether lymphatic insufficiency post-TAC also may involve molecular profile changes in LEC capillaries (LEC1) impacting lymphatic cell-junctional organization or immune cross-talk. Analysis of LEC1, comparing TAC and sham, revealed 90 DEGs, including 65 downregulated genes (Fig. 4C; Dataset EV9). Among these were the tight junction components *Cldn5* and *Tjp1* (Fig. 4F,G). Reduced expression of tight junction components may influence lymphatic junctional organization (button/zipper junction). In lymphatic capillaries, looser cell–cell adhesion may facilitate fluid uptake during myocardial edema, while loss of junctional stability in precollectors would lead to fluid leakage and hence reduced transport capacity.

Among potential mechanical stress-induced changes post-TAC, we observed downregulation of several mechanosensitive genes, including *Klf2* (Kruppel-Like 2), *Dtx1* (deltex E3 ubiquitin ligase-1), and *Itgb1* (Integrin β1), but not the ion channel *Piezo1*, in cardiac LECs (Fig. 4G; Dataset EV9). In lymphatics, *Klf2* has been shown to regulate pressure-induced LEC sprouting, in part through upregulation of *Dtx1* (Choi et al, 2022), while *Itgb1* mediates shear-stress and interstitial fluid pressure-induced stimulation of lymphangiogenesis (Planas-Paz et al, 2012; Urner et al, 2019). Downregulation of these genes may reflect chronic adaptation to altered shear-stress profiles in expanded lymphatics. Moreover, the reduction of these mechanosensitive genes may serve to limit cardiac lymphangiogenesis despite the persistent myocardial edema and inflammation post-TAC.

Concerning lymphatic immune attraction, while *Lyve1* expression was increased, the levels of major chemokines (*Ccl21, Ccl2, Cxcl12, Ccl25*) were not significantly increased in the LEC1 cluster (Fig. 4G,H). Similarly, the expression of atypical chemokine receptors (*Ackr3, Ackr2*), which bind and scavenge chemokines, were not significantly altered (Fig. 4H). Of note, a recent scRNAseq study of inflamed dermal LECs reported a subpopulation of capillary lymphatics, denoted iLEC for immune-interacting cluster (Petkova et al, 2023), characterized by expression of *Ptx3* (Pentraxin-3), *Aqp1, Stab1* (Clever-1*), Cd200, Mrc1* (CD206*), Ackr2*, and *Plxnd1* (PlexinD1). In our dataset, we did not find such an "immune" capillary LEC subset emerging post-TAC. Indeed, *Ptx3* was only expressed by <6% of cardiac LEC1 cells in both healthy and post-TAC mice (Fig. EV7A). However, the frequency of *Ackr2* and *Mrc1* expression increased in the LEC1 cluster post-TAC (*Ackr2*, 25% vs <5% of cells; *Mrc1*, 41% vs. 12% of cells in healthy hearts). Nevertheless, cells expressing these proposed iLEC markers did not display convincing clusterisation within cardiac LECs in our study (Fig. EV7A,B).

In the LEC2 cluster, the 38 DEGs identified post-TAC (Dataset EV9) included 12 upregulated genes, notably *Lyve1* and *Ccl21*, and 26 downregulated genes, including *Cldn5, Klf2, Cavin3* (Caveolae-associated protein-3), *Itm2b (*Amyloid-binding Integral Membrane Protein-2b), and *Tgfrb3* (Transforming growth factor receptor β3).

DEG analyses in the LEC3 cluster lacked sensitivity due to the limited number of cells present post-TAC. Among the few identified upregulated genes was *Cavin2* (Dataset EV9). In contrast, there was no significant change in expression levels or frequency of expression of valve-inducing transcription factors (*Prox1, Nfat5, Foxc2, Nfatc1, Gata2*) in the remaining LEC3 cluster post-TAC (Figs. 4F,G and EV4A).

## Few modifications of cardiac ECs during pressure overload in C57 strain

To determine whether loss of lymphatic valves and dysregulation of cardiac EC expression profiles also occurred in a setting of less severe pressure-overload-induced cardiomyopathy, we repeated the cardiac EC sorting and scRNAseq in C57 adult female mice. We previously showed that in this strain of mice chronic pressure overload, by the same surgical procedure used in BALB/c, leads to pathological LV hypertrophy, accompanied by milder cardiac inflammation and less LV dysfunction at 8 weeks post-TAC (Heron et al, 2023a). We also reported limited expansion post-TAC in these mice of cardiac lymphatics, which seemed to retain a more physiological-like organization, as evaluated by whole-mount imaging. Sorting cardiac single cell suspension from 10 mice per group, we obtained, after quality control filtering (removing genes expressed by <10 cells; and removing cells expressing <300 genes, or >10% mitochondrial-encoded genes), 634 ECs from sham-operated mice and 892 ECs from TAC-operated C57 mice. On average, in each cell, we detected 3500–5400 transcripts and 1600–2200 unique genes per cell (Appendix Table S1). The transcriptomes clustered into LECs, BECs, and vBECs, similar as described for BALB/c, with 142 genes distinguishing these main cardiac EC populations (Fig. 5A,B,E).

Comparing LEC expression profiles between healthy BALB/c and C57 mice, we found that most lymphatic marker genes were expressed at comparable levels (Fig. EV8A–C). However, 132 genes were differentially expressed in cardiac LECs between strains (Dataset EV10). For example, BALB/c LECs expressed significantly higher levels of *Ccl21a, Aqp1, Nfat5,* and *Klf4*, while *Cldn5, Selenof, Mmrn1, Ramp2,* and *H2-K1* were more expressed in C57 LECs (Fig. EV8).

Significantly fewer cardiac LECs were recovered from C57 mice, either sham (N = 150 single-cell transcriptomes) or TAC (N = 81 single-cell transcriptomes), as compared to BALB/c. Nevertheless, reclustering of the global LEC cluster revealed three small, but distinct, LEC subpopulations (Fig. 5C,D): LEC1 (48% of all LECs), LEC2 (40% of all LECs), and LEC3 (12% of all LECs) in healthy C57 hearts. The identified markers for these subclusters mirrored those found in BALB/c (Fig. 5E,G). For example, the C57 LEC3 cluster expressed lymphatic valve marker genes (e.g., *Cldn11, Nfat5, Gata2*).

Next, comparing expression levels in all main EC clusters between sham and TAC C57 groups, we found that chronic pressure overload induced only very minor shifts in cardiac EC molecular profiles, with a total of 34 DEGs identified in cardiac ECs post-TAC (Dataset EV11). This included upregulation of *Cldn5* in BECs, and *Klf4* in both BEC and vBEC clusters (Fig. 5F).

Within the global cardiac LEC cluster, we only detected 3 DEGs post-TAC in C57 mice, and none of them involved lymphatic chemokines, immune adhesion molecules, or barrier components (Fig. 5G; Dataset EV11). Indeed, cardiac LECs post-TAC in C57 mice had a very similar molecular profile as healthy LECs, in strong support of a more physiological-like state of cardiac lymphatics during pressure overload in C57, as compared to in failing BALB/c hearts (Fig. EV8D).

Enrichment analysis (ORA) of LEC transcriptomes in C57 mice revealed upregulation post-TAC of processes including "*positive regulation of cell activation*" (GO:0050867, *padj 9E$^{-3}$*), "*positive*

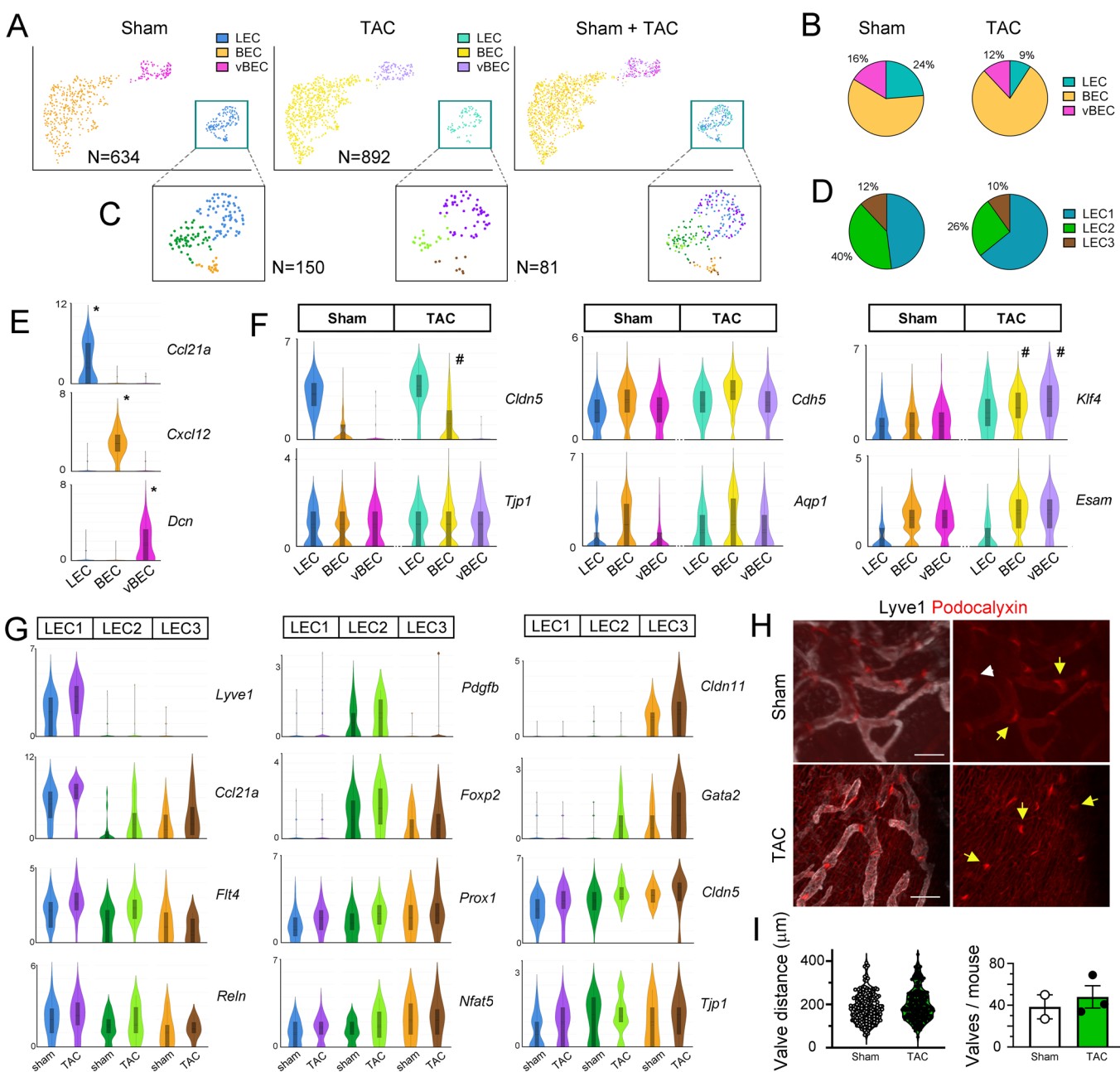

**Figure 5. Limited impact of chronic pressure overload on cardiac ECs in C57 mice.**

(A, B) Unsupervised clustering of cardiac ECs, from healthy or post-TAC C57 hearts ($n = 10$ mice per group, $N = 634$ and $N = 892$ transcriptomes, respectively) revealed three main EC populations: BEC, LEC, and vBEC. (C, D) Cardiac LECs ($N = 150$ healthy; $N = 81$ post-TAC) clustered into three main lymphatic subpopulations: LEC1, LEC2, and LEC3. (E) Visualization of gene expression levels (Log2 normalized read count) of cardiac EC marker genes. Violin plots were generated in Loupe Browser, the median range is indicated by a box. (F) Examples of other differentially regulated genes in TAC vs sham C57 mice. Marker genes significantly enriched in a given cluster indicated by * in (E), and significantly altered genes indicated with # for each cluster in (F). For the full list of DEGs and exact $P$ values, see Dataset EV11. (G) Visualization of select LEC cluster markers and other relevant genes in LEC subpopulations in sham and TAC C57 hearts (Log2 normalized read count, $n = 10$ mice/group). (H) Cardiac whole-mount visualization of lymphatic valves in healthy (top) versus post-TAC (bottom) C57 mice (Lyve1, gray; Podocalyxin, red; yellow arrows: lymphatic valves in capillaries, white arrowheads: valved precollectors). Scale bar, 100 μm. (I) Quantification of lymphatic valves per mouse in sham ($n = 2$, white circles) and TAC ($n = 3$, black circles) groups, and assessment of intervalve distances in cardiac lymphatics. Violin plots indicate upper and lower quartiles, and median in dashed lines. Source data are available online for this figure.

*regulation of leukocyte activation*" (GO:0002696, *padj 3E⁻²*), and "*positive regulation of B cell proliferation*" (GO:0030890, *padj 3E⁻²*).

Concerning lymphatic valves, we found that the frequency of valvular LECs were comparable between healthy C57 and BALB/c hearts, and strikingly we only found a minor reduction in the frequency of LEC3 cells post-TAC in C57 (Fig. 5D). Whole-mount analyses further confirmed that lymphatic valves were essentially retained after pressure overload in C57, with additionally no change in cardiac lymphatic intervalve distances (Fig. 5H,I).

### Validation of altered molecular profiles post-TAC

Next, we set out to verify, at the protein level, some of the identified cardiac lymphatic DEGs in BALB/c during pressure overload. We focused our analyses on lymphatic: (1) Lyve1 and Ccl21 expression levels; and (2) cell junctional components (Cldn5).

Whole-mount confocal imaging and flow cytometry both confirmed a striking increase in cardiac lymphatic Lyve1 levels post-TAC (Fig. 6A,B). Similarly, *Ccl21* upregulation post-TAC was confirmed by quantitative analyses of peri-lymphatic Ccl21 deposition in cardiac tissue sections, as recently reported (Heron et al, 2023a, 2023b). Thus, active Ccl21-mediated lymphatic recruitment of CCR7⁺ immune cells, and/or Lyve1-mediated transcytosis of hyaluronic acid-decorated immune cells (Vieira et al, 2018), is likely to be increased post-TAC in BALB/c. Conversely, and in agreement with potent downregulation of *Cldn5* observed in LECs during chronic pressure overload, our immuno-histochemical analyses revealed preferential expression of Claudin-5 in lymphatics in healthy hearts, but striking reduction of lymphatic Claudin-5 levels in BALB/c at 8 weeks post-TAC (Fig. 6C).

Inflammation has been reported to induce lymphatic dysfunction in many different organs (Scallan et al, 2016). We thus speculated that part of the dysregulated molecular profile observed in lymphatics post-TAC in BALB/c could be linked to elevated cardiac pro-inflammatory cytokine levels in failing hearts, notably IL1β (Heron et al, 2023a). To address this possibility, we stimulated human LEC primary cultures in vitro with IL1β for 24 h followed by bulk RNAseq to establish inflammation-induced DEGs (Dataset EV12). Among the 135 DEGs identified in the LEC cluster post-TAC (Dataset EV4), we found 50 genes that displayed similar changes in IL1β-treated human LECs (Figs. 6D and EV9). This suggests that a considerable part of the transcriptomic changes observed in lymphatics during pressure overload in Balb/c may be induced by cardiac pro-inflammatory cytokines, including IL1β. Specific examples of shared DEGs include upregulation of *Pdlim7* (PDZ And LIM Domain 7), *Selenof*, *Loxl2*; and downregulation of *Aplp2*, and of several lymphatic transcription factors (*Prox1*, *Nfat5*, *Nfatc1*), and of *Plvap* (Plasmalemma vesicle-associated protein). Of note, while *Tjp1* was reduced by IL1β in human LECs, *Cldn5* expression was unaltered.

In human LECs, we also found that IL1β upregulated *CD274* (PDL1) and several MHC genes. This suggests that LEC-mediated antigen presentation during cardiac inflammation may drive immune tolerance rather than T-cell activation, similar as reported in tumor models (Li et al, 2022). However, *Cd274* expression was not significantly increased in cardiac LECs post-TAC. Indeed, both our scRNAseq and immunohistochemical analyses showed low expression levels of Pdl1 in lymphatics post-TAC (Fig. EV7C,D).

## Discussion

### Distinguishing traits of cardiac lymphatic subpopulations in health and disease

To the best of our knowledge, this study represents the first targeted molecular analysis of cardiac LEC subpopulations. As lymphatics represent a rare cell population in the heart, being restricted in rodents to the subepicardial layers and the septum, an approach of LEC enrichment through FACS sorting was required. Of note, we only obtained, by FACS, around 200–300 cardiac LECs per mouse in BALB/c, and even fewer cells in C57 mice, and our scRNAseq thus included cells derived from 10 to 20 mice per group to yield sufficient numbers to allow lymphatic subpopulation analyses. Previously published expression analyses in cardiac LECs in health or disease include: (1) targeted qPCR in C57 mice (Bizou et al, 2021); (2) bulk RNAseq in C57 mice (Song et al, 2020); and (3) scRNAseq of sorted PECAM1⁺ vascular ECs in human fetal hearts (McCracken et al, 2022), and in adult C57 mice (Kalucka et al, 2020), including a single small LEC cluster each. Importantly, our study of LECs from healthy BALB/c and C57 hearts showed overall similar molecular profiles as these previous reports. However, our work goes well beyond these datasets by: (1) providing insight into what differentiates lymphatic capillaries from precollector and valve LECs in the heart; and (2) allowing cellular dissection of which compartment(s) of the lymphatic tree is affected during myocardial inflammation triggered by pressure overload in a strain-dependent manner.

Our molecular studies revealed that both gene and protein levels of Lyve1 were higher in cardiac lymphatic capillaries (LEC1) as compared to precollectors (LEC2), in line with findings from other tissues (Petkova et al, 2023). Further, different from our expectations, our study did not allow identification of new molecular markers specific for precollector lymphatics. Indeed, although the cardiac LEC2 cluster more frequently expressed *Pdgfb* and *Foxp2*, their expression levels were not homogenous, and would not allow unequivocal identification of precollectors in histological sections. Moreover, although several valve LEC-specific markers were readily identifiable, corresponding to markers described previously for lymphatic valves in other tissues (Petkova et al, 2023), the scarcity of valve LECs, notably in CVD settings, precludes analyses of these cells in tissue sections. Thus, it is still necessary to perform 3D imaging to differentiate, by immunohistochemistry, between the different segments composing the lymphatic vasculature of the heart.

### Molecular impact of inflammation on cardiac lymphatic functions

In the heart, where lymphatic drainage essentially depends on the force of cardiac contractions (Brakenhielm et al, 2020), the negative inotropic effects of IL1β (Li et al, 2021) may contribute, indirectly, to insufficient cardiac lymphatic transport in CVDs. However, our scRNAseq analyses, coupled to our in vitro study in human LECs, revealed that the pro-inflammatory microenvironment in CVDs also may directly impact lymphatic function by altering LEC expression of genes regulating: 1) cell junctional organization/stability impacting uptake and/or transport of fluids and macro-molecules (*Cldn5*, *Tjp1*, *Plvap*); 2) immune cell attraction and

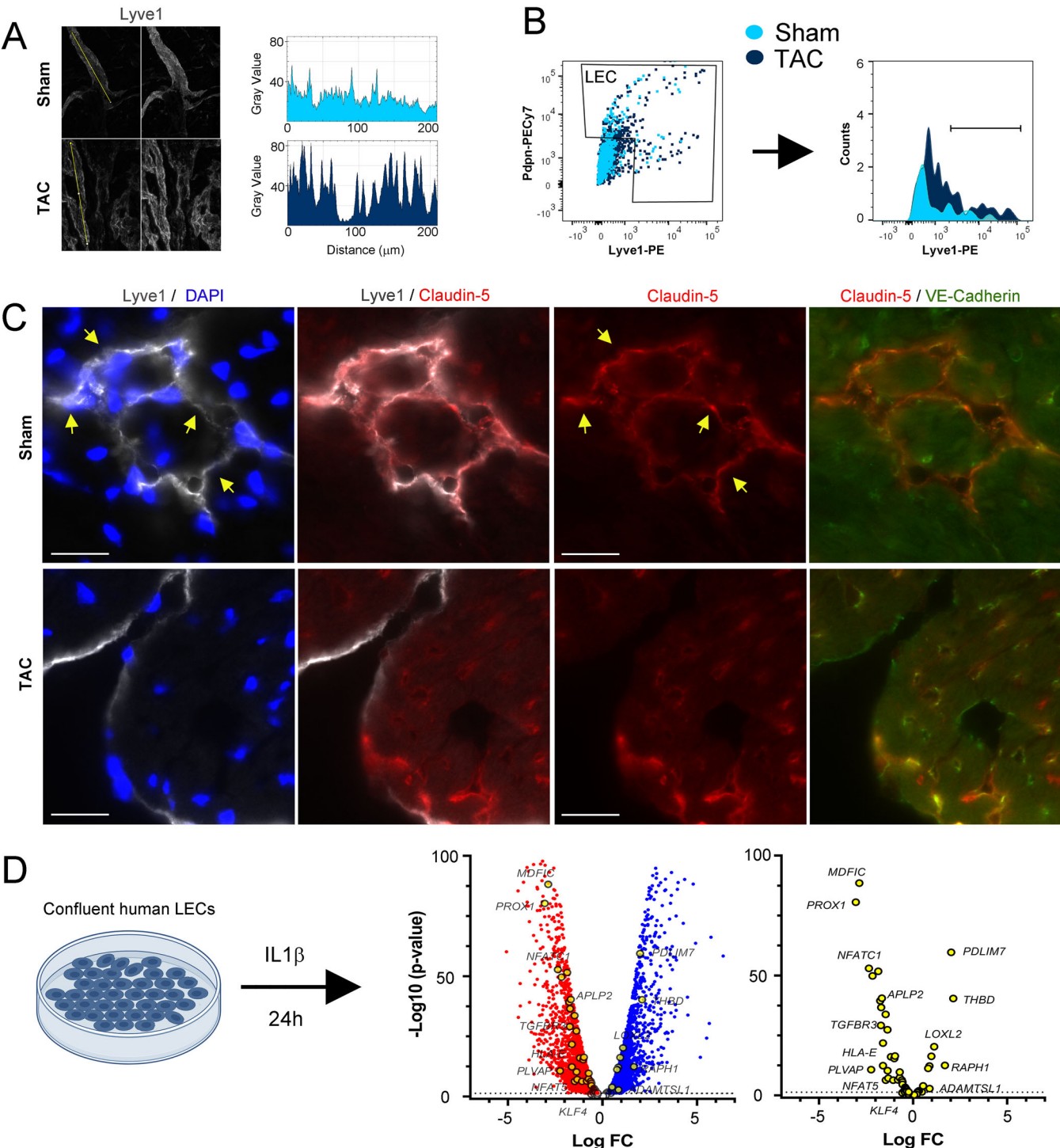

**Figure 6.  Lymphatic profile changes induced by cardiac inflammation.**

(A) Confocal imaging used for quantification of Lyve1 intensity in cardiac wholemounts in healthy and post-TAC BALB/c. Yellow line: lymphatic segment analyzed for signal intensity using ImageJ. (B) Flow cytometric assessment of Lyve1 expression in isolated cardiac LECs from healthy and TAC BALB/c. (C) Examples of Claudin-5 (red) expression in healthy and post-TAC cardiac lymphatics (Lyve1, gray, DAPI, blue) and blood vessels (VE-Cadherin, green). Scale bar 20 μm. Arrows: Claudin-5-expressing lymphatics. (D) Volcano plot visualization of DEGs, either downregulated (red) or upregulated (blue) genes, following IL1β stimulation of confluent human LECs in vitro, as calculated by DESeq2. Genes similarly altered by IL1β in human LECs, as observed in BALB/c cardiac LECs post-TAC, are highlighted (yellow circles). For details, see Dataset EV12 and Fig. EV9. Source data are available online for this figure.

adhesion (*Ccl21a, Lyve1, Thy1, Ackr3*); and 3) immune modulation (MHC molecules, *Fgl2, Aplp2, Cd274*).

### Reduced lymphatic junctional stability during pressure overload?

We observed a reduction post-TAC of *Cldn5* and *Tjp1* in cardiac LECs only in BALB/c mice. The reduction of Claudin-5 was confirmed in cardiac sections. However, a recent study reported that LEC *Cldn5* deletion did not cause complete breakdown, but only remodeling, of lymphatic junctions (Frye et al, 2020). In agreement, we previously reported no striking differences in the frequency of lymphatic capillary button-junctions post-TAC (Heron et al, 2023a), while they were clearly reduced, and replaced by zipper-junctions, post-MI (Houssari et al, 2020). Thus, the functional impact of low Cldn5 levels in cardiac LECs post-TAC remains to be determined. Of note, while IL1β altered the expression of *TJP1* and several claudins in human LECs, *CLDN5* was unaltered. This indicates that additional inflammatory mediators may contribute to loss of Cldn5 in cardiac lymphatics during HF. Interestingly, it has been proposed that angiopoietin-1 may restore lymphatic Cldn5 expression (Kajiya et al, 2012). We further found downregulation of *Plvap* in cardiac lymphatics (global LEC and LEC1 clusters) post-TAC and in IL1β-treated human LECs. This gene is a marker of fenestrated ECs, and has been proposed to restrict immune cell passage through lymphatics (Rantakari et al, 2015), hence downregulation of *Plvap* post-TAC may result in increased cardiac lymphatic permeability.

Our scRNAseq data can be compared with cardiac lymphatic transcriptomes, obtained by bulk RNAseq, in another model of chronic pressure-overload induced by Angiotensin-2 (AngII) (Song et al, 2020). This model, performed in male C57 mice, resulted in chronic hypertension, moderate cardiac hypertrophy and dysfunction, and rarefaction of cardiac lymphatics, which all were improved by VEGF-C$_{C156S}$ therapy. In contrast, we and others have previously reported maintenance, but not rarefaction, of cardiac lymphatics post-TAC in C57 mice, with accelerated disease after anti-VEGFR3 treatment-induced inhibition of lymphangiogenesis (Bizou et al, 2021; Heron et al, 2023a). Different from our findings in BALB/c post-TAC, but in line with our results in C57, Song et al (Song et al, 2020) did not report reduced expression in cardiac LECs of genes relevant for lymphatic barrier, including *Cldn5* and *Tjp1* (Appendix Fig. S1) (Data ref: Song et al, 2020).

### Enhanced cardiac lymphatic immune cell attraction and adhesion in pressure overload?

Our study revealed striking upregulation of both gene and protein levels of the main lymphatic chemokine, Ccl21, post-TAC only in BALB/c. In addition, *Lyve1* and *Thy1* were upregulated. In contrast, these genes, essential for immune cell attraction and adhesion, remained unaltered in cardiac lymphatics post-TAC in C57 mice. Similarly, previous bulk transcriptomic studies of cardiac LEC in C57 reported unaltered or reduced levels of *Ccl21* and *Lyve1* following pressure overload (Bizou et al, 2021; Song et al, 2020). Thus, not only does the BALB/c strain display a more potent lymphangiogenic response post-TAC as compared to C57 mice, but the immune recruitment capacity of their expanded cardiac LECs may also be increased. However, we were surprised by the limited number of chemokines expressed by cardiac LECs in our study. Indeed, no other chemokine, besides *Ccl21a*, was differentially expressed in cardiac lymphatics post-TAC. This may reflect poor

sensitivity of scRNAseq approaches for weakly expressed genes. In contrast, bulk RNAseq of cardiac LECs (Song et al, 2020) as well as of lymph node LECs (Berendam et al, 2019), have revealed a significantly wider range of lymphatic chemokines in both health and disease. In agreement, using bulk RNAseq, we found potent upregulation of many chemokines (e.g., *CCXL1, CXCL8, CCL2, CXCL6*) in human LECs following IL1β stimulation. Previous studies using bulk approaches have reported upregulation of *Cxcl12* in cardiac LECs in AngII-induced, but not in TAC-induced, pressure overload in C57 mice (Bizou et al, 2021; Song et al, 2020). In parallel, recent scRNAseq of cardiac-infiltrating immune cells post-MI and post-TAC in C57 mice have demonstrated that while *Cxcr4* (the receptor for Cxcl12), is mainly expressed by myeloid cells, including *Trem2*-expressing macrophages (Rizzo et al, 2023), *Ccr7* (the receptor for Ccl21), is preferentially expressed by cardiac B and T cells (Martini et al, 2019). We conclude that in our pressure-overload model in BALB/c, cardiac lymphatics are well-equipped to drain cardiac-infiltrating immune cells, including Ccr7-expressing B and T cells, and hyaluronan-coated CD44-expressing myeloid cells interacting with Lyve1. In agreement, we previously reported low cardiac B cell levels post-TAC in BALB/c, while intriguingly CD4[+] T-cell levels remained significantly elevated as compared to healthy mice (Heron et al, 2023a).

### Immune modulation by cardiac LECs during cardiac inflammation?

Cardiac LEC activation by the pro-inflammatory microenvironment characteristic of chronic pressure overload, notably in Balb/c, including elevated IL1β, could potentially influence lymphatic immune modulation beyond immune cell attraction and adhesion. As mentioned above, *Ptx3* has been proposed as a marker of an iLEC lymphatic subpopulation (Petkova et al, 2023). Moreover, *Ptx3* was found to be expressed by LECs in human fetal hearts (McCracken et al, 2022). In contrast, we did not find a cardiac LEC subpopulation homogenously expressing *Ptx3*. Indeed, this gene was expressed at low levels and frequency in murine cardiac LECs. Further, in our study, LEC1 expression of additional markers proposed for iLECs (*Mrc1, Plxnd1, Aqp1, Ackr2, Cd200*) were also unaltered post-TAC, although the frequency of expression of these markers increased. Nevertheless, we could not identify an iLEC subpopulation within cardiac LECs. We speculate that this may in part reflect differences in LEC proliferative status, with active expansion of dermal LECs occurring in the model of Petkova et al (Petkova et al, 2023), as compared to the more quiescent state of cardiac lymphatics at 8 weeks post-TAC in our study. It is also possible that differences in the levels of pro-inflammatory immune mediators could be involved. Indeed, our in vitro study demonstrated that acute IL1β stimulation sufficed to increase the expression of several iLEC markers (*PTX3, PLXND1, CD274, ACKR3, CD200*) in human LECs.

We noted in our study that cardiac LECs post-TAC in BALB/c mice displayed altered expression of genes relevant for antigen presentation, including more frequent expression of several MHC-I molecules and reduced *Aplp2* expression levels. Further studies are needed to assess the potential immune-modulatory, pro-tolerogenic role of cardiac LECs in different CVD settings.

## Conclusion and perspectives

The molecular alterations found in our study in cardiac LECs during the development of HF include both seemingly adaptive

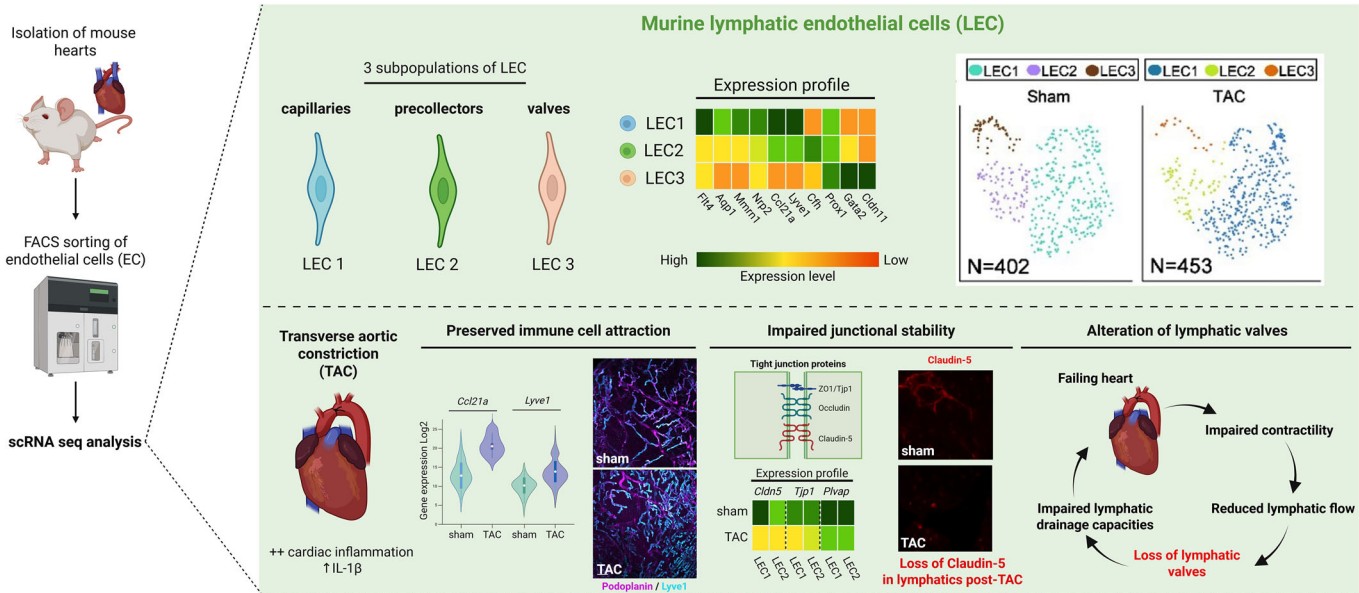

**Figure 7. Overview of changes to cardiac lymphatics in Heart Failure.**

Our study uncovered distinct molecular profiles of cardiac lymphatic subpopulations, including capillary, precollector, and valve LECs. Chronic pressure overload, induced by TAC, led to structural, cellular, and molecular changes in cardiac lymphatics. This included upregulation of genes involved in immune cell attraction, downregulation of genes involved in junctional stability, and rarefaction of valvular LECs in expanded lymphatic capillaries. We speculate that cardiac contractile dysfunction during the development of HF leads to reduced lymphatic transport, causing reduced shear-stress signals necessary for the generation of new lymphatic valves. Insufficient lymphatic valve formation, together with inflammation-induced alterations of LEC molecular profiles, results in further lymphatic dysfunction contributing to poor resolution of edema and inflammation, promoting progression of HF.

changes, set to improve inflammatory resolution (upregulated chemokines, immune adhesion molecules, and MHC molecules), but also potential maladaptive changes that may contribute to lymphatic transport dysfunction (loss of lymphatic valves, reduction of *Cldn5, Tjp1*). Our data indicate that many of these changes could be caused by inflammatory mediators, including IL1β. Previous work investigating lymphatics in other organs during inflammation has similarly reported both beneficial and deleterious changes induced by immune mediators (Baluk et al, 2007; Aebischer et al, 2014; Scallan et al, 2016; Harlé et al, 2021; Czepielewski et al, 2021; Li et al, 2022; Kim et al, 2023). It remains to be determined whether anti-inflammatory treatments may improve lymphatic function in CVDs. Of note, some anti-inflammatory treatments are currently investigated in clinical trials in patients with lymphedema (Rockson et al, 2018).

Our study, together with previous reports (Bizou et al, 2021; Song et al, 2020), indicates that cardiac lymphatics are more activated, on a cellular level, following pressure overload in BALB/c, as compared to in C57 mice. This is in line with more potent cardiac lymphangiogenesis in the former (Heron et al, 2023a). Indeed, it appears that the less expanded lymphatic network in hypertrophic C57 hearts, irrespective of the pressure-overload model (AngII (Song et al, 2020) or TAC (Bizou et al, 2021)), remains more physiological-like, as compared to the structurally and molecularly altered expanded lymphatics found in failing hearts in BALB/c. Of note, cardiac levels of several pro-inflammatory cytokines are higher post-TAC in BALB/c, as compared to in C57 (Heron et al, 2023a). This difference may contribute not only to more potent lymphatic expansion in the

former, given the emerging pro-lymphangiogenic impact of cardiac IL1β (Heron et al, 2023b), but also to dysregulation of LEC molecular profiles during chronic pressure-overload-induced HF (Fig. 7). Indeed, our in vitro data in human LECs provided compelling evidence for a key role of IL1β in mediating cardiac inflammation-induced alterations of lymphatic transcriptomes. In agreement, we recently reported that anti-IL1β treatment prevented pressure-overload-induced lymphangiogenesis as well as lymphatic Ccl21 upregulation (Heron et al, 2023b). However, it did not restore myocardial fluid balance in this model (Heron et al, 2023b), indicating that other changes in the cardiac microenvironment may contribute to lymphatic molecular alterations and transport dysfunction in HF.

Concerning new molecular targets to restore lymphatic function in CVDs, our current study revealed among the most notable changes a radical loss of Cldn5 and of lymphatic valves in cardiac lymphatics post-TAC. Single-cell analyses of the few remaining cells in the LEC3 cluster post-TAC in BALB/c lacked sensitivity to reveal potential dysregulated pathways, beyond loss of *Cldn5*, to explain this selective rarefaction. Of note, valve formation in lymphatics is strongly influenced by mechanosensitive (*Piezo1, Klf2, Klf4*) and/or calcineurin pathways (*Nfat*) (Petrova & Koh, 2018; Schulte-Merker et al, 2011). Intriguingly, we found that several mechano-relevant genes (*Klf2, Dtx1, Nfat5, Nfatc1, Itgb1*) were reduced in BALB/c cardiac LECs post-TAC. Based on these alterations, we speculate that reduced lymphatic flow in failing hearts may be involved in the reduced capacity of expanded lymphatic capillaries to generate new lymphatic valves, which in turn will limit lymphatic drainage efficiency. However, given that

the reduction of the *Klf2* gene in cardiac LECs post-TAC was a feature shared with cardiac BECs and vBECs, it is also possible that generalized vascular loss of mechanosensitive pathways may reflect global adaptations of cardiac ECs to chronic pressure overload.

In conclusion, our study revealed severe molecular and structural changes in cardiac lymphatics during pressure-overload-induced HF in BALB/c, with in contrast very few alterations noted during pressure-overload-induced pathological hypertrophy in C57. These striking strain-dependent differences uncovered in our study may be used to dissect what part of the lymphatic transport dysfunction in CVDs is *intrinsic* to altered lymphatics, and what part is *extrinsic,* i.e., due to cardiac contractile dysfunction. We expect that this may yield clinically relevant information on how to improve lymphatic drainage in CVDs to resolve myocardial edema and inflammation through modalities beyond therapeutic lymphangiogenesis.

# Methods

### Reagents and tools table

| Reagent/resource | Reference or source | Identifier or catalog number |
|---|---|---|
| **Experimental models** | | |
| Balb/c (*M. musculus*) | Janvier Labs | Adult female |
| C57BL6/J (*M. musculus*) | Janvier Labs | Adult female |
| Human dermal LECs, *HDLEC-c* | PromoCell | C-12216 |
| **Antibodies** | | |
| Anti-alphaSMA-FITC | Sigma-Aldrich | *F3777* |
| Anti-CD31/PECAM | BD | *553371* |
| Anti-CCL21 | R&D Systems | *AF457* |
| Anti-Claudin-5 | Invitrogen | *34-1600* |
| Anti-F4/80 | Abd Serotec | *MCA497R* |
| Anti-LYVE-1 | Reliatech | *103-PA50* |
| Anti-Biotinylated LYVE-1 | eBiosciences | ALY7 13-0443-82 |
| Anti-Pdl1 | Biolegend | 124302 |
| Anti-Podoplanin | eBioscience | *14-5381-82* |
| Anti-VE-cadherin | R&D Systems | *AF1002* |
| Donkey anti-rat AF488 | Jackson Immunoresearch | *712-545-153* |
| Donkey anti-rat Cy3 | Jackson Immunoresearch | *712-166-153* |
| Goat anti-rat AF647 | Thermo Fisher Scientific/ Invitrogen | *A-21247* |
| Donkey anti-rabbit Cy3 | Jackson Immunoresearch | *711-165-152* |
| Donkey anti-rabbit AF647 | Jackson Immunoresearch | *711-605-152* |
| Donkey anti-goat Dy488 | Interchim | *A50-201D2* |
| Donkey anti-goat Dy550 | Interchim | *A50-201D3* |
| Streptavidin FP547 | Interchim | *FP-CA5570* |

| Reagent/resource | Reference or source | Identifier or catalog number |
|---|---|---|
| Streptavidin FP647 | Interchim | *FP-CA5640* |
| Goat anti-Hamster AF488 | Thermo Fisher Scientific/ Invitrogen | *A-21110* |
| Anti-CD16/32 (Fc block) | BD Pharmingen | 553142 |
| Anti-CD31 | BD Pharmingen | 551262 |
| Anti-CD45 | Sony Biotechnology | 1115650 |
| Anti-Lyve1 | R&D Systems | FAB212SP |
| Anti-podoplanin | Sony Biotechnology | 1320070 |
| **Oligonucleotides and other sequence-based reagents** | | |
| Single Cell RNA Purification Kit | Norgen | 51800 |
| Chromium Next Gem Single Cell | 10X Genomics | 3' library kit v.3.1 |
| Viability Dye (L/D) | ebioscience | 65-0868-14 |
| Flow cell | Illumina | High-throughput kit 2×75 |
| **Chemicals, enzymes, and other reagents** | | |
| ECGM-MV2 | PromoCell, | CC-22121 |
| Recombinant human IL1β | PeproTech | 200-01B |
| Collagenase type XI | Sigma | #C7657 |
| Collagenase type I | Sigma | #C0130 |
| Hyaluronidase type I | Sigma | #H3506 |
| DNase 1 | Sigma | #DN25 |
| **Software** | | |
| FlowJo | FlowJo LLC | V10.8.1 |
| CellLoupe | 10X Genomics | NA |
| FIJI | NIH ImageJ | 1.54 g, Java 1.8.0_322, 64-bit |
| GraphPad | Prism | v. 8 |
| **Other** | | |
| FACS ARIA II | BD Biosciences | NA |
| Illumina Nextseq500 | Illumina | NA |
| Blaze ultramicroscope | Miltenyi | NA |
| TCS SP8 confocal microscope | Leica Microsystems | NA |
| Leica Thunder Tissue 3D microscope. | Leica Microsystems | NA |
| Epifluorescence microscope equipped with an apotome and Zen 2012 software | Zeiss | AxioImager J1 |
| GentleMACS Dissociator | Miltenyi Biotec | 8-well |

## Study approval

Animal experiments were approved by the regional Normandy University ethical review board Cenomexa according to French and EU legislation (APAFIS #23175-2019112214599474 v6; APAFIS #32433-2022070712508369 v2). Female BALB/c and C57BL6/J

mice (22–24 g) were obtained from Janvier (France). A total of 60 BALB/c and 30 C57BL6/J female mice, surviving TAC or sham-operation, were included in this study. Animal housing and experiments were in accordance with European Directive 2010/63/EU on the protection of animals. Animals were kept in a controlled conventional environment (12 h light/dark cycles, 23 °C) with free access to food and water, housed in standard cages in stable social groups of five mice per cage, and provided access to cage-enrichment according to EU legislation.

## Mouse in vivo model

Minimally invasive transversal aortic banding constriction (TAC) was performed on adult 8-week-old mice, using double-banding across a 26 G needle, as previously described (Heron et al, 2023a). Mice received intraperitoneal injection of xylazine (10 mg/kg Rompun® 2%, Bayer Health Care) and were placed on mechanical ventilation with Isoflurane (2–3%). The operator performed a minimal thoracotomy with an incision at the level of the first intercostal space. The aortic arch was visualized under low-power magnification. A snare, made of 7-0 polypropylene suture, was passed under the aorta between the origin of the right innominate and left common carotid arteries. Two suture bands were placed side-by-side to create an elongated stenosis and prevent inter-nalization of the suture, as described.(Lygate et al, 2006) A bent 26-gauge needle was placed next to the aortic arch, and the sutures were snugly tied around the needle and the aorta. After banding, the needle was quickly removed. The skin was closed, and mice recovered on a warming pad until fully awake. The sham procedure was identical except that the aorta was not banded. Buprenorphine (50 µg/kg, Buprecare®, Axcience) was injected subcutaneously 6 h after surgery and twice per day until 3 days post-operation. Euthanasia was performed by pentobarbital overdose (100 mg/kg Euthoxin) at 8 weeks post-TAC.

## Cardiac single-cell sorting, sequencing, and analyses

Cardiac single-cell suspensions were sorted by FACS (ARIA II, BD Biosciences). Briefly, at 8 weeks post-TAC, animals were eutha-nized by barbiturate overdose (100 mg/kg intraperitoneal, Eutoxin), and hearts were recovered after perfusion with warmed physiolo-gical saline through the abdominal aorta. LV samples were minced with scalpels and placed in ice-cold DMEM medium. Single cell suspensions were prepared, as described (Heron et al, 2023a), by 30 min incubation in tissue-dissociating solution (125 U/mL collagenase type XI, #C7657, Sigma; 450 U/mL collagenase type I, #C0130, Sigma; 60 U/mL hyaluronidase type I, #H3506, Sigma; and 60 U/mL DNase 1 #DN25, Sigma) using gentleMACS Dissociator (MACS; Miltenyi Biotec, Auburn, CA). Digested tissues were washed with DMEM culture medium and filtered to remove undigested tissue pieces (80 µm mesh and 40 µm mesh, BD Biosciences). Cardiac single cell suspensions were prepared in FACS buffer (5% BSA in PBS). To block nonspecific binding of antibodies to Fcγ receptors, isolated cells were first incubated with Fc-Block for 15 min at 4 °C. Subsequently, cells were stained with specific antibody cocktails (see Appendix Table S2) for 20 min at r.t, followed by washing with FACS buffer and resuspension in 700 µL phenol red-free RPMI with 5% FCS for sorting.

FACS sorting was performed on an FACS ARIA II (BD Biosciences). BECs were defined as live CD45-/CD31 + /Lyve1-cells, whereas LECs were identified as CD45-/CD31 + /Lyve1 + /Pdpn+ cells. In total, 3000–100,000 cells were collected into 50–500 µL 100% FCS and kept on ice prior to scRNAseq. Cells were then centrifuged and resuspended in PBS at a concentration of 1000 cells/µL for inclusion in 10X Genomics pipeline.

BECs were defined as live CD45$^-$/CD31$^+$/Lyve1$^-$ cells, whereas LECs were selected as CD45$^-$/CD31$^+$/Lyve1$^+$/Pdpn$^+$ cells. A 1:3 mix of cardiac LECs and BECs was included in each sample prepared using the 10X Genomics Chromium Next Gem Single Cell 3' kit.

Briefly, we charged 16,000 cells per reaction. For each group, 1–2 reactions were performed using cells from 10 pooled mouse hearts. The 10X Genomics Chromium Next Gem Single Cell 3' library kit v.3.1 was used to prepare barcoded mRNA following the guidelines from the supplier. Single Index kit T Set A was used to label each reaction. Samples were loaded onto 3 separate Illumina kit High-throughput $2 \times 75$ ( < 400 million reads) flow-cells and sequenced using Illumina Nextseq550, at an estimated sequencing depth of 20,000 reads per cell, using the settings of: *Read1*: 28 cycles; *Read2*: 8 cycles; *Read3 (i7)*: 114 cycles.

Raw FASTQ files, integrating single-index and cell barcodes, were generated and demultiplexed from BCL sequencing output files using bcl2fastq (v2.20.0)(Illumina). FastQC (v0.11.9)(Babra-ham Bioinformatics) was used for quality control, revealing >98% perfect index and high-quality reads (86% bases >= Q30), requiring no trimming or filtering, with the exception of BALB/c sham1 sample trimmed using FASTP (v0.20.0)(Chen et al, 2018) to remove adapters, poly-A, and poly-G present in Reads 2. In each sample, the STARsolo RNAseq aligner (v2.7) (Kaminow et al, 2021) mapped over 88% of reads to the mouse reference genome (GRCm39.111, Ensembl ftp server), generating a feature-barcode matrix (including *Read 1*, *Read 2*, and *i7* index) for each sample, based on feature (gene) counting per cell (UMI count matrix).

Data filtering, normalization, integration, clustering, and differential analysis were performed using the Seurat package (version 5.0.2) (Hao et al, 2024) in R (version 4.1.2). For low-quality data filtering, genes expressed by <10 cells, and cells expressing <500 genes for Balb/c samples or <300 for C57 samples, were excluded, as were cells with UMI numbers <500 or cells with >10% mitochondria-derived UMI counts. Matrices were normal-ized using "NormalizeData()" (LogNormalize, scale factor 10,000). The top 3000 highly variable features were selected using "FindVariableFeatures()" (vst method). To correct batch effects and integrate datasets, Seurat's anchor-based method with SNN clustering and Harmony (Korsunsky et al, 2019) were applied. Cell cycle heterogeneity was mitigated with "CellCycleScoring()" and mitochondrial UMI counts via "SCTransform()". The top 3000 variable genes were selected ("SelectIntegrationFeatures()"), and data integration was performed using "FindIntegrationAnchors()" and "IntegrateData()", producing integrated matrices.

Principal component analysis (PCA) was applied to the normalized and integrated data matrices, followed by Harmony to correct for any remaining batch effects. Principal components (PCs) were selected by accumulating their explained variance until reaching 90%, ensuring no unique component exceeded 5%. The first PC collection meeting these criteria was selected for downstream analysis. This included 42 dimensions for Balb/c data and 43 dimensions for C57 data. Unsupervised

cell clustering was performed using "FindNeighbors()" and "FindClusters()", applying the Leiden algorithm (Traag et al, 2019; Kelly). Cell clusters were visualized using 2D UMAP (Uniform Manifold Approximation and Projection). The resolution was increased from 0.1 to 1.0 in increments of 0.1. The optimal resolution was determined using the clustree functions from the R clustree package (Zappia and Oshlack, 2018), i.e., 0.6 for Balb/c data, and 0.9 for C57 data. Cell type automatic annotation was performed using the SingleR and Celldex R packages (Aran et al, 2019) with the ImmGen reference(Heng et al, 2008), before manual validation by inspecting key marker genes for each cluster. The filtered, integrated, and normalized transcriptomes clustered into 11 and 9 main populations, respectively. Among 3908 total sequenced cells from Balb/c mice, 3436 cells remained post-filtering, including 1932 cells from sham-operated controls and 1504 cardiac cells from TAC-operated mice. Among 4178 total sequenced cells from C57 mice, 1577 cells remained post-filtering, including 665 cells from sham-operated controls and 912 cells from TAC mice.

After identification of the three main endothelial cell (EC) cluster, BEC and LEC clusters were subdivided into sub-clusters for a more detailed analysis of subpopulations. Cardiac EC marker gene identification was performed using "FindMarkers" (test.use = "MAST", logfc.threshold=0) (Finak et al, 2015).

Functional enrichment analysis was performed using the R package "clusterProfiler" (v4.10.2) (Yu et al, 2012) with over-representation analysis (ORA) method for Gene Ontology and Pathways (KEGG, Reactome). Statistical output from the processed dataset, including Marker genes, DEGs, and functional enrichment analyses, are provided in Datasets EV1–EV11. DEGs were defined as genes up- or downregulated by at least 0.2 log2 fold change and a *padj*. <0.05. Only DEGs whose mean normalized read counts surpassed 0.3 are included.

Processed data (*expression_data_value.csv* for BALB/c and *expression_data.csv* for C57) was exported for visualization in CellLoupe (Loupe Browser 8.1.2, *10x Genomics*) used to generate UMAP and Violin plots. Violin plots were generated based on Kernel Density Estimation (KDE). The outermost points of violin correspond to the minimum and maximum values, and the median range is indicated by a box. Heatmaps and Volcano plots were generated based on Seurat-identified mean gene expression levels (plotted as log2-transformed *normalized read counts*), and log2 fold change (FC) and adjusted *P* values, respectively. Cell Loupe uses the exact negative binomial test (sSeq method) for the identification of differentially expressed genes.

At an estimated sequencing depth of ≈20,000 reads per cell, we detected 2500-6800 transcripts (unique molecular identifier, UMIs) and 1300–2300 genes (features) per cell in the processed datasets (Appendix Table S1). Statistical analyses were performed to determine differences in mean gene expression levels: (1) between clusters of cardiac ECs in either healthy or post-TAC mice, to identify marker genes; (2) between TAC and sham groups, within specific clusters, to identify impact of pathology. Only genes with a mean normalized read count >0.3 were included in differential analyses. Results of DEG analysis and functional enrichment analysis are presented in Datasets EV1–EV11.

## Validation by immunohistochemistry

Cardiac sections were analyzed by immunohistochemistry, as described (Heron et al, 2023a), to determine lymphatic expression of target genes identified by scRNAseq. Briefly, murine hearts were sectioned into a central slice, which was snap-frozen. Cardiac cryosections (8 µm thickness) were collected on SuperFrost plus glass slides. After fixation in acetone for 10 min, nonspecific binding sites were blocked in 3% BSA in PBS, followed by the Biotin-Avidin Blocking kit (Thermo Scientific) when streptavidin (SA)-conjugates were used to detect biotinylated secondary antibodies. Primary antibodies (see Appendix Table S3), diluted in 1% BSA in PBS, were incubated on the sections at r.t. for 1 h, followed by repeated washing in PBS and incubation with secondary antibodies for 30 min to 1 h. Multi-stainings were performed sequentially, and negative controls included the omission of primary antibodies. Slides were mounted in Vectashield containing DAPI, and images were acquired using ×20 or ×40 objectives on a Zeiss epifluorescence microscope (AxioImager J1) equipped with an apotome and Zen 2012 software (Zeiss), or using ×63 objectives on a Leica Thunder Tissue 3D microscope. Images were analyzed using Fiji imaging software (NIH) by an operator blinded to the treatment groups.

Whole-mount staining of cardiac lymphatics was performed, using a modified iDISCO$^+$ clearing protocol, for imaging by lightsheet and confocal laser scanning microscopy, as described (Heron et al, 2023a). Briefly, prior to sacrifice, deeply anesthetized mice were perfused with warm saline solution, followed by perfusion-fixation with warm 3% paraformaldehyde (PFA). Hearts were removed and postfixed (3% PFA) for 6 h. Following dehydration in graded methanol baths, and post-fixation in Dent's fixative, samples were bleached in graded $H_2O_2$ baths. After extensive blocking of nonspecific binding sites and tissue permeation with Triton-X100, cardiac lymphatics were visualized using antibodies (see Appendix Table S4) reactive against Lyve1, Podoplanin, CCL21, and/or Podocalyxin, followed by fluorescence-coupled secondary antibodies. Blood vasculature was visualized using anti-alpha-smooth muscle actin or Podocalyxin antibodies. Extensive washing was performed to remove nonspecific binding. Hearts were clarified using a modified iDISCO+ protocol based on incubation in graded methanol baths followed by incubation in dichloromethane (DCM, Sigma-Aldrich, #270997-12X100ML) and dibenzyl ether (DBE, Sigma-Aldrich, #108014-1KG) before lightsheet and confocal imaging.

3D imaging was performed by lightsheet microscopy as described (Heron et al, 2023a). Acquisitions were performed with an ultramicroscope II (LaVision BioTec) or Blaze (Miltenyi) using the ImspectorPro software (LaVision BioTec). The lightsheet was generated by a laser (wavelengths 561 or 640 nm, Coherent Sapphire Laser, LaVision BioTec) focused using two cylindrical lenses. A binocular stereomicroscope (MXV10, Olympus) with an ×2 objective (MVPLAPO, Olympus) was used at different magnifications (×0.8 and ×4). A dipping cap protective lens, including correction optics for MVPLAPO ×2 objective, was applied for working distances inferior or equal to 5.7 mm. Samples were placed in an imaging reservoir made of 100% quartz (LaVision BioTec) filled with DBE and illuminated from the side by the laser lightsheet. A PCO Edge SCMOS CCD camera (2560 × 2160 pixel size, LaVision BioTec) was used to capture images. The step size between each image was fixed at 6 µm (x0.8 zoom) or 2 µm (×3.2 zoom). All tiff images were generated in 16-bit.

For confocal laser scanning microscopy, imaging was performed with an upright fixed-stage TCS SP8 confocal microscope (Leica

Microsystems, France) equipped with multiple laser lines (wavelengths 561 or 640 nm). In order to acquire deep confocal views, a ×25 objective was used (numerical aperture 0.95, working distance 2500 μm, water immersion). Images were captured using a hybrid detector (Hamamatsu) in photon-counting z-stack mode. Maximal intensity projection views were generated with Fiji ImageJ software (1.54 g, Java 1.8.0_322, 64-bit).

Images, 3D volume, and movies from Lightsheet microscopy were generated using Imaris x64 software (version 8.0.1, Bitplane). Z-stack images were first converted to imaris file (.ims) using ImarisFileConverter, and 3D reconstruction was performed using the "volume rendering" function. To facilitate image processing, images were converted to 8-bit format. Optical slices were obtained using the "orthoslicer" tool. 3D pictures and movies were generated using the "snapshot" and "animation" tools. Movie reconstruction with .tiff series was performed with Fiji ImageJ software.

## Validation by in vitro culture of human LECs

Briefly, human LECs (PromoCell, Clonetics™ HMVEC-dLy, *CC-2812*) were seeded in six-well plates ($3 \times 10^5$/well) and grown to confluence in complete Endothelial Cell Basal Medium-2 (Promo-Cell, ECGM-MV2, CC-22121), including 5% FCS with VEGF, IGF, EGF, and FGF supplement, according to recommendations of the supplier. Cells were tested for Mycoplasma contamination using Invivogen MycoStrip®. After overnight serum-starvation (1% FCS, w/o VEGF, ECGF, EGF), cells were exposed, in at least triplicate, to 20 ng/mL recombinant human IL1β (200-01B, PeproTech) in 2 mL per well of serum-starvation media. After 24 h, cells were recovered and RNA extracted (Single Cell RNA Purification Kit, Norgen, 51800) for subsequent mRNAseq analyses and bioinformatics analyses by Biomarker Technologies (BMK) company (Munster, GE) for identification of differentially expressed genes. Briefly, poly-A mRNA-seq, sequencing length of 150 nt paired-end (PE150), were performed with IlluminaSeq. Reads were aligned to the human genome (GRCh38). Transcript quantification was performed with featureCounts18, and differential analysis, including normalization of the original readcount, was performed with DESeq2 bioconductor package19. Statistical test (DESeq2 Wald test) was conducted on the expression matrix after standardization. Genes significantly up- or downregulated upon treatment were identified (for RNAseq data see Dataset EV12), and results are reported as Fpkm and as log2 fold of control condition (1% FCS in ECM without cytokines and growth factors).

## Graphical design

Graphics in Figs. 7 and EV3B, and the synopsis illustration were created with BioRender.com https://BioRender.com/ftwcfku.

## Statistics for bulk RNAseq and immunohistochemistry

Bulk RNAseq data from human LEC cultures were analyzed for gene expression levels and DEGs by BMK using the DESeq2 Bioconductor package (Wald test). Data is reported as Fpkm normalized reads and as fold change (FC) of control unstimulated cells. Animals were randomly assigned to either sham or TAC

**The paper explained**

**Problem**
The cardiac lymphatic system is essential for clearing immune cells, debris, and interstitial fluid after cardiac injury, thereby limiting adverse remodeling. In heart failure and other cardiovascular diseases, inflammatory mediators may compromise lymphatic integrity and impair drainage function, exacerbating myocardial edema and inflammation. We sought to define the cellular and molecular mechanisms underlying lymphatic dysfunction in pressure-overload-induced Heart Failure.

**Results**
Single-cell transcriptomic profiling of cardiac lymphatic endothelial cells revealed that pressure overload triggers downregulation of key lymphatic barrier genes, including *Cldn5* and *Tjp1*, which may contribute to increased lymphatic leakage and inefficient transport. Newly formed lymphatic capillaries were also found to lack functional valves, impairing lymph transport from the myocardium toward draining lymph nodes. In contrast, our data showed that cardiac lymphatics upregulated immune-modulatory and chemoattractant molecules, such as *Ccl21* and *Lyve1*, suggesting compensatory enhancement of immune cell clearance.

**Impact**
Our findings uncover multiple inflammation-driven mechanisms that may compromise cardiac lymphatic function in Heart Failure. By identifying molecular pathways governing lymphatic barrier integrity and valve formation, this study highlights potential therapeutic targets to restore lymphatic transport, accelerate resolution of myocardial inflammation and edema, and mitigate Heart Failure progression.

groups. Exclusion criteria were not used. Immunohistochemical analyses were performed by observers blinded to the treatment groups. Sample sizes ranged from 10 to 20 mice per group for in vivo studies, and three samples per group for in vitro RNAseq analyses. Statistical analyses for immunohistochemical data were performed using GraphPad Prism software. Data are presented as mean ± s.e.m. Comparisons of two independent groups were performed using either Student's two-tailed *t* test for groups with normal distribution, or alternatively by the Mann–Whitney *U* test for samples where normality could not be ascertained based on D'Agostino & Pearson omnibus normality test.

## Data availability

The original data included in this study have been deposited in GEO (GSE289738, *Effect of interleukin-1beta on confluent human lymphatic endothelial cells*; GSE290576 *Effect of pressure-overload on cardiac endothelial cells in C57BL6/J mice*; GSE291154 *Effect of pressure-overload on cardiac endothelial cells in Balb/c mice*). The raw sequencing files (*fq.gz.*) for GSE289738 were partially missing (storage error) and only the expression matrix files (*all gene.csv*) and associated metadata (*.xlsx*) are available online for this dataset.

The source data of this paper are collected in the following database record: biostudies:S-SCDT-10_1038-S44321-025-00345-w.

# Peer review information

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

## Acknowledgements

We thank Dr Maria Goes (Max-Planck-Institute for Heart and Lung Research) for helpful suggestions for Ccl21 whole-mount staining; Ms Marine Panza (Univ Rouen) for assistance with immunohistochemistry; and Mr Clément Lecler (Univ Rouen) for assistance with library generation for scRNAseq. CH, TL, and CV were funded by fellowships from the Normandie Doctoral School (EdNBISE). OL was co-funded by a fellowship "Allocation 50% Normandie Recherche".

This work was supported by the Fondation pour la Recherche Médicale (FRM) [*FDT202204014960*] to CH and [*FDT202404018270*] to TL. VT was supported by the Chair of Excellence program "*Lymphcosign*" from the Normandy Region (RIN Recherche), which also supported the work. The Inserm UMR1096 laboratory was supported by a grant from the GCS G4 and the Normandy region (FHU CARNAVAL). The project also benefited from a joint grant (EB and AZ) from the French National Research Agency (ANR) with the Deutsche ForschungsGemeinschaft (DFG) for the project "CITE-LYMPH" [*ANR-22-CE92-0040-001*; DFG project number 505700170], the DFG project #453989101-CRC1525 to AZ, and generalized institutional funds (Inserm UMR1096) from the French Inserm, Rouen University (BQRI 2022, 2023), and targeted funding from the Normandy Region (CPER 2021). The project also benefited from EU-Normandy region co-funds (RIN 2018 SINGLE C, to SC): "*L'Europe s'engage en Normandie avec le Fonds Européen de Développement Régional*". We acknowledge the France-BioImaging infrastructure (https://ror.org/01y7vt929) supported by the French National Research Agency (ANR-24-INBS-0005 FBI BIOGEN).

## Author contributions

**Coraline Heron**: Conceptualization; Data curation; Investigation; Writing—review and editing. **Theo Lemarcis**: Conceptualization; Data curation; Formal analysis; Investigation; Writing—review and editing. **Océane Laguerre**: Data curation; Formal analysis; Validation; Investigation; Writing—review and editing. **Bénjamin Bourgeois**: Data curation; Software; Formal analysis; Visualization; Methodology; Writing—review and editing. **Corentin Thuilliez**: Data curation; Software; Formal analysis; Investigation; Writing—review and editing. **Chloé Valentin**: Validation; Investigation; Visualization; Writing—review and editing. **Anais Dumesnil**: Data curation; Formal analysis; Investigation; Visualization; Methodology; Writing—review and editing. **Manon Valet**: Investigation; Methodology; Writing—review and editing. **David Godefroy**: Data curation; Investigation; Methodology; Writing—review and editing. **Damien Schapman**: Investigation; Visualization; Methodology; Writing—review and editing. **Gaetan Riou**: Resources; Formal analysis; Validation; Methodology; Writing—review and editing. **Sophie Candon**: Conceptualization; Supervision; Funding acquisition; Validation; Writing—review and editing. **Céline Derambure**: Resources; Formal analysis; Investigation; Methodology; Writing—review and editing. **Alma Zernecke**: Visualization; Methodology; Writing—review and editing. **Caroline Berard**: Formal analysis; Methodology; Writing—review and editing. **Hélène Dauchel**: Conceptualization; Formal analysis; Supervision; Writing—original draft; Project administration; Writing—review and editing. **Virginie Tardif**: Conceptualization; Supervision; Writing—review and editing. **Ebba Brakenhielm**: Conceptualization; Data curation; Formal analysis; Supervision; Funding acquisition; Investigation; Visualization; Methodology; Writing—original draft; Project administration; Writing—review and editing.

Source data underlying figure panels in this paper may have individual authorship assigned. Where available, figure panel/source data authorship is listed in the following database record: biostudies:S-SCDT-10_1038-S44321-025-00345-w.

## Disclosure and competing interests statement

The authors declare no competing interests.

# Expanded View Figures

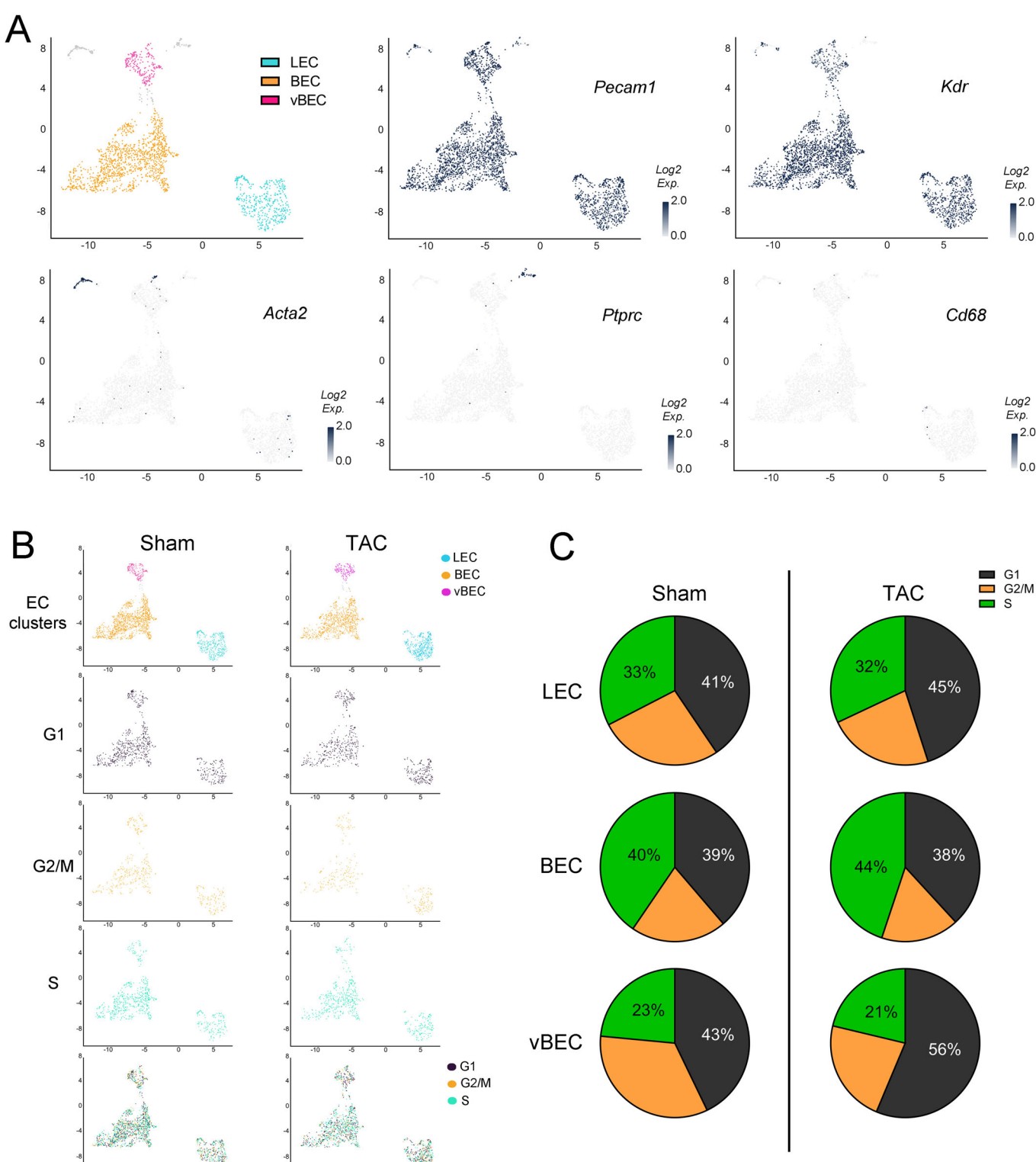

**Figure EV1.  Marker genes and cycle phase distribution of cardiac endothelial cells.**

(**A**) Visualization of cardiac EC clusters from healthy BALB/c mice ($n = 20$). The three main EC clusters expressed vascular markers (*Pecam1, Kdr*), but not immune (*Ptprc, Cd68*) or mural cell markers (*Acta2*). Expression levels shown as *Log2 normalized read counts*. (**B, C**) Visualization and quantification of CellLoupe-based scoring of cell cycle phases in main EC clusters from healthy and post-TAC BALB/c hearts.

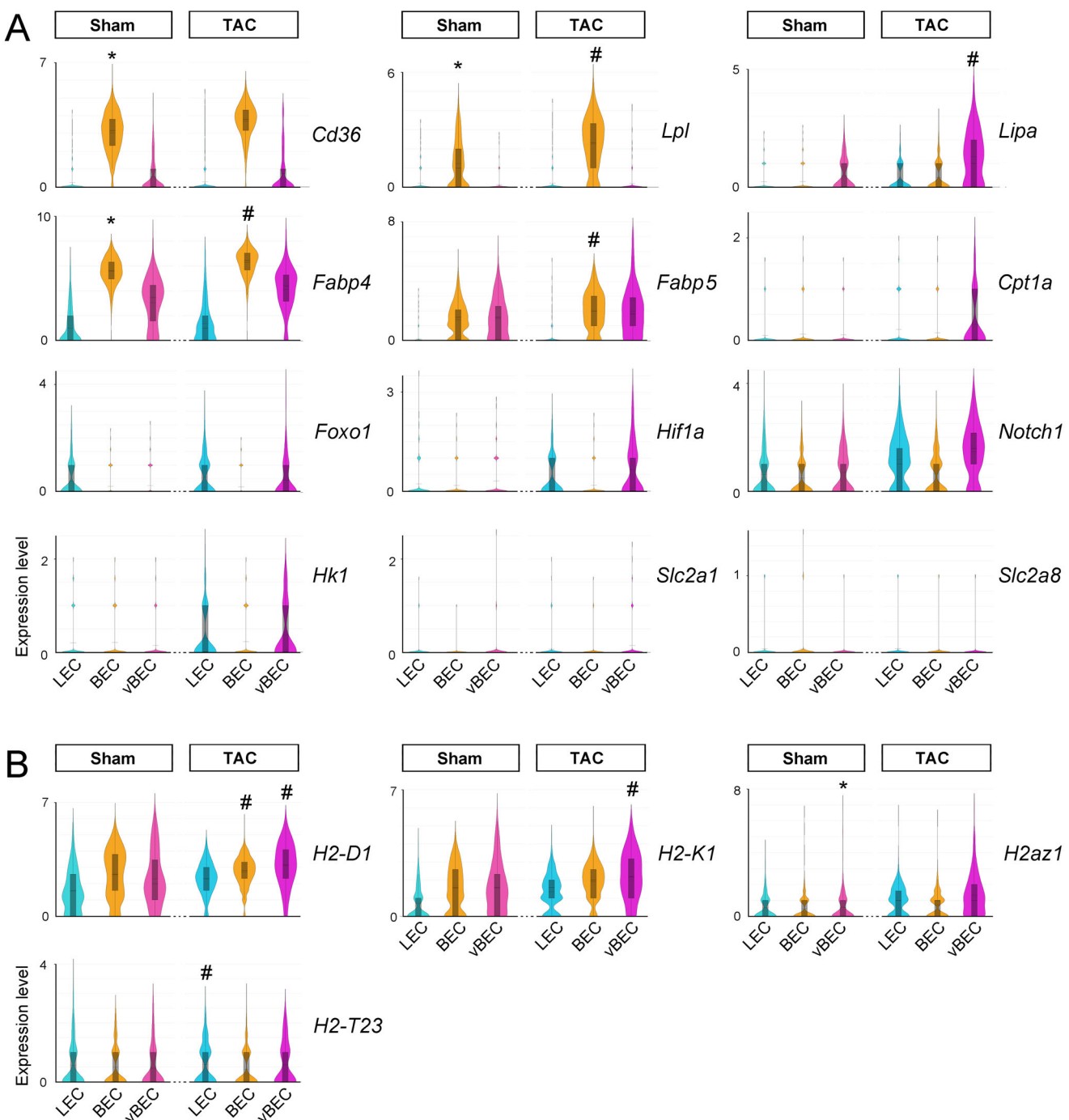

**Figure EV2.   Distinguishing EC markers related to metabolic and MHC class I genes.**

(A) Examples of expression levels (*Log2 normalized read counts*) of genes involved in regulation of metabolism, including lipolysis (*Cd36, Lpl, Lpa, Fabp4, Fabp5, Cpt1a*) and glycolysis (*Foxo1, Hif1a, Notch, Hk1, Slc2a1, Slc2a8*). (B) Comparison of gene expression levels of main MHC class I molecules for antigen presentation in cardiac EC clusters from healthy ($n = 20$) and post-TAC ($n = 10$) BALB/c mice. Clusters significantly enriched for a given gene are denoted (*). For a full list of marker genes of EC clusters in healthy hearts, and exact *P* values, see Dataset EV1. Significantly altered genes post-TAC are denoted (#). For full lists of DEGs post-TAC see Datasets EV4 (LEC), EV5 (BEC), and EV6 (vBEC). *Cpt1a*, Carnitine palmitoyl transferase*; Fabp*, Fatty acid binding protein*; Foxo1*, Forkhead box protein O; *Hif1a*, Hypoxia-inducible factor; *Hk1*, Hexokinase-1; *Lpl*, Lipoprotein Lipase; *Lpa*, Lipase A; *Slc2a1*, Glut1; *Slc2a8*, Glut8.

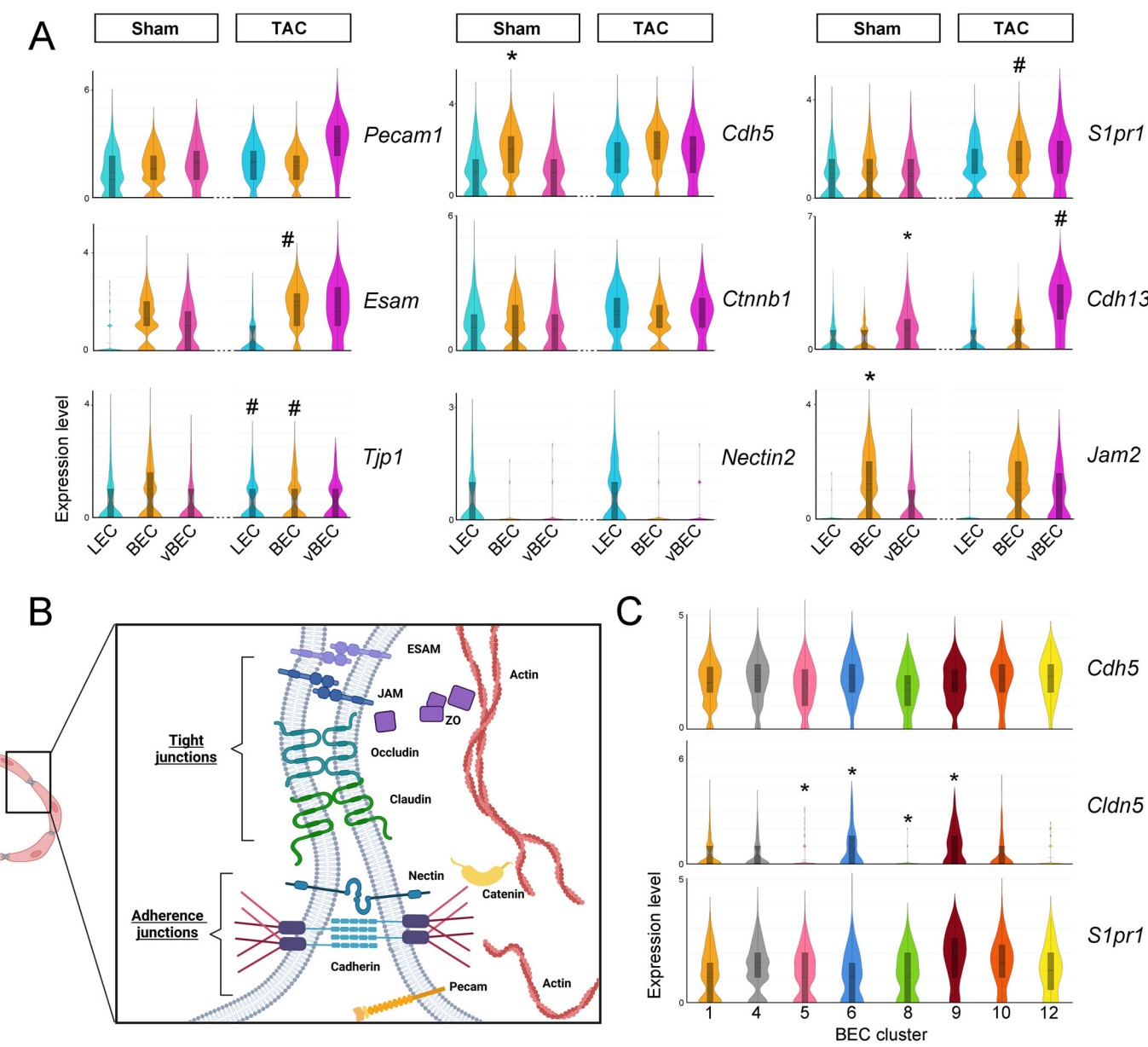

**Figure EV3. EC cluster differences in key genes involved in vascular barrier regulation.**

(A) Examples of expression levels (*Log2 normalized read counts*) of genes involved in adherence junctions and tight junctions in ECs from healthy (*n* = 20) and post-TAC (*n* = 10) BALB/c mice. Clusters significantly enriched for a given gene are denoted (*); whereas genes significantly altered post-TAC are denoted (#). For a full list of EC marker genes, and exact *P* values, see Dataset EV1, and for DEGs post-TAC see Datasets EV4 (LEC), EV5 (BEC), and EV6 (vBEC). (B) Schematic overview of junctional assembly. (C) Examples of barrier-relevant gene expression levels in healthy cardiac BEC subpopulations. For a list of BEC subpopulation marker genes see Dataset EV2, and for full list of DEGs in BEC subpopulations post-TAC, see Dataset EV8. *Cdh5*, VE-Cadherin; *Cdh13*, T-cadherin; *Ctnnb1*, β-Catenin; *Esam*, Endothelial cell adhesion molecule; *Jam*, Junctional adhesion molecule; *S1pr1*, Sphingosine-1-phosphate receptor 1; *Tjp1*, Tight junction protein-1/ZO-1.

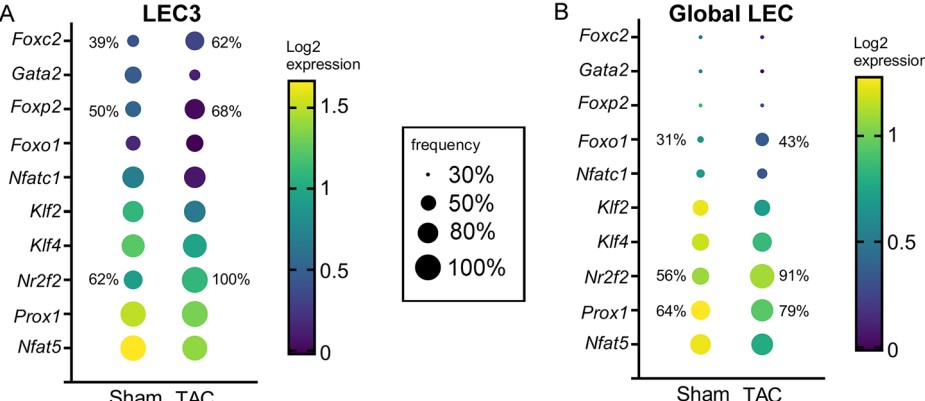

**Figure EV4. Frequency of expression of LEC marker genes.**

(A) Bubble plot illustration of lymphatic transcription factor expression in LEC3 subpopulation in healthy ($n = 20$) and post-TAC ($n = 10$) BALB/c mice (expression levels color-coded as *Log2 normalized read counts*, population frequency of expression coded as bubble size). (B) Bubble plot illustration of these genes in the global LEC cluster. For a full list of LEC marker genes, see Dataset EV3.

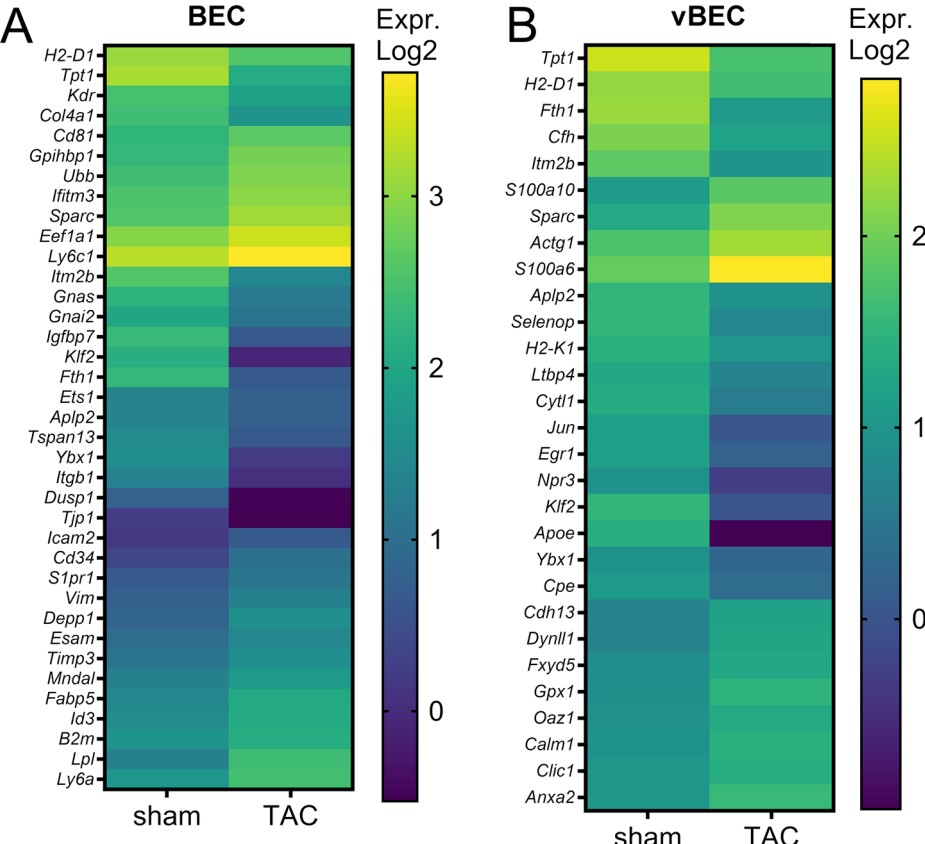

**Figure EV5. Differentially expressed genes post-TAC in cardiac BEC and vBEC clusters.**

(A) Examples of genes differentially expressed post-TAC (*Log2 normalized read counts*) in cardiac BECs in BALB/c mice. (B) Examples of genes differentially expressed post-TAC in vBECs. For a full list of DEGs see Dataset EV6 and EV7, respectively.

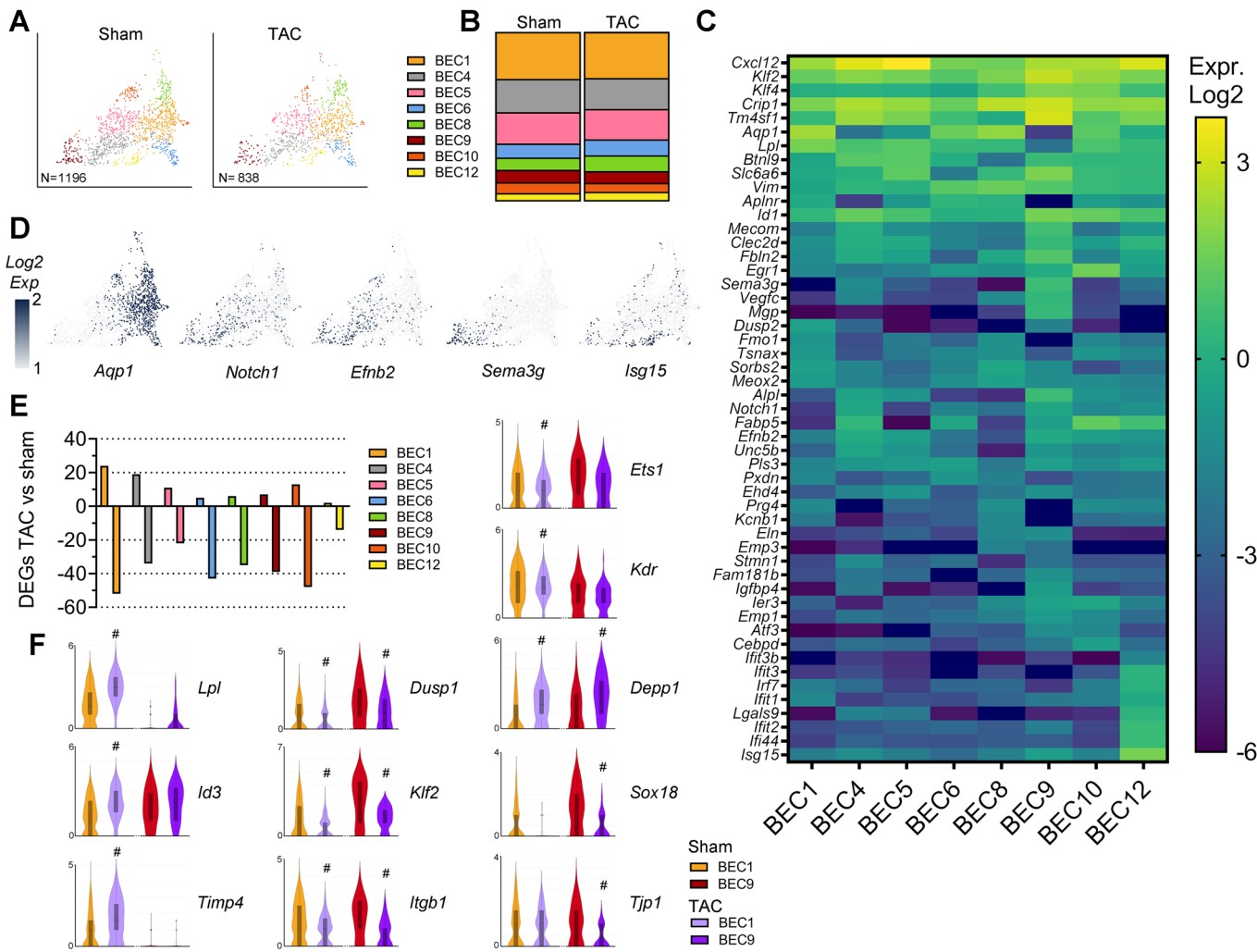

**Figure EV6.  Subpopulation analyses of cardiac BECs post-TAC in Balb/c.**

(A) Visualization of cardiac BEC subpopulations in healthy ($n = 20$) and post-TAC ($n = 10$) BALB/c mice. (B) BECs from healthy and TAC-operated mice clustered into 8 subpopulations, with the majoritarian BEC1 cluster representing around 30% of cells in both sham and TAC hearts. (C) Examples of significantly enriched marker gene expression levels (*Log2 transformed normalized read counts*) in BEC subpopulations. Capillary gene markers were enriched in clusters BEC1, 5, 6, 8, and 10, while arterial gene markers were enriched in clusters BEC4, 5, 9, and 10, while the rare BEC12 cluster displayed an interferon-type signature. For list of marker genes see Dataset EV2. (D) Examples of marker gene expression distribution in the global BEC cluster. (E) Quantification of DEGs identified post-TAC in each BEC cluster. (F) Examples of gene expression levels in sham (*orange* or *red*) and TAC (*purple*) groups in BEC1 and BEC9 clusters. Significantly altered genes indicated (#) for the respective cluster. For a full list of DEGs for BEC clusters, and exact p-values, see Dataset EV8.

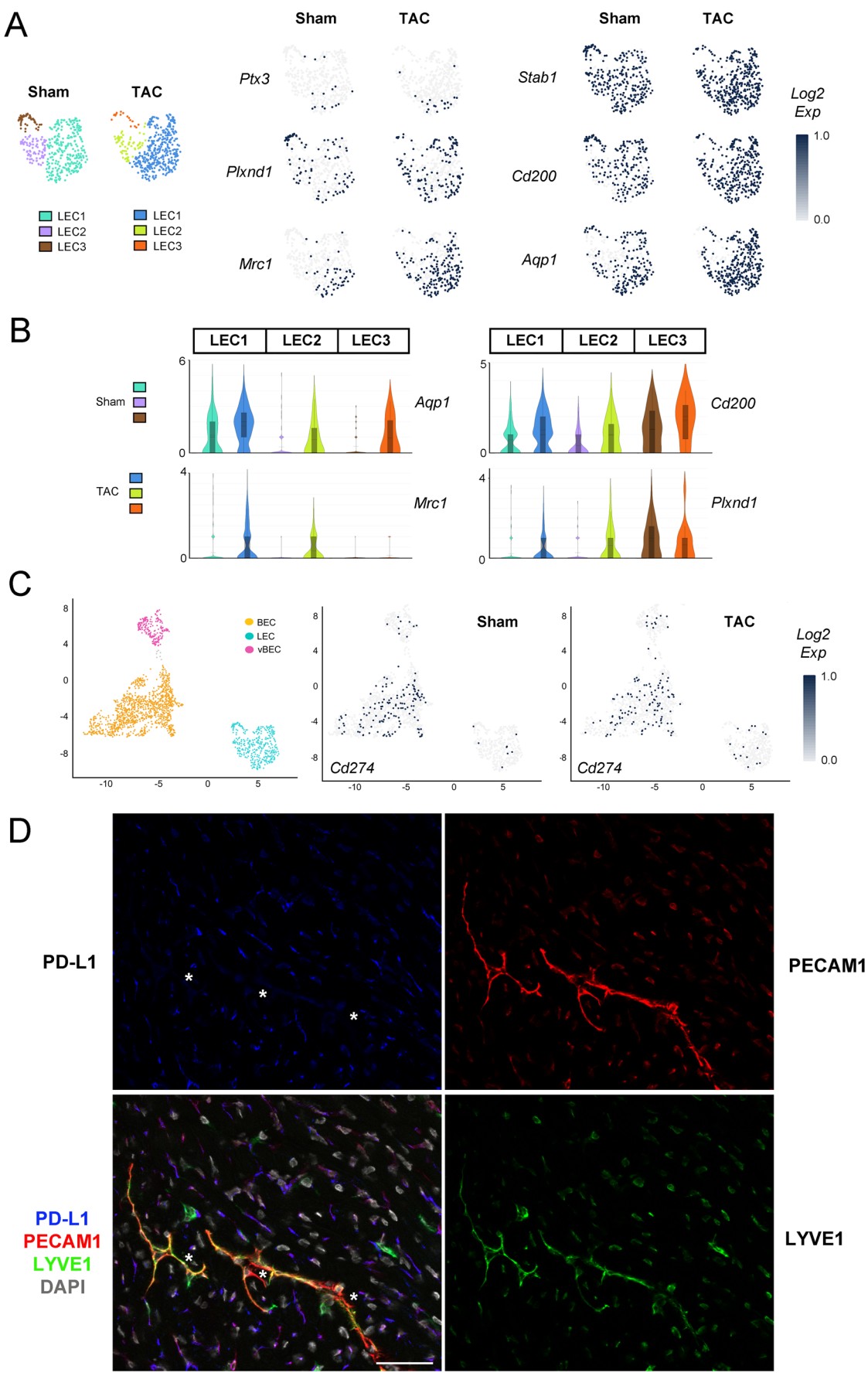

◀ **Figure EV7. Cardiac LEC expression of "immune LEC" marker genes.**

(A) Visualization of cardiac LEC subpopulations in healthy ($n = 20$) and post-TAC ($n = 10$) BALB/c mice, and LEC cluster distribution of marker genes proposed for immune" LEC (iLEC) proposed by Petkova et al. (B) Gene expression levels (*Log2 normalized read counts*) of proposed iLEC markers analyzed in LEC subpopulations in healthy and post-TAC Balb/c mice. (C) Cardiac EC expression distribution [umap] of *Cd274* (Pdl1) in BECs, vBECs, and LECs in healthy and post-TAC Balb/c mice. (D) Vascular PD-L1 protein levels evaluated by immunohistochemistry in cardiac section at 8 weeks post-TAC in Balb/c. PD-L1, *blue*; Lyve1, *green*; CD31, *red*: DAPI, gray. Scale bar 50 μm. *White Asterix*: lymphatics lacking PD-L1 expression. *Bottom left panel*: Purple cells, CD31[+] blood vessel capillaries expressing Pd-L1; Green cells, Lyve1[+] CD31[neg] macrophages.

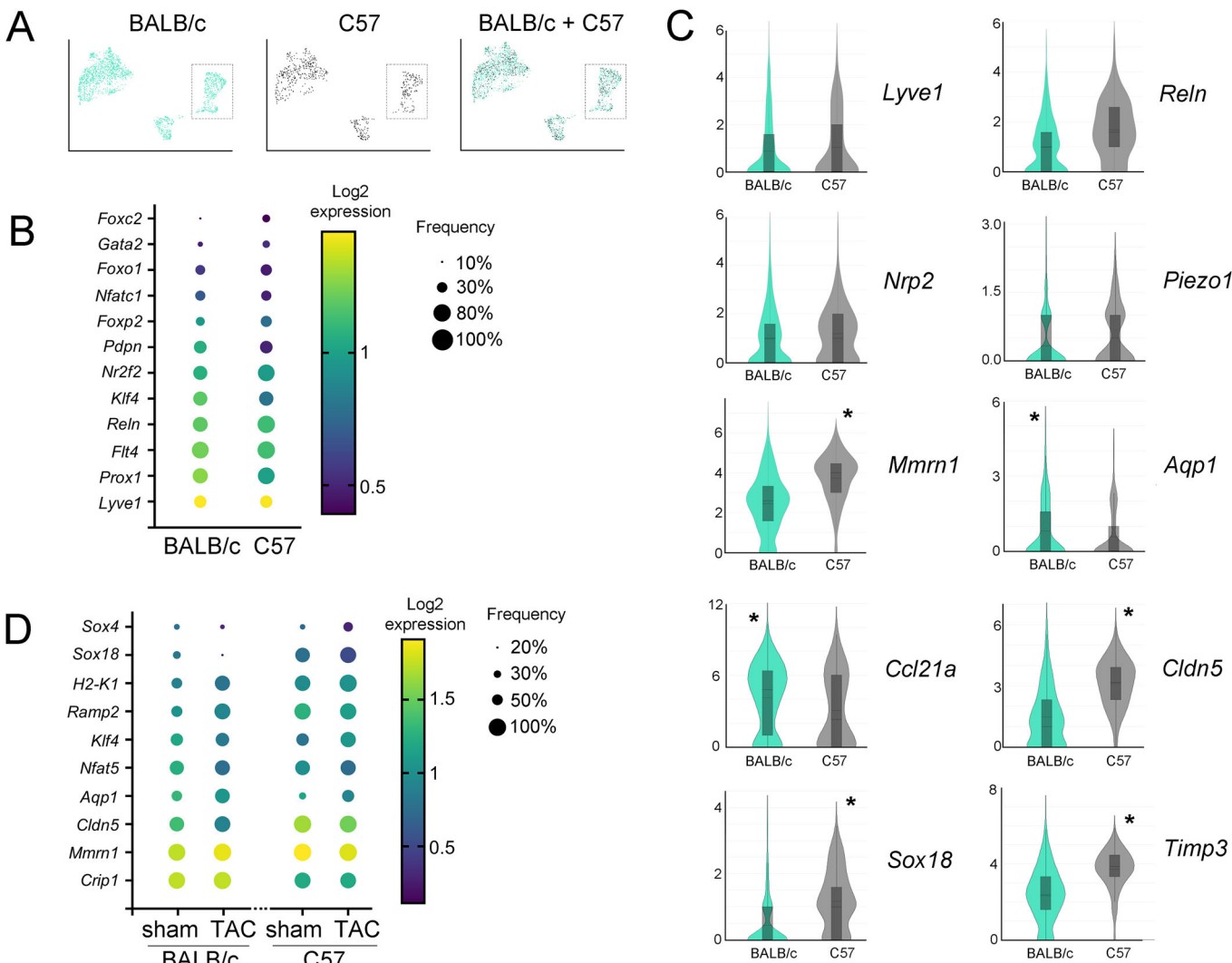

**Figure EV8. Comparing LEC markers between strains.**

(A) UMAP clustering of cardiac vascular EC populations in BALB/c (N = 1814 transcriptomes, *blue*) and C57 (N = 615, *gray*) from healthy mice (n = 10–20 mice/group). The LEC populations are highlighted by dashed lines. (B) Comparison of cardiac LEC expression of key lymphatic genes (expression levels color-coded as *Log2 normalized read counts*, population frequency of expression coded as bubble size) in BALB/c vs C57 healthy mice. (C) Examples of expression levels (*Log2 normalized read counts*) of key lymphatic markers, including some differentially expressed between BALB/c and C57 healthy mice (indicated by *). For a full list of DEGs, and their associated exact p-values, distinguishing the two strains see Dataset EV10. (D) Comparison of cardiac LEC expression of key lymphatic genes (expression level color-coded as *Log2 normalized read counts*, population frequency of expression coded as bubble size) in BALB/c vs C57 healthy or post-TAC mice. Of note, while these genes were not significantly altered post-TAC in C57 mice, many were significantly changed post-TAC in BALB/c (e.g. *Ccl21a, Lyve1, Klf4, Nfat5, Cldn5*).

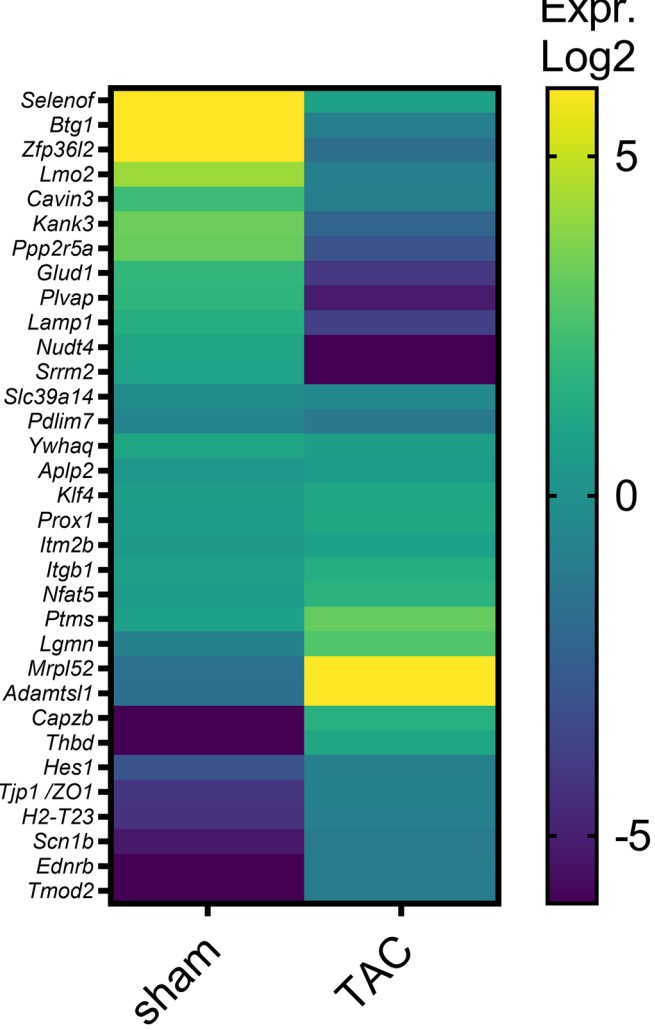

**Figure EV9.    Differentially-expressed genes in cardiac LECs post-TAC shared with IL1β-stimulated LECs.**

Examples of expression levels [*Log2 normalized read counts per cluster*] of genes significantly altered post-TAC in BALB/c mice. These genes were all similarly altered in vitro in IL1β-treated human LECs. For a list of all DEGs in human LECs, see Dataset EV12.

