## [Peer Review File · EMBO Molecular Medicine]

Molecular determinants of cardiac lymphatic dysfunction in a chronic pressure-overload model

Coraline Heron, Theo Lemarcis, Oceane Laguerre, Benjamin Bourgeois, Corentin Thuilliez, Chloe Valentin, Anais Dumesnil, Manon Valet, David Godefroy, Damien Schapman, Gaetan Riou, Sophie Candon, Celine Derambure, Alma Zerneck, Caroline Berard, Helene Dauchel, Virginie Tardif, and Ebba Brakenhielm

Corresponding authors: *Ebba Brakenhielm (ebba.brakenhielm@inserm.fr)* , *Helene Dauchel (helene.dauchel@univ-rouen.fr)*

Review Timeline:

Submission Date:	26th Mar 25
Editorial Decision:	30th Apr 25
Revision Received:	5th Sep 25
Editorial Decision:	8th Oct 25
Revision Received:	23rd Oct 25
Accepted:	4th Nov 25

Editor: Lise Roth

Transaction Report:

30th Apr 2025

Dear Prof. Brakenhielm,

Thank you for the submission of your manuscript to EMBO Molecular Medicine. We have now received the feedback from the three reviewers who agreed to evaluate your manuscript. As you will see from the reports below, the referees acknowledge the interest of the findings and overall quality of the experiments but also regret the descriptive nature of the findings.

We discussed these reports further within the team and with the reviewers, and agreed that with minimal experimental work, additional discussion and clarifications, the study could be well suited as a Resource in our journal. With that in mind, we would therefore like to invite you to submit a revised version of your manuscript.

Addressing the reviewers' concerns in full (experimentally or by adequate discussion) will be necessary for further considering the manuscript in our journal, and acceptance of the manuscript will entail a second round of review. EMBO Molecular Medicine encourages a single round of revision only and therefore, acceptance or rejection of the manuscript will depend on the completeness of your responses included in the next, final version of the manuscript. For this reason, and to save you frustration at the end, I would strongly discourage you from returning an incomplete revision.

We are expecting your revised manuscript within three to four months, if you anticipate any delay, please contact us.

We require:

4) A .docx formatted letter INCLUDING the reviewers' reports and your detailed point-by-point responses to their comments. As part of the EMBO Press transparent editorial process, the point-by-point response is part of the Review Process File (RPF), which will be published alongside your paper.

5) A complete author checklist, which you can download from our author guidelines (<https://www.embopress.org/page/journal/17574684/authorguide#submissionofrevisions>). Please insert information in the checklist that is also reflected in the manuscript. The completed author checklist will also be part of the RPF.

6) All Materials and Methods need to be described in the main text using our 'Structured Methods' format. According to this format, the Methods section includes a Reagents and Tools Table (listing key reagents, experimental models, software and relevant equipment and including their sources and relevant identifiers) followed by a Methods and Protocols section describing the methods, ideally using a step-by-step protocol format. The aim is to facilitate adoption of the methodologies across labs. Please download and fill our Reagents and Tools Table template (.docx), which you can find in our author guidelines:

7) Please note that all corresponding authors are required to supply an ORCID ID for their name upon submission of a revised manuscript.

8) It is mandatory to include a 'Data Availability' section after the Materials and Methods. Before submitting your revision, primary datasets produced in this study need to be deposited in an appropriate public database, and the accession numbers and database listed under 'Data Availability'. Please remember to provide a reviewer password if the datasets are not yet public (see <https://www.embopress.org/page/journal/17574684/authorguide#dataavailability>).

9) For data quantification: please specify the name of the statistical test used to generate error bars and P values, the number (n) of independent experiments (specify technical or biological replicates) underlying each data point and the test used to calculate p-values in each figure legend. The figure legends should contain a basic description of n, P and the test applied. Graphs must include a description of the bars and the error bars (s.d., s.e.m.). Please provide exact p values.

10) Our journal encourages inclusion of *data citations in the reference list* to directly cite datasets that were re-used and obtained from public databases. Data citations in the article text are distinct from normal bibliographical citations and should directly link to the database records from which the data can be accessed. In the main text, data citations are formatted as follows: "Data ref: Smith et al, 2001" or "Data ref: NCBI Sequence Read Archive PRJNA342805, 2017". In the Reference list, data citations must be labeled with "[DATASET]". A data reference must provide the database name, accession number/identifiers and a resolvable link to the landing page from which the data can be accessed at the end of the reference. Further instructions are available at .

11) We replaced Supplementary Information with Expanded View (EV) Figures and Tables that are collapsible/expandable online. EV Figures should be cited as 'Figure EV1, Figure EV2' etc... in the text and their respective legends should be included in the main text after the legends of regular figures.

12) The paper explained: EMBO Molecular Medicine articles are accompanied by a summary of the articles to emphasize the major findings in the paper and their medical implications for the non-specialist reader. Please provide a draft summary of your article highlighting

13) Author contributions: CRediT has replaced the traditional author contributions section because it offers a systematic machine readable author contributions format that allows for more effective research assessment. Please remove the Authors Contributions from the manuscript and use the free text boxes beneath each contributing author's name in our system to add specific details on the author's contribution. More information is available in our guide to authors.

Please also suggest a visual abstract to illustrate your article as a PNG file 550 px wide x 300-600 px high. A cropped portion of this image will serve as thumbnail for the table of content on our webpage.

16) As part of the EMBO Publications transparent editorial process initiative (see our Editorial at <http://embomolmed.embopress.org/content/2/9/329>), EMBO Molecular Medicine will publish online a Review Process File (RPF) to accompany accepted manuscripts.

In the event of acceptance, this file will be published in conjunction with your paper and will include the anonymous referee reports, your point-by-point response and all pertinent correspondence relating to the manuscript. Let us know whether you agree with the publication of the RPF and as here, if you want to remove or not any figures from it prior to publication. Please note that the Authors checklist will be published at the end of the RPF.

I look forward to receiving your revised manuscript.

Yours sincerely,

Lise Roth

***** Reviewer's comments *****

Referee #1 (Comments on Novelty/Model System for Author):

The authors have identified 2 strains of mice in which the cardiac lymphatic vessels respond distinctly to pressure overload. I think this is a powerful discovery and can be used efficiently to identify the mechanisms of lymphatic vessel growth.

Referee #1 (Remarks for Author):

The Brankenhielm lab has been the pioneers in investigating the relationship between lymphatic vessels and cardiac health. In 2016 they showed that the induction of lymphangiogenesis with VEGF-C reduces myocardial edema and fibrosis following myocardial infarction (MI) in rats (Henri et al., 2016).

In 2020 they paradoxically showed that the inhibition of lymphangiogenesis with soluble VEGFR3 improves cardiac function following MI (Houssari et al., 2020).

In 2023 they showed that pressure overload promotes cardiac lymphangiogenesis in mice in a strain- and sex-dependent manner (Heron et al., Cardiovascular Research. 2023). Specifically, pressure overload induced lymphatic vessel growth only in female BALB/c mice resulting in reduced inflammation, but did not affect fibrosis or heart failure. Lymphatic vessel density did not increase in C57BL6 mice although soluble VEGFR3 reduced lymphatic vessel density, increased inflammation and fibrosis, increased left ventricle dilatation and aggravated heart failure. However, there was no change in myocardial edema.

Furthermore, they showed that this lymphatic growth in BALB/c mice is at least partially mediated by IL-1b signaling pathway (Heron et al., Biorxiv. 2023). Inhibition of IL-1B only delayed left ventricle dilation and did not prevent heart failure in these mice. In this follow up study Heron et al use scRNA-seq to characterize the mechanisms that regulate lymphangiogenesis in BALB/c, but not C57BL6 mice. This work is well-done and potentially important. However, due to the above-mentioned, sometimes conflicting and confusing data, a better explanation of their models is needed. A few additional experiments are also needed to understand the cell-autonomous and non-cell autonomous mechanisms that regulate lymphatic vessel growth.

Major concerns and comments

1. As described above, it is not entirely clear if lymphatic vessel growth is preventing or aggravating MI, edema, fibrosis and heart failure. These phenotypes could be species (rat, mouse, human), strain (C57BL6, BALB/c), disease (MI, pressure-overload, ischemia reperfusion, fibrosis, inflammation) and sex specific. A figure or a table at the very beginning of the manuscript summarizing these various results and the working hypothesis of this manuscript based on these results will be very helpful. If possible, I also suggest incorporating the works of other labs that have also presented conflicting results (Oliver, Kahn, Caron etc).
2. The authors state that they performed scRNA-seq after enriching for Lyve1+ Pdpn+ LECs. If so, there should be no BECs or vBECs in the sample. I think they have enriched for all ECs. This mistake must be rectified.
3. Vegf-c, Vegf-d (Figure 2, Heron et al., 2023) and presumably IL-1b (Heron et al., Biorxiv 2023) are necessary for promoting lymphangiogenesis. However, we currently do not know where these signals are coming from. Therefore, it is also important to perform scRNA-seq by using the whole heart. Additionally, it is not clear which cells are inducing fibrosis by secreting collagen (Figure 6, Heron et al., 2023). scRNA-seq using entire heart will help resolve these questions.
4. IL-1B is likely responsible for the cell-junction defects in BALB/c mice. However, I am not sure how to interpret the

downregulation of Prox1, a critical molecule for lymphangiogenesis.

5. In figure 4G the sham and TAC induced changes in the 3 types of LECs in BALB/c mice are presented nicely. A similar comparison must be done for LECs in C57BL6 mice in figure 5 (not just for junctional molecules).

6. Please compare the LECs of BALB/c and C57BL6 mice under sham and TAC conditions.

Minor concern

1. The figure legends are very hard to comprehend. For example,

Examples of mean expression levels (Log₂ normalized read counts) of altered genes [heatmap, f], [violin plot, g] in cardiac LEC subpopulations from healthy and post-TAC hearts. Significant genes indicated (#) for each cluster in g.

Please try to make it easier for the readers to understand the legends. Describe the various panels in separate sentences. The panels in the figures are with capital letters, but figure legends are with small letters and placed at the end of the sentences. Something like the following will be good with the panel's identity mentioned at the beginning.

Fig. 1 Foxc2 loss disrupts transport function of adult lymphatics.

(A) FOXC2 in adult mesenteric LVs. Green, FOXC2; red, PROX1. Scale bars, 50 μ m. Yellow arrowheads, FOXC2^{high} vLECs; blue arrowhead, FOXC2⁺ lymphangion LECs. (B) Kaplan-Meier curve of WT or Foxc2^{lecKO} mice survival. Tamoxifen administration at 3 weeks (*P < 0.001; n = 6 WT; n = 7 Foxc2^{lecKO}), 4 weeks (*P < 0.01; n = 3 WT; n = 8 Foxc2^{lecKO}), and 5 weeks of age (*P < 0.01; n = 12 WT; n = 13 Foxc2^{lecKO}). (C) Computed tomography (CT) image of WT and Foxc2^{lecKO} lungs. Yellow arrowhead, fluid accumulation. Scale bar, 0.25 cm. Tamoxifen administration at 5 weeks, analysis at 22 weeks.

Referee #2 (Comments on Novelty/Model System for Author):

This study addresses a timely and topical issue using a technically sound, rigorous single cell RNA sequencing approach. The data obtained are of high quality and in excellent agreement with previous studies on the topic based on imaging, gene-knock out and bulkRNA sequencing data. Concerning its medical impact, the study suffers from the lack of a clear molecular hypothesis and its experimental exploration probing the impact of lymph vessel function / valve rarefication after TAC on the progression of heart failure.

Potentially lymphatic transport, the time course of valve deterioration and inflammation levels and immune cell infiltration could be followed after Tac to substantiate the hypothetical chain of events proposed in Fig. 7

Referee #2 (Remarks for Author):

In their study entitled "Molecular determinants of cardiac lymphatic dysfunction in a chronic pressure-overload model" Heron et al. investigated changes in cardiac lymphatic endothelial cells (LECs) after transaortic constriction (TAC) with the aim to uncover disease-specific transcriptional reprogramming and thereby identify potential new targets to restore lymphatic drainage in CVD. The manuscript is largely based on single cell RNA sequencing technology of lymphatic enriched endothelial cells isolated by FACS. Unsupervised clustering of an initial population of ECs obtained from sham-treated Balb/c mice resulted in the identification arterial, venous and lymphatic populations in good agreement with a large number of available studies, demonstrating the validity of the approach taken in this study. Within the LEC population subclustering identified capillary, pre-collecting and valve LECs.

A first experimental group comprised cardiac ECs from 10 TAC-operated Balb/c mice at the stage of chronic heart failure. These cells clustered similar to sham-operated controls, displaying largely comparable cell cycles stages, which the authors attributed to the cessation of angiogenic / lymphangiogenic growth during chronic heart failure, which had been previously described by them. Less than 10% of expressed genes showed altered expression post-TAC with a slight preference of downregulated genes in LECs. Upregulated gene included LYVE1 and CCL21, potentially involved in immune cell recruitment and SERINC1 potentially increasing S1P levels. Downregulated were tight junction proteins CLDN5 and ZO1 and transcription factors including PROX1, NFAT5 and NFATC1. Of interest due to several recent publications, RELN expression was not increased on a per cell basis but expressed in more cells. Pathway enrichment analysis identified focal adhesion and fluid shear stress responses as down regulated. In BECs, lipid uptake and cytosolic trafficking pathways as well as ECM regulation were upregulated in particular, which was also in agreement with previous descriptions.

In summary, no uniquely new cell population was arising after TAC, which was in agreement with a previous study where the authors had demonstrated that TAC resulted in a massive expansion of cardiac lymphatic capillaries, which here was reflected in a relative expansion of the LEC1 cluster. At the same time the number of LECs in the valve cluster 3 was reduced, resulting in a strong rarefication of LEC3 cells. Quantitative analysis revealed a reduction in the number of valves per mouse by about 50% while their distance remained unchanged, which is somewhat counterintuitive. Surprisingly the remaining LEC3 cluster displayed no difference in expression of the valve-inducing transcription factors.

Based on a previous study, in which the authors had shown less severe TAC-induced cardiomyopathy in C57BL6/J female mice, they repeated the transcriptional analysis in the C57/BL mouse strain. Unsupervised clustering of LECs from healthy C57/BL mice resulted in clusters comparable to Balb/c LECs. TAC operation did not result in significant gene expression changes and a rarefication of valves was not observed. Again, the stronger lymphangiogenic properties of Balb/c versus C57/BL6 mice have been described in several studies before.

To obtain mechanistic insights the authors turned to human LECs. Based on a partial overlap of the changes in gene expression after TAC and IL1 β treatment they speculate that the observed TAC-induced changes in LECs could at least partly result from exposure to inflammatory conditions.

Major Comments:

The study uses a technically sound state of the art approach to investigate transcriptional changes in cardiac LECs after TAC. The topic is timely as a number of investigations over the last years revealed an unexpected contribution of lymphatic vessels to cardiac development and disease progression. The results presented remain in large parts confirmatory. Admittedly the study confirms on the single cell transcriptional level findings, including studies by the authors, in cardiac LECs subpopulations after TAC that have not been subjected to a targeted analysis previously. However, this largely descriptive study could neither uncover alterations in the cardiac LEC populations that provided potential new targets to restore lymphatic function in CVD nor open novel insights into the molecular mechanisms underlying the LEC insufficiency in CVD. Inclusion of different time points after TAC induction might have provided such additional data.

Overall, the single cell transcriptomic analysis described here is in excellent agreement with previous bulk RNAseq and imaging-based data. It is however arguable that the authors put more emphasis on scRNA data compared to previously obtained imaging data, which are based on actual protein levels. (page 13, Discussion, line 4 from bottom: "Our molecular studies revealed that, different from our wholemount imaging of cardiac lymphatics (which suggested lower Podoplanin levels in capillaries as compared to precollectors), the gene expression levels of Pdpn were comparable between LEC subpopulations"). Due to the exclusive focus of this study on ECs, mechanistic context based on the action of inflammatory cytokines or important growth factors like VEGF-C and D remains speculative and therefore part of the discussion as the relevant cellular sources of these cytokines are not included in the dataset. Important points like the potential relevance of the upregulation of LYVE1 and CCL21 for increased immune cell transcytosis into LVs are not further addressed experimentally. Unfortunately, this study could not shed more light on the recently much-discussed function of Reelin as a relevant lymphangiocrine factor in cardiac disease, as it was not changed transcriptionally after TAC and could not be investigated in immunohistochemistry.

Minor comments:

The cardiac EC populations were FACS-sorted for capillary LECs (CD45-PECAM1+PDPN+LYVE1+) yet only 15% of the ECs clustered in the LEC1 population? The authors did not comment on the high percentage of cells in the FACS-sorted population that did not express the desired markers.

Unfortunately, the suppl. Videos appeared to be not available.

How was population LEC 2 defined as pre-collectors, how was this population distinguished from collector LECs?

The quality of Fig. 2F in particular the confocal image is poor and does not allow the assessment of LYVE1 localization (i.e. button vs. zipper junctions).

Fig.3F, it appears surprising that the expression change of Nfatc1 is significant if e.g. Reln and Mmm1 are not, can the authors indicate significance levels?

Fig. 4, the rarefication of valves is not convincingly obvious from these pictures. There seem to be more valves present in the preparation than the ones indicated by yellow arrows. On what basis were these valves specifically chosen and why? At first sight valve distance appears to be smaller in the TAC preparation. Why did the authors visualize valves by podocalyxin instead of the more common PROX1, or INTA9?

Why was a different type of graphic representation chosen for Fig. 4H compared to 4G?

Page 10, the first two paragraphs are partially redundant as these data had already been described previously.

Page 10, the last paragraph refers to Ccl21 and Akr4 with reference to Fig. 4h, which however contains no data on these genes.

Fig. 6 Claudin5 should show a junctional staining pattern, i.e. be localized in LECs either to buttons or zippers, similarly in BECs TJs provide a fine distinct outline of the cells. The quality in Fig6 C is insufficient to make statements on the cellular localization or distinguish the staining in LECs from BECs.

References: Heron 2023b, citation is incomplete, please provide volume and page or equivalent information.

Referee #3 (Comments on Novelty/Model System for Author):

NGS performed and analyzed according to current standard methodology.

Model is gold-standard for cardiology, and the use of 2 strains goes above the field standard

Medical impact is high but work is needed to draw some more general conclusions

Referee #3 (Remarks for Author):

This manuscript describes a scRNAseq analysis of cardiac lymphatics in the gold-standard non-ischemic heart failure model. The model is applied in two different strains, known to yield different severity outcomes. Lymphatic regulation of pathology is a topic of very high interest in cardiology and cardioimmunology. As far as I know, this is the first study attempting an scRNAseq analysis of cardiac lymphatics, thus yielding potentially high-interest results and a useful resource once the underlying data is published.

The procedures and methodologies used for the scRNAseq are standard, and appropriate, as is the identification of lymphatic

endothelial cells.

Indeed all my comments (see below) are limited on data description and interpretation.

It is commendable that the processed gene expression data are included in the supplementary materials. This study will have great value as a resource. Reducing the hurdles required to access and process the data by researchers wishing to work on them is likely to amplify the study's impact.

Comments:

Pg5 "Reclustering analyses allowed investigations of BEC subpopulations (see Table S3 for marker genes), revealing elevated *Cldn5* expression only in two clusters: BEC9 (Suppl. Fig. S3c), enriched for arteriolar markers (e.g. *Id1*, Notch-target gene *Hey1*), and BEC6, enriched for angiogenic markers (e.g. *Aplnr2*, Apelin receptor 2)."

The logic of what is done here and why it was decided to do it must be made more clear.

In the description of the results of Fig 3, the authors have selected to outline the findings with no concurrent speculative comments on the possible functional significance of the findings. Whilst this is formally correct, it would aid the readability of the manuscript if speculative (and clearly marked as such) comments were added in Results, after each section. The amount of detail emerging from a scRNAseq analysis (especially in a "novel" tissue) is overwhelming for most readers to be able to defer the discussion of the findings in the Discussion section.

The results shown in subsequent figures are described in a manner that includes more function-related comments, rendering them more readable.

Pg10 "finding an average interval distance of 200 μm in healthy mice (Fig. 4e). In the remaining valves in TAC-operated mice this valve distance was unaltered. "

Whilst the data in Fig4e is quite clear, the phrase above is somewhat misleading - please rephrase.

The *Ackr2* and *Mrc1* upregulation (Fig S7a,b) may be an attempt at immune deviation from a "classical" inflammatory response. It would be worth discussing more.

The utilization of a less severe form of TAC (by using another strain) is an innovative solution, and commendable. The lack of major differences in the female TAC vs sham - in the C57 version of the model, where disease severity is reduced- suggest that LEC presence is associated with disease severity. This, in turn, means that LEC are either a (component of a) driver of disease, or a negative feedback loop against disease; in both cases LEC would not appear changed in low-severity disease.

This message is quite novel and substantial, and of clear interest to cardiologists and immunologists beyond those interested specifically in Lymphatics - but it does not emerge easily in the current write-up. Indeed, whilst the manuscript is great in describing the detailed aspects of LEC biology in cardiopathic and healthy mice, it could substantially increase its impact in the cardiology and immunology community, by drawing some more general system-level conclusions, seeing the forest instead of the trees.

31/08/2025

We hereby submit a revised version of our manuscript **EMM-2025-21696-v2** entitled “Molecular determinants of cardiac lymphatic dysfunction in a chronic pressure-overload model” for consideration in **EMBO Mol Med**. We wish to deeply thank the Editors and the three Reviewers for their valuable input to help us improve our study. We have revised the manuscript in accordance with their suggestions and in line with the specific requirements of the journal. Please find below our point-by-point responses to the Reviewers’ questions and suggestions (*highlighted in blue*).

Referee #1:

The authors have identified 2 strains of mice in which the cardiac lymphatic vessels respond distinctly to pressure overload. I think this is a powerful discovery and can be used efficiently to identify the mechanisms of lymphatic vessel growth.

We sincerely thank the Reviewer for appreciating our study.

The Brakenhielm lab has been the pioneers in investigating the relationship between lymphatic vessels and cardiac health. In 2016 they showed that the induction of lymphangiogenesis with VEGF-C reduces myocardial edema and fibrosis following myocardial infarction (MI) in rats (Henri et al., 2016). In 2020 they paradoxically showed that the inhibition of lymphangiogenesis with soluble VEGFR3 improves cardiac function following MI (Houssari et al., 2020). In 2023 they showed that pressure overload promotes cardiac lymphangiogenesis in mice in a strain- and sex-dependent manner (Heron et al., Cardiovascular Research. 2023). Specifically, pressure overload induced lymphatic vessel growth only in female BALB/c mice resulting in reduced inflammation, but did not affect fibrosis or heart failure. Lymphatic vessel density did not increase in C57BL6 mice although soluble VEGFR3 reduced lymphatic vessel density, increased inflammation and fibrosis, increased left ventricle dilatation and aggravated heart failure. However, there was no change in myocardial edema. Furthermore, they showed that this lymphatic growth in BALB/c mice is at least partially mediated by IL-1b signaling pathway (Heron et al., Biorxiv. 2023). Inhibition of IL-1B only delayed left ventricle dilation and did not prevent heart failure in these mice. In this follow up study Heron et al use scRNA-seq to characterize the mechanisms that regulate lymphangiogenesis in BALB/c, but not C57BL6 mice. This work is well-done and potentially important. However, due to the above-mentioned, sometimes conflicting and confusing data, a better explanation of their models is needed. A few additional experiments are also needed to understand the cell-autonomous and non-cell autonomous mechanisms that regulate lymphatic vessel growth.

We thank the Reviewer for this accurate summary of our previous work, and for the kind remarks and most helpful suggestions.

Major concerns and comments

1. As described above, it is not entirely clear if lymphatic vessel growth is preventing or aggravating MI, edema, fibrosis and heart failure. These phenotypes could be species (rat, mouse, human), strain (C57BL6, BALB/c), disease (MI, pressure-overload, ischemia reperfusion, fibrosis, inflammation) and sex specific. A figure or a table at the very beginning of the manuscript summarizing these various results and the working hypothesis of this manuscript based on these results will be very helpful. If possible, I also suggest incorporating the works of other labs that have also presented conflicting results (Oliver, Kahn, Caron etc).

Thank you for this valuable comment. We fully agree that the current literature includes some seemingly contradictory findings, notably in experimental post-MI models. However, the general consensus in the field is that lymphatic vessel growth represents a beneficial functional adaptation stimulated by cardiac inflammation and edema, with the ultimate aim to restore tissue homeostasis. The conflicting findings in the literature are essentially limited to observations in transgenic animals where manipulation of cardiac lymphatics (suppression of lymphatic growth post-MI) has not always yielded the expected results. We have in several previous review articles and editorials (Brakenhielm et al ATVB 2024; Brakenhielm et al JCI

2021) discussed in detail these opposing findings. We have added a sentence to the introduction. Concerning the role of lymphatics in pressure-overload, our recent work (Heron et al Cardiovasc Res 2023) surprisingly showed that lymphatic expansion was not enough to resolve myocardial edema and inflammation, and thus prevent HF, in BALB/c mice post-TAC. In parallel, our data in C57 revealed less inflammation, coupled to less potent lymphangiogenesis, and persistent myocardial edema post-TAC, indicating insufficient lymphatic clearance also in this model. Similarly, other groups have demonstrated lymphatic transport dysfunction following pressure-overload in C57 mice (Bizou et al Sci Rep 2021; Song et al Elife 2020). In the current study, we hypothesized that newly formed lymphatics, notably in BALB/c mice, may be dysfunctional due to molecular and/or structural changes induced by chronic cardiac inflammation. We thus set out to identify the impact on cardiac lymphatics during pressure-overload. One of the key novelties of our study is the demonstration that beyond lymphatic density, which so far has remained the key metric of lymphatic remodelling in CVDs, the *molecular fingerprints* of lymphatics may change considerably in pathology, likely contributing to lymphatic dysfunction. Thus, we hope with the current study to shift the focus in the field, currently mainly based on analyses of lymphatic densities, to also include assessment of lymphatic molecular profiles relevant for lymphatic functions.

In agreement with the Reviewer's suggestions to better acknowledge some of the outstanding questions in the field, and to more clearly place our hypothesis in the present study, we have expanded the introduction and now write:

Line 71 *"While suppression of lymphangiogenesis in CVD models has not always been associated with aggravated cardiac dysfunction (Brakenhielm et al, 2024), promisingly, therapeutic lymphangiogenesis suffices to reduce cardiac dysfunction in models of myocardial infarction (MI) and chronic pressure-overload (Henri et al, 2016; Yang et al, 2014; Vieira et al, 2018; Trincot et al, 2019; Liu et al, 2020a; Houssari et al, 2020; Song et al, 2020; Ginton et al, 2022)."*

Line 105 *"We hypothesized that cardiac lymphatic dysfunction in CVDs may be caused not only by insufficient lymphangiogenesis, but also by structural and molecular changes in cardiac LECs induced by cardiac inflammation. Here, we set out to investigate, using scRNAseq and 3D imaging, the cellular molecular shifts that accompany cardiac lymphangiogenesis in the setting of inflammation induced by chronic pressure-overload in BALB/c mice, with the ultimate aim to uncover potential new targets, beyond lymphangiogenic growth factor therapy, to restore lymphatic drainage in CVDs. We further compared the profiles of cardiac LECs between BALB/c and C57BL6/J (C57) strains in health and disease. Of note, the TAC model in the C57 strain is associated with less myocardial inflammation and cardiac dysfunction, as compared to BALB/c (Heron et al, 2023a), although lymphatic transport dysfunction also has been noted in C57 mice following chronic pressure-overload (Bizou et al, 2021; Song et al, 2020). Finally, we compared our datasets to published reports on lymphatic molecular profiles in other organs and disease settings to identify potential targets specific to cardiac lymphatics. In parallel, we investigated molecular changes occurring in cardiac blood endothelial cells (BEC) in pressure-overload, as a comparison to highlight target genes selectively altered in cardiac LECs."*

2. The authors state that they performed scRNA-seq after enriching for Lyve1+ Pdpn+ LECs. If so, there should be no BECs or vBECs in the sample. I think they have enriched for all ECs. This mistake must be rectified.

We thank the Reviewer for spotting this mistake. Indeed, we enriched cardiac samples for BOTH blood endothelial cells (sorted as CD45⁻ CD31⁺ Lyve1⁻) and LECs. We have in the revised manuscript stated this more clearly:

Line 123: “To define the transcriptome of healthy murine cardiac vascular endothelial cells (ECs), we sorted by FACS cardiac CD45⁻/CD31⁺ cells from 20 sham-operated adult female BALB/c mice, further enriching the samples for Lyve1 and Podoplanin (Pdpn)-expressing LECs and for lymphatic-marker-negative BECs prior to scRNAseq”

Line 677: “Cardiac single cell suspensions were sorted by FACS (ARIA II, BD Biosciences). BECs were defined as live CD45⁻/CD31⁺/Lyve1⁻ cells, whereas LECs were selected as CD45⁻/CD31⁺/Lyve1⁺/Pdpn⁺ cells. A 1:3 mix of cardiac LECs and BECs was included in each sample”

3. *Vegf-c*, *Vegf-d* (Figure 2, Heron et al., 2023) and presumably *IL-1b* (Heron et al., Biorxiv 2023) are necessary for promoting lymphangiogenesis. However, we currently do not know where these signals are coming from. Therefore, it is also important to perform scRNA-seq by using the whole heart. Additionally, it is not clear which cells are inducing fibrosis by secreting collagen (Figure 6, Heron et al., 2023). scRNA-seq using entire heart will help resolve these questions.

We thank the Reviewer for these highly pertinent suggestions. We previously showed that *Vegf-c*, rather than *Vegf-d*, is a key driver of lymphangiogenesis in Balb/c mice. We also previously demonstrated, using immunohistochemistry, that cardiac macrophages are a rich source of *Vegf-c* in both Balb/c and C57 strains of mice (Suppl Fig S7, Heron et al 2023a). Similarly, a scRNAseq study by Rizzo G et al (*Cardiovasc Res.* 2023;119(3):772-785) revealed that resident MHCII⁺ or Trem2⁺ macrophages are an important source of *Vegfc* in mice. Another recent line of evidence indicating cardiac macrophages as the main source of *Vegf-c* in mice was provided by Glinton KE et al (*JCI* 2022;132:e140685), demonstrating that selective deletion of *Vegfc* in macrophages sufficed to reduce cardiac lymphangiogenesis post-MI. Additionally, in a yet unpublished bioRxiv study we showed that IL1 β stimulation of macrophages (*in vitro*) enhanced both their *Vegf-c* production and processing capacity, and conversely that antibody-mediated blockage of IL1 β *in vivo* sufficed to reduce cardiac *Vegf-c* levels, notably the mature, cleaved, and fully-active isoform. Based on these data, we estimate that the main source of cardiac *Vegf-c* driving potent lymphangiogenesis in Balb/c mice post-TAC is cardiac macrophages, likely exposed in a paracrine manner to IL1 β (see *beneath*).

Concerning *Vegf-d*, which also may be involved in driving cardiac lymphangiogenesis in C57 mice, our previous work (**Suppl Fig S7**, Heron et al 2023a) indicated some expression in cardiac blood vessels. In agreement, our scRNAseq analysis in the current study confirm expression of *Vegfd* in BECs, albeit at low levels (see **Figure A** beneath). Moreover, recent studies suggest cardiac *fibroblasts* as a main source of *Vegfd* (Gong H et al. *Theranostics*. 2024, PMID: 38505621). In our scRNAseq dataset, we inadvertently captured during our EC sorting a small population ($N=14-38$ cells /group) of *Fbln1*⁺ *Col1a2*⁺ fibroblasts and of *Acta2*⁺ stromal cells ($N=8-61$ cells /group) (**Suppl. Fig. S1**). Our very preliminary data confirm these cells as potential sources of both *Vegfc* and *Vegfd* in the heart (**Figure A**). These non-EC populations are however too small in our dataset, and not reproducibly captured between strains and groups, which is why we prefer to not present them in the current EC focused manuscript (except for showing their existence in **Suppl. Fig. S1a**).

Figure A – Lymphangiogenic growth factor expression in cardiac vascular endothelial cells, Fibroblasts and Stromal cells. *Left* : Very few non-EC transcriptomes were found in the scRNAseq data from either BALB/c or C57 strains. ECs in *blue*, Fibroblasts in *yellow*, stromal (mural) cell clusters in *red*. *Right* : Violin plots of growth factor genes (Log2 normalized expression levels) indicating that cardiac fibroblasts and stromal cells may be potential sources of *Vegfc* in C57 and BALB/c mice, respectively, while *Vegfd* is weakly expressed in BECs in BALB/c, and by cardiac fibroblasts in C57 mice. As a control gene, *Tgfb2* is expressed by both fibroblasts and stromal cells in our data set.

As for the source of cardiac IL1 β , a scRNAseq study by Rizzo G et al (*Cardiovasc Res.* 2023;119(3):772-785) indicate that *neutrophils* express high levels of *Il1b* in mice post-MI, while a recent study by Amrute JM et al (*Nature* 2024;635(8038):423-433) used spatial transcriptomics to identify *Ccr2+* cardiac-infiltrating *macrophages* as the main source of *Il1b* in mouse hearts after MI but also after pressure-overload. We recently investigated this question in our preprint article (Heron et al 2023b Suppl Fig 2), where we found, using immunohistochemistry, strong IL1 β expression in areas enriched with cardiac *macrophages* in the setting of pressure-overload. Of note, data is emerging that **it is not the same subset of macrophages that produce IL1 β as those that produce Vegf-c in the heart**. However, as this topic is not at the centre of our current study, we prefer to simply refer to ‘macrophages’ as a key source of cardiac Vegf-c and IL1 β during pressure-overload.

As for the regulation of cardiac collagen content (with reference to Fig 6 in *Heron et al 2023a*), we ofcourse expect cardiac fibroblasts to be the main producers in both physiology and pathology. In our previous study, and the follow-up preprint article *Heron et al 2023b*, we intriguingly found that potent lymphangiogenesis was associated with reduced perivascular collagen deposits, and we speculated that these vascular niche lymphatics may influence perivascular edema and/or inflammation, thus limiting activation of cardiac fibroblasts. In the current study, our finding that cardiac LECs post-TAC upregulate some ECM components (*Eln*, *Fbln2*, *Lox12*) is only an indirect indication that lymphatics potentially also may influence *directly* local remodelling of the extracellular matrix. This remains a totally open question, as the current consensus is that lymphatics prevent fibrosis by draining activated immune cells and excess fluids to restore homeostasis.

To better incorporate these different aspects, we now write in the revised manuscript:

Line 81: “... *inflammatory mediators, including IL1 β and IL6, produced in the heart notably by macrophages during pressure-overload (Heron et al, 2023a, Amrute 2024)*”

Line 83: “Previous work has identified cardiac macrophages as a key source of Vegf-c (Heron et al, 2023a; Glinton et al, 2022), and we recently demonstrated that IL1 β promotes cardiac lymphangiogenesis post-TAC by stimulating macrophage production of Vegf-c (Heron et al, 2023b).”

Line 280: “It remains to be determined whether the activated lymphatics post-TAC may contribute to local remodelling of the extracellular matrix during pressure-overload”

4. IL-1B is likely responsible for the cell-junction defects in BALB/c mice. However, I am not sure how to interpret the downregulation of Prox1, a critical molecule for lymphangiogenesis.

We thank the Reviewer for this interesting question. One probable reason for the observed reduction of *Prox1* levels in the global LEC population post-TAC is the major rarefaction in this group of highly *Prox1*-expressing valvular LEC3 cells. Indeed, when comparing expression levels *within* the three subclusters, we found that *Prox1* levels were comparable between sham and TAC groups for LEC1, LEC2 and LEC3 subpopulations (Fig. 4g). In further support, the reduction in *Prox1* expression levels in the global LEC cluster was accompanied by maintenance of the % of cardiac LECs expressing *Prox1* post-TAC (see revised Suppl Fig. S4a, b). Concurrently, the expression levels of several other key LEC transcription factors (*Nfat5*, *Nfatc1*, *Klf4*) were also reduced (Fig. 3e), but their frequencies of expression were similarly maintained. In contrast, we observed that the frequency of expression of the transcription factors *Nr2f2* and *Foxc2* was strikingly increased post-TAC (Suppl Fig. S4a, b). Currently, little is known about the regulation and impact of transcription factor co-operation in LECs, but it is likely that a reduction of some factors may be countered by an increase in others.

To investigate another potential cause of reduced *Prox1* expression in the global LEC population, we looked at *yap/tafazzin/tead* gene expression levels in cardiac LECs, as well as of the Yap/Taz target genes *Ankrd* and *Ccn2*, as the Hippo pathway is known to reduce *Prox1* expression (PMID: 30582452). However, we found that these actors were not significantly altered post-TAC (see Figure B).

Figure B – Hippo pathway regulators of Prox1 expression. Normalized gene expression levels in global LEC population shown as violin plots for sham or TAC groups in BALB/c mice (Log2 expression levels).

Another main regulatory mechanism of *Prox1* expression in LECs is fatty acid (FA) metabolism-dependent histone acetylation (PMID: 35589749). In our current study, we did not explore epigenetic marks that could underlie altered gene expression, nor did we directly analyse LEC FA metabolism. It is however likely that TAC leads to compensatory increased

FA uptake in the heart (PMID: 30104639), and our finding of increased *Cd36* in BECs (Suppl. Fig. S2a) supports this idea.

Figure C Examples of gene expression of Prox1 target genes in global LEC cluster in BALB/c mice

Finally, our data demonstrating that the expression of key Prox1-regulated genes, including *Flt4*, *Cdkn1c*, *Lyve1*, as well as FA transporters *Cd36* and *Cpt1a*, were either unaltered or increased post-TAC (see Figure C above, and new panel Fig. 3f), indicate that the reduction of *Prox1* in LECs post-TAC did not have major impact on LEC identity.

To better address these point we now write in the revised manuscript:

Line 269: “We further found that the expression levels of several lymphatic transcription factors, including *Prox1*, *Nfat5*, *Nfatc1*, and *Klf4* (Kruppel-Like factor-4), were reduced (Fig. 3d-f, Table S5). However, their frequency of expression, as well that of other transcription factors, notably *Nr2f2*, was maintained or increased in cardiac LECs post-TAC (Suppl. Fig. S4b). These changes indicate a shift in the transcription factor landscape in LECs, which may influence lymphatic activity and function following chronic pressure-overload. Of note, key *Prox1* target genes, including *Flt4* and *Cdkn1c*, were not reduced, while *Lyve1*, as mentioned above, was increased (Fig 3f), indicating that *Prox1* activity is still sufficient to drive expression of these genes.”

5. In figure 4G the sham and TAC induced changes in the 3 types of LECs in BALB/c mice are presented nicely. A similar comparison must be done for LECs in C57BL6 mice in figure 5 (not just for junctional molecules).

We thank the Reviewer for the kind comment. We now have updated the presentation of LEC subpopulations in C57 in revised Fig. 5g and new Suppl. Fig. S8 to match Fig. 4g. We also point out in the figure legends that none of the lymphatic genes shown in Fig. 5g were significantly altered post-TAC in C57, whereas many were significantly changed in BALB/c.

6. Please compare the LECs of BALB/c and C57BL6 mice under sham and TAC conditions.

Many thanks for this excellent suggestion. We have now generated a new supplemental figure (Fig. S8) and a supplemental data table (new table S11) highlighting the differences, and similarities, between C57 and BALB/c cardiac LECs. We now write in the revised manuscript:

Line 420: “Comparing LEC expression profiles between healthy BALB/c and C57 mice, we found that most lymphatic marker genes were expressed at comparable levels (Suppl. Fig. S8). However, 132 genes were differentially expressed in cardiac LECs between strains (Table S11). For example, BALB/c LECs expressed significantly higher levels of *Ccl21a*, *Aqp1*, *Nfat5*,

and *Klf4*, while *Cldn5*, *Selenof*, *Mmrn1*, *Ramp2*, and *H2-K1* were more expressed in C57 LECs (Suppl. Fig. S8).”

Line 437; “Within the global cardiac LEC cluster, we only detected 3 DEGs post-TAC in C57 mice, and none of them involved lymphatic chemokines, immune adhesion molecules, or barrier components (Fig. 5h, Table S12). Indeed, cardiac LECs post-TAC in C57 mice had very similar molecular profile as healthy LECs, in strong support of a more physiological-like state of cardiac lymphatics during pressure-overload in C57, as compared to in failing BALB/c hearts (Suppl. Fig. S8).”

Minor concern

1. The figure legends are very hard to comprehend. For example, Examples of mean expression levels (Log2 normalized read counts) of altered genes [heatmap, f], [violin plot, g] in cardiac LEC subpopulations from healthy and post-TAC hearts. Significant genes indicated (#) for each cluster in g.

Please try to make it easier for the readers to understand the legends. Describe the various panels in separate sentences. The panels in the figures are with capital letters, but figure legends are with small letters and placed at the end of the sentences. Something like the following will be good with the panel's identity mentioned at the beginning.

Fig. 1 *Foxc2* loss disrupts transport function of adult lymphatics. (A) FOXC2 in adult mesenteric LVs. Green, FOXC2; red, PROX1. Scale bars, 50 μ m. Yellow arrowheads, FOXC2^{high} vLECs; blue arrowhead, FOXC2⁺ lymphangion LECs. (B) Kaplan-Meier curve of WT or *Foxc2*lecKO mice survival. Tamoxifen administration at 3 weeks (**P* < 0.001; *n* = 6 WT; *n* = 7 *Foxc2*lecKO), 4 weeks (**P* < 0.01; *n* = 3 WT; *n* = 8 *Foxc2*lecKO), and 5 weeks of age (**P* < 0.01; *n* = 12 WT; *n* = 13 *Foxc2*lecKO). (C) Computed tomography (CT) image of WT and *Foxc2*lecKO lungs. Yellow arrowhead, fluid accumulation. Scale bar, 0.25 cm. Tamoxifen administration at 5 weeks, analysis at 22 weeks.

Thank you for this suggestion. We have modified figure legends accordingly. We have also during the revision also corrected some of the panels to limit visualization of each marker or differentially-expressed gene to either heatmap or violin plot, but not both, as was erroneously done for some key genes (eg *Ccl21a*, *Prox1*, *Lyve1*) in the first version of the manuscript. This has led to slightly modified panels in Fig 1d, Fig 2c, Fig 3e-f, Fig. 4f in the revised manuscript.

Referee #2

This study addresses a timely and topical issue using a technically sound, rigorous single cell RNA sequencing approach. The data obtained are of high quality and in excellent agreement with previous studies on the topic based on imaging, gene-knock out and bulkRNA sequencing data. Concerning its medical impact, the study suffers from the lack of a clear molecular hypothesis and its experimental exploration probing the impact of lymph vessel function / valve rarefication after TAC on the progression of heart failure. Potentially lymphatic transport, the time course of valve deterioration and inflammation levels and immune cell infiltration could be followed after Tac to substantiate the hypothetical chain of events proposed in Fig. 7

We deeply thank the Reviewer for appreciating the soundness of our technical approach. We agree that additional work is needed to confirm the working hypothesis proposed in Fig. 7. Based on our previous studies, we expect that the valve phenotype will only be apparent following expansion of cardiac lymphatics post-TAC. This occurs between weeks 5-8 in our model. Indeed, at week 4 post-TAC, lymphatics in BALB/c appear rather normal and cardiac IL1 β expression is still weak. We agree that investigating lymphatic expression of Cldn5 and the presence of valves in lymphatic capillaries at earlier timepoints (5-6 weeks) may reveal whether the molecular changes develop early or late post-TAC. Our preliminary data indicate that Cldn5 expression is not altered in cardiac lymphatics at 4 weeks post-TAC (see Figure D). In addition, we have promising, but still preliminary, scRNAseq and IHC data (see Figure E) in cardiac LECs from anti-IL1 β -treated Balb/c TAC mice, demonstrating that blockade of this cytokine suffices to restore Cldn5 expression at 8 weeks post-TAC. However, as these data would need further validation, including scRNAseq analyses performed at additional timepoints, we prefer not to include mention of these findings in the current manuscript.

Figure D – Lymphatic Cldn5 levels at 4 weeks post-TAC

Examples of immunohistochemical analyses of Cldn5 expression (*red*) in Lyve1+ (*grey*) lymphatic vessels at 4 weeks in sham-operated or TAC BALB/c mice reveal that Cldn5 levels appear unaltered at this early time point prior to development of heart failure.

Figure for reviewers removed

In their study entitled "Molecular determinants of cardiac lymphatic dysfunction in a chronic pressure-overload model" Heron et al. investigated changes in cardiac lymphatic endothelial cells (LECs) after transaortic constriction (TAC) with the aim to uncover disease-specific transcriptional reprogramming and thereby identify potential new targets to restore lymphatic drainage in CVD. The manuscript is largely based on single cell RNA sequencing technology of lymphatic enriched endothelial cells isolated by FACS. Unsupervised clustering of an initial population of ECs obtained from sham-treated Balb/c mice resulted in the identification arterial, venous and lymphatic populations in good agreement with a large number of available studies, demonstrating the validity of the approach taken in this study. Within the LEC population subclustering identified capillary, pre-collecting and valve LECs.

We thank the Reviewer for well appreciating the aims of our study and the validity of our results. We would like to emphasize that our manuscript is the very first scRNAseq study focused on cardiac LECs, which represent a rare cardiac cell population that has not been sufficiently captured in previous work by other groups investigating vascular endothelial cell transcriptomes in the heart.

A first experimental group comprised cardiac ECs from 10 TAC-operated Balb/c mice at the stage of chronic heart failure. These cells clustered similar to sham-operated controls, displaying largely comparable cell cycles stages, which the authors attributed to the cessation of angiogenic / lymphangiogenic growth during chronic heart failure, which had been previously described by them. Less than 10% of expressed genes showed altered expression post-TAC with a slight preference of downregulated genes in LECs. Upregulated gene included LYVE1 and CCL21, potentially involved in immune cell recruitment and SERINC1 potentially increasing S1P levels. Downregulated were tight junction proteins CLDN5 and ZO1 and transcription factors including PROX1, NFAT5 and NFATC1. Of interest due to several recent publications, RELN expression was not increased on a per cell basis but expressed in more cells. Pathway enrichment analysis identified focal adhesion and fluid shear stress responses as down regulated. In BECs, lipid uptake and cytosolic trafficking pathways as well as ECM regulation were upregulated in particular, which was also in agreement with previous descriptions. In summary, no uniquely new cell population was arising after TAC, which was in agreement with a previous study where the authors had demonstrated that TAC resulted in a massive expansion of cardiac lymphatic capillaries, which here was reflected in a relative expansion of the LEC1 cluster. At the same time the number of LECs in the valve cluster 3 was reduced, resulting in a strong rarefaction of LEC3 cells. Quantitative analysis revealed a reduction in the number of valves per mouse by about 50% while their distance remained unchanged, which is somewhat counterintuitive. Surprisingly the remaining LEC3 cluster displayed no difference in expression of the valve-inducing transcription factors.

We thank the Reviewer for this excellent overview of our results. We have in the revised version rephrased the interpretation of valve distances to better explain how we view these data. We saw in our images entire segments of the lymphatic vasculature devoid of valves (see **Fig. 4d** in the revised manuscript), and consequently intervalve distances could not be assessed in these areas. Thus, the measurements of valve distances were made principally in precollector segments, and a few remaining capillary segments endowed with valves post-TAC. If there had indeed been a loss of valves, due to lymphatic 'dedifferentiation' post-TAC, intervalve distances would have increased. This was clearly not the case. Thus, we estimate that the "reduction" of capillary lymphatic valves relates to failure to generate valves in the newly expanded lymphatics. Thus, it is not surprising that the remaining LEC3 cells show "normal" molecular profiles, as these cells potentially represent valves formed *prior* to the TAC procedure. To better capture this interpretation, we now write:

Line 357: *"In the remaining valves in TAC-operated mice this valve distance was unaltered, suggesting that insufficient valvulogenesis, not valve drop-out, may be involved. Taken together, our data indicate that the newly expanded capillary lymphatic network developing post-TAC may lack signals to drive lymphatic valvulogenesis, and this structural deficit is very likely to restrict lymphatic drainage capacity despite elevated lymphatic density in the heart."*

Based on a previous study, in which the authors had shown less severe TAC-induced cardiomyopathy in C57BL6/J female mice, they repeated the transcriptional analysis in the C57/BL mouse strain. Unsupervised clustering of LECs from healthy C57/BL mice resulted in clusters comparable to Balb/c LECs. TAC operation did not result in significant gene expression changes and a rarefication of valves was not observed. Again, the stronger lymphangiogenic properties of Balb/c versus C57/BL6 mice have been described in several studies before. To obtain mechanistic insights the authors turned to human LECs. Based on a partial overlap of the changes in gene expression after TAC and IL1 β treatment they speculate that the observed TAC-induced changes in LECs could at least partly result from exposure to inflammatory conditions.

Major Comments:

The study uses a technically sound state of the art approach to investigate transcriptional changes in cardiac LECs after TAC. The topic is timely as a number of investigations over the last years revealed an unexpected contribution of lymphatic vessels to cardiac development and disease progression. The results presented remain in large parts confirmatory. Admittedly the study confirms on the single cell transcriptional level findings, including studies by the authors, in cardiac LECs subpopulations after TAC that have not been subjected to a targeted analysis previously. However, this largely descriptive study could neither uncover alterations in the cardiac LEC populations that provided potential new targets to restore lymphatic function in CVD nor open novel insights into the molecular mechanisms underlying the LEC insufficiency in CVD. Inclusion of different time points after TAC induction might have provided such additional data.

We thank the Reviewer for appreciating the soundness of our approach. We respectfully disagree that our study did not uncover new targets. Indeed, we identified two major previously unknown structural and molecular changes in remodelled cardiac lymphatics specific to BALB/c mice: 1) loss of lymphatic valves in lymphatic capillaries; and 2) loss of lymphatic Cldn-5 in capillaries, precollectors, and valves. Moreover, we identified IL1 β as a potent regulator of human LEC expression profiles, mirroring many of the changes observed *in vivo* in mouse hearts during pressure-overload. Currently, we do not know whether blocking IL1 β during pressure-overload would suffice to restore 'physiological' molecular profiles in cardiac LECs. What we do know from our parallel study investigating anti-IL1 β treatment (*Heron et al* 2023b), is that cardiac Vegf-c levels and lymphatic density is reduced. Our preliminary data from ongoing work promisingly suggests that blockage of IL1 β may restore Cldn5 expression in cardiac lymphatics. Whether limitation of lymphatic capillary expansion additionally would improve capillary valvulogenesis remains to be determined.

While there is, as of yet, no therapy available to stimulate lymphatic valvulogenesis, there are some additional potential therapies that could be considered to restore lymphatic Cldn5 expression. For example, angiopoietin-1 signalling has been proposed to counteract inflammation-induced reduction of Cldn5 in other organs (Kajiya K et al. *Am J Pathol.* 2012;180(3):1273-1282. PMID: 22200616). This could be tested in future studies. We plan to investigate whether restoring Cldn5 expression may improve lymphatic function, and thus clearance of inflammation and edema, post-TAC in BALB/c mice.

Another major finding in our current study is the lack of molecular changes in cardiac lymphatics post-TAC in C57 mice. We expect that these striking strain-dependent differences may allow us in the future to better dissect what part of the lymphatic transport dysfunction in CVDs is *intrinsic* to altered lymphatics (and can be rescued by treatments that restore Cldn5 expression and/or capillary valves), and what part is *extrinsic*, *i.e.* due to cardiac contractile dysfunction. Such studies may yield clinically-relevant information on how to enhance lymphatic drainage to resolve myocardial edema and inflammation through modalities other than therapeutic lymphangiogenesis.

To better emphasize the new potential targets identified in our study, and the clinical-perspectives of our work, we now write in the revised manuscript:

Line 535: *“This indicates that other inflammatory mediators may underly loss of Cldn5 in cardiac lymphatics during HF. Interestingly, it has been proposed that angiotensin-1 may restore lymphatic Cldn5 expression (Kajiya et al, 2012)”*

Line 637: *“Concerning new molecular targets to enhance lymphatic function in CVDs, our current study revealed among the most notable changes a radical loss of Cldn5 and of lymphatic valves in cardiac lymphatics post-TAC.”*

Line 652: *“In conclusion, our study revealed severe molecular and structural changes in cardiac lymphatics during pressure-overload-induced HF, with in contrast very few alterations noted during pressure-overload-induced pathological hypertrophy. The striking strain-dependent differences uncovered in our study may be used to dissect what part of the lymphatic transport dysfunction in CVDs is intrinsic to altered lymphatics, and what part is extrinsic, i.e. due to cardiac contractile dysfunction. We expect that this may yield clinically-relevant information on how to improve lymphatic drainage in CVDs to resolve myocardial edema and inflammation through modalities beyond therapeutic lymphangiogenesis.”*

Overall, the single cell transcriptomic analysis described here is in excellent agreement with previous bulk RNAseq and imaging-based data. It is however arguable that the authors put more emphasis on scRNA data compared to previously obtained imaging data, which are based on actual protein levels. (page 13, Discussion, line 4 from bottom: “Our molecular studies revealed that, different from our wholemount imaging of cardiac lymphatics (which suggested lower Podoplanin levels in capillaries as compared to precollectors), the gene expression levels of Pdpn were comparable between LEC subpopulations”).

We apologize for this confusing statement. We meant to say that upon reinspection of our whole mount images (performed before molecular analyses) it became apparent that they were indeed in excellent agreement with our scRNAseq data. The phrase in the discussion was to acknowledge that it was an error of perception that led us to conclude (in previous published works) that cardiac precollectors are characterized by elevated Pdpn, when in fact it was higher Lyve1 in capillaries that ‘masked’ the Pdpn signal in these segments. To avoid confusion, we removed the specific phrase in the discussion, and now write in the results section:

Line 206: *“We confirmed, using confocal and lightsheet imaging, that precollector segments displayed similar Pdpn expression, but lower levels of Lyve1 and Ccl21a, as compared to capillary lymphatics (Fig. 2c-f, Table S4).”*

Due to the exclusive focus of this study on ECs, mechanistic context based on the action of inflammatory cytokines or important growth factors like VEGF-C and D remains speculative and therefore part of the discussion as the relevant cellular sources of these cytokines are not included in the dataset.

We thank the Reviewer for this important comment. While we agree that inclusion of additional cell populations, such as CD45+ immune cells, in our scRNAseq analyses could have opened for more detailed discussion of the molecular cross-talk between lymphatics and immune cells, we would like to point out that some of the key points raised by the Reviewer have already been previously investigated. For example, we demonstrated by immunohistochemistry in BALB/c hearts after pressure-overload that cardiac Vegf-c is mostly produced by cardiac macrophages (Heron et al 2023a), while other groups have demonstrated by immune cell-focused scRNAseq (Rizzo G et al Cardiovasc Res. 2023;119(3):772-785) or macrophage-

specific *Vegfc* deletion (Glinton KE et al JCI 2022) that resident and/or efferocytotic macrophages are the main sources in the heart of Vegf-c.

As for the origin of the cytokines, we have previously demonstrated by immunohistochemistry in BALB/c hearts after pressure-overload that IL-1 β is enriched in areas infiltrated by cardiac macrophages (Heron et al 2023b), while recent spatial transcriptomic studies further identified the main source of *IL1b* in the heart as CCR2+ monocyte-derived proinflammatory macrophages (Amrute JM et al Nature 2024;635(8038):423-433).

Additionally, in our responses to Reviewer 1, we have provided a figure revealing that in the heart there is also some expression of *Vegfc* and *Vegfd* in cardiac fibroblasts and vascular mural cells (see **Figure A** above). However, as these cell populations were only 'contaminations' in our current study, thus not providing sufficient cell numbers for detailed analyses, we prefer to refrain from discussing this data in the current manuscript. Instead, we refer to our previous work, where we assayed cardiac Vegf-c, Vegf-d, and IL-1 β expression in the same model, and to scRNAseq studies published by others on immune cell and fibroblast subsets in the heart.

To better highlight these aspects, we now write in the introduction:

Line 80: "... inflammatory mediators, including IL1 β and IL6, produced in the heart notably by macrophages during pressure-overload (Heron et al, 2023a, Amrute 2024) "

Line 83: "Previous work has identified cardiac macrophages as a key source of Vegf-c (Heron et al, 2023a; Glinton et al, 2022), and we recently demonstrated that IL1 β promotes cardiac lymphangiogenesis post-TAC by stimulating macrophage production of Vegf-c (Heron et al, 2023b)."

Important points like the potential relevance of the upregulation of LYVE1 and CCL21 for increased immune cell transcytosis into LVs are not further addressed experimentally.

We agree that it is a shortcoming that we could not demonstrate experimentally whether immune cell adhesion or transmigration post-TAC were enhanced in BALB/c mice following increased lymphatic Lyve1 and/or Ccl21 expression. However, previous work has clearly demonstrated that loss of Lyve1 reduces macrophage uptake by lymphatics post-MI in C57 mice (ref Vieira *et al*, 2018) and conversely that mice lacking *Ccl21* have defective dendritic cell homing to lymph nodes (*Britschgi MR et al. Eur J Immunol.* 2010 PMID: 20201039) as do mice lacking its cognate receptor *Ccr7* (*Forster R et al. Nature Reviews Immunology* 2008 PMID: 18379575). Of note, scRNAseq analyses of cardiac immune cells post-TAC has revealed that *Ccr7* is expressed not only by dendritic cells but also by B cells and T cells in the heart (ref Martini E et al Circulation 2019). In our previous studies (Heron et al 2023a) we determined (by flow cytometry and IHC) that BALB/c mice have low B cell levels, but elevated CD4 and CD8 T cell levels at 8 weeks post-TAC. Potentially, treatment with antibodies blocking *Ccl21* and/or *Ccr7* could have confirmed the functionality of the altered cardiac LEC profiles in regard to their immune recruitment capacity (reduced lymphatic clearance upon *Ccl21* inhibition expected to cause an increase in cardiac B and T cells). However, systemic treatment is likely to influence lymphatic homing, not just in the heart, but in lymphoid organs globally, which may shift circulating immune populations and thus their cardiac recruitment, causing results 'independent' of cardiac lymphatics.

To better address this important issue in the future, we are developing intravital microscopy approaches that we expect will allow us to follow in real-time the recruitment and uptake of

immune cells into lymphatics in the beating heart. This however represents a major technical challenge, and we regret that it is not currently possible to answer the functional question *in vivo*, beyond immunohistochemical or flow cytometry data, such as generated by other groups using *Lyve1*-deficient C57 mice.

Unfortunately, this study could not shed more light on the recently much-discussed function of Reelin as a relevant lymphangiocrine factor in cardiac disease, as it was not changed transcriptionally after TAC and could not be investigated in immunohistochemistry.

We discussed this technical issue with a leading expert on Reelin, who confirmed that the current antibodies are not very sensitive to detect Reelin by immunohistochemistry. Additionally, as the changes observed in *Reln* expression were not significant, we chose not to pursue the matter further. We have in the revised manuscript removed the mention of failed Reelin immune-detection, but instead show in new **Suppl Fig. S8** a strain comparison of *Reln* expression levels, which indicate lower levels in cardiac LECs from adult BALB/c versus C57 mice.

Minor comments:

The cardiac EC populations were FACS-sorted for capillary LECs (CD45-PECAM1+PDPN+LYVE1+) yet only 15% of the ECs clustered in the LEC population? The authors did not comment on the high percentage of cells in the FACS-sorted population that did not express the desired markers.

We apologize for this erroneous statement. Indeed, we sorted in parallel cardiac blood vascular endothelial cells (BECs) and LECs. The BECs were present in the heart at much higher numbers as compared to LECs, as expected. Of note, while we could recover around 10 000 BECs/mouse heart, we only obtained 200-400 LECs per mouse in Balb/c mice, and even fewer in C57 mice. Thus, the low % of LECs in each experiment is not a mistake, but the direct consequence of the scarcity of these cells. To reach the experimentally required number of total cells per reaction (16 000 cells loaded to obtain the predicted 10 000 cells per reaction), we pooled BECs and LECs after separate sorting at a ratio of 3:1. Because of this operator-mediated post-sorting pooling, the relative % of BECs vs LECs are not physiologically-meaningful, and numbers are given to demonstrate that we performed the experiments in similar manner between sham and TAC mice and between strains.

As for the identity of the non-selected (non-EC) cardiac cells observed in the FACS dot plots, these are a mix of CD45⁺ immune cells and CD45^{neg} interstitial cells, including fibroblasts, smooth muscle cells, and non-viable cardiomyocyte debris. These other cell populations have been the focus of scRNAseq or single nuclear RNAseq analyses by many other groups, which is why we chose to only include vascular ECs in our molecular analyses.

We now write in the revised manuscript:

Line 123: *“To define the transcriptome of healthy murine cardiac vascular endothelial cells (ECs), we sorted by FACS cardiac CD45⁻/CD31⁺ cells from 20 sham-operated adult female BALB/c mice, further enriching the samples for *Lyve1* and Podoplanin (*Pdpn*)-expressing LECs vs. lymphatic-marker-negative BECs prior to scRNAseq »*

Line 677: *“Cardiac single cell suspensions were sorted by FACS (ARIA II, BD Biosciences). BECs were defined as live CD45⁻/CD31⁺/Lyve1⁻ cells, whereas LECs were identified as CD45⁻/CD31⁺/Lyve1⁺/Pdpn⁺ cells. A 1:3 mix of cardiac LECs and BECs was included in each sample”.*

Unfortunately, the suppl. Videos appeared to be not available.

We apologize for this problem, and can only confirm that videos were indeed uploaded to the journal submission website.

How was population LEC 2 defined as pre-collectors, how was this population distinguished from collector LECs?

The current definition of collectors is “*valved transporting lymphatic segments surrounded by smooth muscle cells*”. As previous immunohistochemical and whole mount analyses by our group and others have revealed that subepicardial lymphatics lack smooth muscle cells, the term “precollector” is reserved for these vessels, while the term “collector” is reserved for extra-cardiac lymphatics that extend from the cardiac surface to project into periaortic lymph nodes. These structures were not captured in our scRNAseq analyses. The main defining molecular criterion for LEC2, which identified them as non-capillary, were: low level of *Lyve1* and *Ccl21*, and elevated *Pdgfb*. This expression profile has consistently been reported in precollectors/collectors in other organs by several other teams. In the revised manuscript we now more clearly highlight the different segments in the photos, e.g. using different colour arrows to point out capillaries versus precollectors.

The quality of Fig. 2F in particular the confocal image is poor and does not allow the assessment of LYVE1 localization (i.e. button vs. zipper junctions).

Agree, the confocal image from whole mount samples was a bit blurry. We have replaced it with a new image that we hope better captures what we were trying to illustrate. The aim of this stain is not to show button vs zipper junctions by Lyve1, but to demonstrate ample Ccl21 expression in lymphatic capillaries, rather than in Lyve1^{low} precollectors in the heart.

Fig.3F, it appears surprising that the expression change of Nfatc1 is significant if e.g. Reln and Mmm1 are not, can the authors indicate significance levels?

BALB/c cardiac LEC *Nfatc1* mean expression level was reduced by 38% post-TAC as compared to sham, with a *p. adj* of 0.045 (please see *table S5*). In contrast, *Reln* expression level was non-significantly increased by 35% with a *p. adj* of 0.430, as was *Mmrn1* expression, which was increased by 27% with a *p adj.* of 0.340.

Fig. 4, the rarefication of valves is not convincingly obvious from these pictures. There seem to be more valves present in the preparation than the ones indicated by yellow arrows. On what basis were these valves specifically chosen and why? At first sight valve distance appears to be smaller in the TAC preparation. Why did the authors visualize valves by podocalyxin instead of the more common PROX1, or INTa9?

We have now added additional arrows to indicate valves in **Fig. 4d**. While we agree that the total number of valves is rather similar between groups in these images, our main finding was a difference in the frequency of valves in lymphatic capillaries. Whereas the sham group displayed many valves in both lymphatic capillaries (Lyve1^{high} vessels) and precollectors (Lyve1^{low} segments), the valves found in the TAC group are almost exclusively located in precollectors. To better distinguish valves in capillaries from valves in collectors, we have now changed the color of arrows pointing to capillary vs precollector valves in this panel.

As for the choice of valve markers, several groups, including Dr Tatiana Petrova (world-leading expert in lymphatic valves), use Podocalyxin as a reliable marker. We thus selected this one after an initial test with whole-mount staining for Prox1 and Integrin α9 gave inconclusive results. Indeed, commercial Prox1 antibodies have been found by us, and others, to be not ideal for wholemount imaging, and the best images of Prox1 ‘stained’ lymphatic valves in the

literature come from *Prox1* reporter mice. In support of our surprising findings of valves in lymphatic capillaries in murine hearts, there was a paper published, during the preparation of this revision, by another group using such a *Prox1* reporter mouse (C57 strain). In these published images, cardiac lymphatic capillary valves positive for Prox1 can clearly be seen (see **Figure F** beneath), similar to our Podocalyxin stains. However, the authors did not analyse cardiac lymphatic valves in this study, and we thus prefer not to cite this article in our current manuscript.

Figure F – Published images of cardiac wholemounts in *Prox1*-GFP transgenic mice from Pr. Kathleen Caron’s group (Balint et al. ATVB 2025 « Lymphatic Activation of ACKR3 Signaling Regulates Lymphatic Response After Ischemic Heart Injury » <https://doi.org/10.1161/ATVBAHA.124.32228>.) We have added *yellow arrows* to indicate capillary valves, while *white arrows* highlight precollector valves.

Why was a different type of graphic representation chosen for Fig. 4H compared to 4G?

Thank the Reviewer for this question. We chose this representation in **Fig 4h** as it is better suited to display genes with low level of expression (y axis max 1.0), as compared to violin plots used in **Fig 4g** for more highly expressed genes (y axis range 4-12). The umap

presentation allows for visual assessment of genes with very low, or variable, expression in order to appreciate diffuse subpopulation differences. We estimate that violin plots are not well suited for such weakly-expressed genes (see **Figure G** beneath). We have thus kept **Fig 4h** as originally shown for these potentially highly-relevant yet weakly-expressed genes.

Figure G – representation as violin plots of genes shown as umap plots in **Fig 4H** in the manuscript

Page 10, the first two paragraphs are partially redundant as these data had already been described previously.

We thank the Reviewer for this suggestion. Upon inspection, this section discusses for the first time the expression changes seen in the LEC1 subpopulation. We agree that the text may appear somewhat redundant as many of the transcriptional changes observed mirror those seen in the global LEC cluster. This is of course due to the fact that LEC1 constitute the main subpopulation. However, we estimate that it is essential to separately analyse capillary, precollector and valve LECs as this provides clues to how each compartment it altered in disease. Conversely, the discussion of the ‘global’ LEC changes post-TAC is an opportunity to discuss gene type alterations specific to LECs vs. those shared with BECs and vBECs. We thus apologize for the repetitive nature of the text, which appears to discuss several times the same genes. We have in the revised version of the manuscript attempted to improve the flow by removing some non-key elements to cut down overlap as much as possible.

Page 10, the last paragraph refers to Ccl21 and Ackr4 with reference to Fig. 4h, which however contains no data on these genes.

We thank the Reviewer for spotting this error: we removed mention of *Ackr4* and replaced the panel reference for *Ccl21a* with **Fig 4g**.

Fig. 6 Claudin5 should show a junctional staining pattern, i.e. be localized in LECs either to buttons or zippers, similarly in BECs TJs provide a fine distinct outline of the cells. The quality in Fig6 C is insufficient to make statements on the cellular localization or distinguish the staining in LECs from BECs.

We now provide more zoomed views to highlight LECs and BECs in these images (see new **Fig. 6c**), as well as **Fig. E** above.

References: Heron 2023b, citation is incomplete, please provide volume and page or equivalent information.

We thank the Reviewer for spotting this mistake. This reference is a preprint article available from BioRxiv. The full reference has been updated in the revised manuscript to indicate this:

“bioRxiv 2023.04.01.535056; <https://doi.org/10.1101/2023.04.01.535056> »

Referee #3

NGS performed and analyzed according to current standard methodology. Model is gold-standard for cardiology, and the use of 2 strains goes above the field standard Medical impact is high but work is needed to draw some more general conclusions

This manuscript describes a scRNAseq analysis of cardiac lymphatics in the gold-standard non-ischemic heart failure model. The model is applied in two different strains, known to yield different severity outcomes. Lymphatic regulation of pathology is a topic of very high interest in cardiology and cardioimmunology. As far as I know, this is the first study attempting an scRNAseq analysis of cardiac lymphatics, thus yielding potentially high-interest results and a useful resource once the underlying data is published. The procedures and methodologies used for the scRNAseq are standard, and appropriate, as is the identification of lymphatic endothelial cells. Indeed all my comments (see below) are limited on data description and interpretation.

It is commendable that the processed gene expression data are included in the supplementary materials. This study will have great value as a resource. Reducing the hurdles required to access and process the data by researchers wishing to work on them is likely to amplify the study's impact.

We thank the Reviewer for encouraging comments.

Pg5 "Reclustering analyses allowed investigations of BEC subpopulations (see Table S3 for marker genes), revealing elevated Cldn5 expression only in two clusters: BEC9 (Suppl. Fig. S3c), enriched for arteriolar markers (e.g. Id1, Notch-target gene Hey1), and BEC6, enriched for angiogenic markers (e.g. Aplnr2, Apelin receptor

2)." The logic of what is done here and why it was decided to do it must be made more clear.

Agree. We now write in the revised manuscript:

Line 183: *Reclustering analysis of the BEC population was performed to further investigate the expression of vascular barrier components in BEC subpopulations (see **Table S3** for identified marker genes). We found that among the identified eight BEC subpopulations, elevated Cldn5 was present only in two subclusters: BEC9 (**Suppl. Fig. S3c**), enriched for arteriolar markers (e.g. Id1, Notch-target gene Hey1), and BEC6, enriched for angiogenic markers (e.g. Aplnr2, Apelin receptor 2). Similarly, preferential arteriolar localization of Cldn-5 has been demonstrated in mouse kidneys (Morita et al, 1999), whereas most blood vessels in the brain depend on Cldn-5 for maintenance of the blood brain barrier (Nitta et al, 2003).*

In the description of the results of Fig 3, the authors have selected to outline the findings with no concurrent speculative comments on the possible functional significance of the findings. Whilst this is formally correct, it would aid the readability of the manuscript if speculative (and clearly marked as such) comments were added in Results, after each section. The amount of detail emerging from a scRNAseq analysis (especially in a "novel" tissue) is overwhelming for most readers to be able to defer the discussion of the findings in the Discussion section.

The results shown in subsequent figures are described in a manner that includes more function-related comments, rendering them more readable.

We thank the Reviewer for this most helpful suggestion. We fully agree and have in the revised manuscript included a brief discussion in the results section to better highlight the functional implications of the altered gene expressions observed. For example, we now write:

Line 251: *"In addition, Aplp2 (Amyloid-like protein 2) was downregulated (**Fig. 3e, Table S10**). The encoded protein stimulates endocytosis, notably of MHC-I molecules(Tuli et al, 2009),*

thus its reduction may influence MHC-I plasma membrane levels in lymphatics. These shifts indicate that LEC-mediated immune cell recruitment and antigen-presentation may be altered post-TAC."

Line 265: *"This indicates that the vascular barrier in capillary and/or precollector lymphatics may be altered following pressure-overload. Of note, previous work in other organs has demonstrated that inflammation reduces Cldn-5 expression in lymphatics, contributing to increased lymphatic permeability (Kajiya et al, 2012)."*

Line 271: *"These changes, which may relate to alterations in LEC subpopulations, indicate a shift in the transcription factor landscape in LECs expected to influence lymphatic activity and function following chronic pressure-overload. Of note, key Prox1 target genes, including Flt4 and Cdkn1c, were not reduced, while Lyve1, as mentioned above, was increased (Fig 3f), indicating that Prox1 activity is still sufficient to drive expression of these genes.'*

Line 280: *'It remains to be determined whether the activated lymphatics post-TAC may contribute to local remodelling of the extracellular matrix during pressure-overload.'*

Line 313: *'These changes suggest that BEC-mediated lipid uptake may be enhanced during development of HF. Indeed, maintenance of cardiac FA uptake has been shown to limit myocardial metabolic remodelling in pressure-overload (Umbarawan et al, 2018).'*

Pg10 "finding an average intervalve distance of 200 µm in healthy mice (Fig. 4e). In the remaining valves in TAC-operated mice this valve distance was unaltered. " Whilst the data in Fig 4e is quite clear, the phrase above is somewhat misleading - please rephrase.

Thank you. We now write in the revised manuscript:

Line 355: *"To further investigate lymphatic valve properties in the heart, we analysed the distance between consecutive valves, finding an average intervalve distance of 200 µm in healthy mice (Fig. 4e). In the remaining valves in TAC-operated mice, this valve distance was unaltered, suggesting that insufficient valvulogenesis, not valve drop-out, may be involved. Taken together, our data indicate that the newly expanded capillary lymphatic network developing post-TAC may lack signals to drive lymphatic valvulogenesis, which is very likely to restrict lymphatic drainage capacity despite elevated lymphatic density in the heart."*

The Ackr2 and Mrc1 upregulation (Fig S7a,b) may be an attempt at immune deviation from a "classical" inflammatory response. It would be worth discussing more.

We thank the Reviewer for this appreciation. Although these genes indeed appear altered in **Suppl. Fig S7**, the changes were not significant, which is why we have not discussed these genes further. Moreover, while the role of Ackr2 in lymphatics has been previously demonstrated by others, the potential functional impact of Mrc1 in lymphatics remains to be addressed. However, to better acknowledge that inflammation-induced changes in lymphatics observed in our study may have both beneficial (adaptive) and maladaptive effects, we have added a phrase to the conclusion section (see *beneath*).

The utilization of a less severe form of TAC (by using another strain) is an innovative solution, and commendable. The lack of major differences in the female TAC vs sham - in the C57 version of the model, where disease severity is reduced- suggest that LEC presence is associated with disease severity. This, in turn, means that LEC are either a (component of a) driver of disease, or a negative feedback loop against disease; in both cases LEC would not appear changed in low-severity disease.

This message is quite novel and substantial, and of clear interest to cardiologists and immunologists beyond those interested specifically in Lymphatics - but it does not emerge easily in the current write-up. Indeed, whilst the manuscript is great in describing the detailed aspects of LEC biology in cardiopathic and healthy mice, it could substantially increase its impact in the cardiology and

immunology community, by drawing some more general system-level conclusions, seeing the forest instead of the trees.

Thank you for this valuable suggestion. We have expanded the conclusion section with the following paragraph to better open our findings to other fields of research:

Line 605: *“The molecular alterations found in our study in cardiac LECs during development of HF appear to include both adaptive changes, set to improve inflammatory resolution (upregulated chemokines, adhesion molecules, and MHC molecules), but also potential maladaptive changes that may contribute to lymphatic transport dysfunction (loss of lymphatic valves, reduction of Cldn5, Tjp1). Our data indicate that many of these changes may be caused by inflammatory mediators, including IL1 β . Previous work investigating lymphatics in other organs during inflammation have similarly reported both beneficial and deleterious changes induced by immune mediators (Baluk et al, 2007; Aebischer et al, 2014; Scallan et al, 2016; Harlé et al, 2021; Czepielewski et al, 2021; Li et al, 2022; Kim et al, 2023). It remains to be determined whether anti-inflammatory treatments may improve lymphatic function in CVDs. Of note, some anti-inflammatory treatments are currently investigated in clinical trials in patients with lymphedema (Rockson et al, 2018).”*

8th Oct 2025

Dear Prof. Brakenhielm,

Thank you for submitting your revised study, and please accept my apologies for the delay in getting back to you as one referee needed more time to provide their report. We have now received the reports from the referees. As you will see below, they are satisfied with the revisions, and I will therefore be able to accept your manuscript once the following editorial concerns are addressed:

1/ Manuscript text:

- Please indicate in track changes mode any new modification in the text.
- Authors should be listed with first name, then last name (as in the submission system).
- We can accommodate a maximum of 5 keywords, please adjust accordingly.
- The word count of manuscript sections should be removed from the title page.
- The conflict of interest statement provided on the title page needs to have its own section "Disclosure and Competing Interests Statement", placed after Acknowledgments.
- Brief Summary should be removed from the manuscript.
- Please remove the figures from the manuscript file (the legends should stay).
- Materials and Methods should be changed to Methods. Supplementary methods should be integrated in the Methods section in the main manuscript.
- In the methods, please provide the following information:
 - o Animals: housing and husbandry conditions
 - o Cells: whether the cells were tested for mycoplasma contamination.
 - o Statistics: statement on sample size, exclusion/inclusion criteria, randomization and blinding.
- The data availability section should be placed before the Acknowledgements, and should only list the original datasets produced in this study. All datasets must be public before acceptance of the manuscript. Please clarify "Submitter states that missing raw files are due to file loss." for dataset GSE289738.
- Please remove the Authors Contributions from the manuscript and use the free text boxes beneath each contributing author's name in our system to add specific details on the author's contribution.
- Funding should be merged with Acknowledgements. The information provided should be the same as in the submission system (currently, the following items are missing in the system - the Normandie Doctoral School (EdNBISE), "Allocation 50% Normandie Recherche, the Chair of Excellence program "Lymphcosign" from the Normandy Region (RIN Recherche), grant from the GCS G4 and the Normandy region (FHU CARNAVAL), DFG project #453989101-CRC1525, generalized institutional funds (Inserm UMR1096) from French Inserm, Rouen University (BQRI 2022, 2023), and targeted funding from the Normandy Region (CPER 2021), from EU-Normandy region co-funds (RIN 2018 SINGLE C and 2019 7D MICROSCOPY): "L'Europe s'engage en Normandie avec le Fonds Européen de Développement Régional).

2/ Figures:

- Movie EV2 is not referenced in the manuscript text, please correct.
- There is an Excel file with Tables S2-S13 and also a PDF with the same tables (this one is not needed); "Tables S2-S13" nomenclature should not be used as these are all datasets which should be uploaded as separate files Dataset EV1-EV12; the callouts in the manuscript should also be updated as well as each source file. Legends have to be provided in a separate tab
- We replaced Supplementary Information with Expanded View (EV) Figures and Tables that are collapsible/expandable online. EV Figures should be cited as 'Figure EV1, Figure EV2" etc... in the text and their respective legends should be included in the main text after the legends of regular figures. For the figures that you do NOT wish to display as Expanded View figures, they should be bundled together with their legends in a single PDF file called *Appendix*, which should start with a short Table of Content. Appendix figures should be referred to in the main text as: "Appendix Figure S1, Appendix Figure S2" etc. The ToC on the title page need page numbers, legends should be removed. The tables that are not present in the Appendix file need to have the legends removed from the Appendix; only Table S1 has been provided in the PSF, needs to be updated to Appendix Table S1 and its legend can stay in the file; Tables S2-S13 are not part of the Appendix file, but if they need to be, then they should be Appendix Table S2-S13; Figures need to be updated to Appendix Figure S1-S10 throughout the file and in the manuscript text.
- The appendix file needs to be reformatted at a higher resolution. In particular, Appendix Figure S7 is pixelated.
- The movie legends should be removed from the Appendix; Each legend should be provided in a readme.txt file and the should be zipped together with its corresponding movie so that we have folder per movie uploaded. The correct nomenclature is Movie EV1, Movie EV2 (this should be corrected in all places: source file names, titles in the system, legends, zip folders, callouts)
- Please address the queries from our data editors in the figure legends:
 1. Please note that the exact p values are not provided in the legend of figure 4E
 2. Please indicate the statistical test used for data analysis in the legends of figures 3D, 6D
 3. Please note that the box plots need to be defined in terms of minima, maxima, centre, bounds of box and whiskers, and percentile in the legends of figures 1F, 2E; 3D, F; 4G; 5F, G, I
 4. Please note that information related to n is missing in the legends of figures 2E, 5G, I
 5. Please note that the error bars are not defined in the legend of figure 4E

3/ Source Data: Thank you for providing Source Data. Please upload them as 1 folder per figure; each panel in each folder needs to be clearly labeled.

4/ Please provide a complete author checklist, which you can download from our author guidelines (<https://www.embopress.org/page/journal/17574684/authorguide#submissionofrevisions>). Please insert information in the checklist that is also reflected in the manuscript. The completed author checklist will also be part of the RPF.

5/ Our journal encourages inclusion of *data citations in the reference list* to directly cite datasets that were re-used and obtained from public databases. Data citations in the article text are distinct from normal bibliographical citations and should directly link to the database records from which the data can be accessed. In the main text, data citations are formatted as follows: "Data ref: Smith et al, 2001" or "Data ref: NCBI Sequence Read Archive PRJNA342805, 2017". In the Reference list, data citations must be labeled with "[DATASET]". A data reference must provide the database name, accession number/identifiers and a resolvable link to the landing page from which the data can be accessed at the end of the reference. Further instructions are available at .

6/ Please provide The paper explained in the manuscript text file. EMBO Molecular Medicine articles are accompanied by a summary of the articles to emphasize the major findings in the paper and their medical implications for the non-specialist reader. Please provide a draft summary of your article highlighting

7/ Every published paper now includes a 'Synopsis' to further enhance discoverability. Synopses are displayed on the journal webpage and are freely accessible to all readers. They include a short stand first (maximum of 300 characters, including space) as well as 2-5 one-sentences bullet points that summarizes the paper. Please write the bullet points to summarize the key NEW findings. They should be designed to be complementary to the abstract - i.e. not repeat the same text. We encourage inclusion of key acronyms and quantitative information (maximum of 30 words / bullet point). Please use the passive voice. Please attach these in a separate file or send them by email, we will incorporate them accordingly.

Please also suggest a visual abstract to illustrate your article as a PNG file 550 px wide x 300-600 px high. A cropped portion of this image will serve as thumbnail for the table of content on our webpage.

8/ As part of the EMBO Publications transparent editorial process initiative (see our Editorial at <http://embomolmed.embopress.org/content/2/9/329>), EMBO Molecular Medicine will publish online a Review Process File (RPF) to accompany accepted manuscripts.

This file will be published in conjunction with your paper and will include the anonymous referee reports, your point-by-point response and all pertinent correspondence relating to the manuscript. Let us know whether you agree with the publication of the RPF and as here, if you want to remove or not any figures from it prior to publication.

I look forward to receiving your revised manuscript.

Yours sincerely,

Lise Roth

***** Reviewer's comments *****

Referee #1 (Comments on Novelty/Model System for Author):

The authors have performed rigorous analysis of 2 mouse models, which they had previously reported. Currently this is the best that researchers can do. However, whether this is clinically relevant is not known.

Referee #1 (Remarks for Author):

The authors have addressed all my questions. The manuscript is consequently much better. I do not have any further questions.

Referee #2 (Remarks for Author):

The revised manuscript EMM-2025-21696-v2 features significantly rewritten sections with rephrased statements and contextual clarifications, notably the clarification of the sorting strategy and specific reference to the origin of inflammatory cytokines, which enhance overall clarity and readability. Specifically, the proposed mechanistic hypothesis is now more succinctly described, making it more accessible to readers. The inclusion of additional preliminary data in support of the model presented in the point-by-point response to the reviewers is also commendable. Furthermore, the revised images in Figure 2F provide a clearer illustration of the reported findings.

One minor suggestion: when referring to the protein, "VEGF-C" should be written in uppercase letters (e.g., lines 83 and 85).

The authors addressed the remaining editorial issues.

4th Nov 2025

Dear Prof. Brakenhielm,

Thank you for sending your revised files. I am pleased to inform you that your manuscript is accepted for publication and is now being sent to our publisher to be included in the next available issue of EMBO Molecular Medicine.

If you have any questions, please do not hesitate to contact the Editorial Office. Thank you for your contribution to EMBO Molecular Medicine!

Yours sincerely,

Lise
